# Moving beyond bimetallic-alloy to single-atom dimer atomic-interface for all-pH hydrogen evolution

Ashwani Kumar[1,2,9], Viet Q. Bui[1,2,9], Jinsun Lee[1,2], Lingling Wang[1,2], Amol R. Jadhav[1], Xinghui Liu [1,2], Xiaodong Shao[1,2], Yang Liu[1,2], Jianmin Yu[1,2], Yosep Hwang[1,2], Huong T. D. Bui[1,2], Sara Ajmal[1,2], Min Gyu Kim [3], Seong-Gon Kim [4], Gyeong-Su Park[5], Yoshiyuki Kawazoe[6] & Hyoyoung Lee [1,2,7,8]✉

Single-atom-catalysts (SACs) afford a fascinating activity with respect to other nanomaterials for hydrogen evolution reaction (HER), yet the simplicity of single-atom center limits its further modification and utilization. Obtaining bimetallic single-atom-dimer (SAD) structures can reform the electronic structure of SACs with added atomic-level synergistic effect, further improving HER kinetics beyond SACs. However, the synthesis and identification of such SAD structure remains conceptually challenging. Herein, systematic first-principle screening reveals that the synergistic interaction at the NiCo-SAD atomic interface can upshift the d-band center, thereby, facilitate rapid water-dissociation and optimal proton adsorption, accelerating alkaline/acidic HER kinetics. Inspired by theoretical predictions, we develop a facile strategy to obtain NiCo-SAD on N-doped carbon (NiCo-SAD-NC) via in-situ trapping of metal ions followed by pyrolysis with precisely controlled N-moieties. X-ray absorption spectroscopy indicates the emergence of Ni-Co coordination at the atomic-level. The obtained NiCo-SAD-NC exhibits exceptional pH-universal HER-activity, demanding only 54.7 and 61 mV overpotentials at $-10$ mA cm$^{-2}$ in acidic and alkaline media, respectively. This work provides a facile synthetic strategy for SAD catalysts and sheds light on the fundamentals of structure-activity relationships for future applications.

[1] Center for Integrated Nanostructure Physics (CINAP), Institute for Basic Science (IBS), Sungkyunkwan University, Suwon 16419, Korea. [2] Department of Chemistry, Sungkyunkwan University (SKKU), Suwon 16419, Republic of Korea. [3] Beamline Research Division, Pohang Accelerator Laboratory (PAL), Pohang University of Science and Technology, Pohang 37673, Republic of Korea. [4] Department of Physics and Astronomy and Center for Computational Sciences, Mississippi State University, Mississippi State, MS 39762, USA. [5] Department of Materials Science and Engineering, and Research Institute of Advanced Materials, Seoul National University, Seoul 08826, Republic of Korea. [6] New Industry Creation Hatchery Center, Tohoku University, Sendai 980-8579, Japan. [7] Department of Biophysics, Sungkyunkwan University, Suwon 16419, Republic of Korea. [8] Creative Research Institute, Sungkyunkwan University, Suwon 16419, Republic of Korea. [9]These authors contributed equally: Ashwani Kumar, Viet Q. Bui. ✉email: hyoyoung@skku.edu

Owing to the high energy storage density and zero carbon emission, hydrogen ($H_2$) fuel from water electrolysis has been regarded as the most promising alternative to fossil fuels[1,2]. Strikingly, the hydrogen evolution reaction (HER) plays an essential role in electrochemical water splitting for energy conversion. Various water electrolyzers demand different pH values of the electrolyte, such as proton exchange membrane electrolysis in strong acid, seawater electrolysis in neutral medium, and commercial water electrolysis in strong base[3]. To meet the above requirements, pH-universal HER catalysts with superior performance in both acidic and alkali media are highly regarded; however, they are barely accessible[4]. Platinum (Pt) and Pt-based catalysts are still the best-known pH-universal HER electrocatalysts, but their limited availability and high cost hinder their large-scale applications[5]. Therefore, exploring non-precious metal-based electrocatalysts with Pt-like pH-universal HER activity is highly desired, yet challenging.

To date, numerous earth-abundant HER electrocatalysts including oxides, hydroxides, alloys, phosphides, nitrides, sulfides, and their hybrids, have been identified as promising HER catalysts[3,6–11]. However, satisfactory Pt-like activity has been seldomly achieved and only a few of them can be simultaneously active in both acidic and alkaline media[3]. Recently, single-atom catalysts (SACs) with nearly 100% atom economy and unique electronic properties compared to their regular nanoparticle (NPs) counterparts have attracted immense scientific attention in the field of photo/electro/thermo-catalysis[12–14]. Most SACs contain isolated single metal sites coordinated with the neighboring nitrogen atoms in carbon matrix (M-NC), which are only capable of catalyzing simple elementary reactions[15]. Due to the simplicity of the single-atom center, the possibilities for further modification of the active site in SACs are extremely limited, hindering their wide range of applications[16]. In response to this, recent exploration suggests that tuning the coordination site to sulfur/phosphorus or by introducing secondary metal atom to construct metal–metal dual atom sites (single-atom dimer: SAD) can further modulate the electronic structure of SACs and boost their intrinsic activity, attributed to the unique atomic interface and synergistic effect of dual-metal site[17–19]. Recently, Fe-Co, Zn-Co, and Ni-Fe dual-metal sites have been demonstrated as efficient bifunctional oxygen electrocatalysts (oxygen reduction reaction (ORR)/oxygen evolution reaction (OER) and for $CO_2$ reduction reaction ($CO_2$RR)[20–22]. Zhang et al.[23] synthesized noble metal-based Pt-Ru dimer using the advanced atomic layer deposition technique and showed comparable HER performance to commercial Pt in acidic media. However, the evidence for the formation of a single metal–metal bond from X-ray absorption spectroscopy (XAS) was unclear due to the existence of additional atomic clusters in the sample[23]. Although SADs are explored towards ORR/OER/$CO_2$RR, a generalized cost-effective and versatile strategy to fabricate pH-universal low-cost HER catalyst with targeted dimeric sites at atomic precision along with appropriate identification of the dimeric structure and deeper understanding of the dual-metal atom synergism has never been achieved and remain elusive.

Herein, we report a transition metal-based SAD (TM-SAD) atomic interface, which can efficiently catalyze complex HER in a wide pH range (0–14). At first, systematic density functional theory (DFT) screening reveals that among various TM-SADs, the synergistic interaction between the Ni-Co at the atomic level in the SAD configuration can significantly upshift the d-band center, thereby accelerating water dissociation and boosting pH-universal HER activity. Motivated by DFT prediction, we develop a facile methodology to synthesize NiCo-SAD on N-doped carbon (NiCo-SAD-NC) via in situ trapping of targeted metal ions in the polydopamine sphere followed by annealing with precisely controlling the

N-moieties. State-of-the-art techniques including X-ray absorption near edge structure (XANES), extended X-ray absorption fine spectra (EXAFS), aberration-corrected scanning transmission electron microscopy (AC-STEM), and X-ray photoelectron spectroscopy (XPS) along with theoretical calculation are employed to analyze the detailed structure of the NiCo-SAD-NC, which reveal the emergence of Ni-Co bond with strong electronic coupling at the atomic level. The as-prepared NiCo-SAD-NC exhibits an exceptional pH-universal HER activity, which requires only 54.7 and 61 mV overpotential at $-10\,\mathrm{mA\,cm^{-2}}$ in acidic and alkaline media, respectively, outperforms the NiCo-NP and monoatomic Ni/Co-SACs. The activity of NiCo-SAD-NC is comparable/superior to commercial Pt-C/Pt-SAC, as well as superior to most of the recently reported TM-based single-atom electrocatalysts.

## Results

**DFT screening of TM-SADs for HER.** We first employed DFT calculations for screening various bimetallic TM-SAD structures stabilized on the N-doped carbon for HER based on their electronic parameters. To evaluate the stability of TM-SAD structures (heteronuclear: CoCu-SAD-$N_6$C, NiCo-SAD-$N_6$C, CoFe-SAD-$N_6$C, CoMn-SAD-$N_6$C; homonuclear: CuCu-SAD-$N_6$C, NiNi-SAD-$N_6$C, CoCo-SAD-$N_6$C, FeFe-SAD-$N_6$C, MnMn-SAD-$N_6$C), we calculated the formation energies ($E_f$), displayed in Fig. 1a and Supplementary Table 1. All the selected TM-SAD structures were thermodynamically stable, revealed from their respective negative values of $E_f$, which also exhibited an increasing trend with the number of outermost $3d$ orbital valence electrons. Interestingly, we found that there was a consistent trend for the average Mulliken charges distribution ($\Delta q$) of TM-SAD center with the formation energy of TM-SAD-$N_6$G structures, except for the CoCu/CuCu-SAD-$N_6$C (Fig. 1a and Supplementary Fig. 1a). The formation energy increased with the increase of $\Delta q$ in the homo/heterostructures of SAD, suggesting that higher $\Delta q$ confirmed the higher thermodynamically stable SAD structure[24]. In addition, the water adsorption energy exhibited a linear correlation with the Mulliken charge transfer from the metal active site, suggesting that higher charge transfer from the active site exhibited stronger water adsorption strength (Supplementary Fig. 1b). The differential charge density distribution revealed that a significant charge resides between the metal atoms and N coordination, with charge accumulated at the center and depleted at the metal center, consistent with the Mulliken charge analysis (Fig. 1b). The TM atoms with lower electronegativities showed a higher tendency to donate more electrons to N atoms and form stronger TM-N bonds in the SAD structure. Meanwhile, the projected partial density of states of various TM-SAD showed that the $d$-orbitals of the TM atoms were mainly distributed around the Fermi level (Supplementary Fig. 2). We further investigated whether there was a correlation between the $d$-band center and formation energy, and we found a poor linear relationship between them (Fig. 1a); however, $3d$ band center of SAD displayed a linear correlation with kinetic barrier of $H_2O$ dissociation (Fig. 1d). Especially, the comprehensive $d$-band centers of Co and Ni atoms in the NiCo-SAD-$N_6$C ($-0.87$ eV) were the nearest to the Fermi level compared to the other TM-SADs, except for NiNi-SAD-$N_6$C, demonstrating its superior ability to facilitate water dissociation and enhanced proton adsorption, beneficial for HER[25]. In addition, a significant overlap between the Co $3d$ orbitals of NiCo/CoMn-SAD-$N_6$C and Fe $3d$ orbitals of FeFe-SAD-$N_6$C with the O $2p$ orbitals of adsorbed $H_2O$ near the Fermi level consolidated that the Co and Fe atoms in their respective TM-SADs acted as the principal active site for activating the $H_2O$ dissociation, boosting the HER (Supplementary Fig. 3).

For the thermodynamic assessments of these TM-SAD structures towards HER, we calculated the kinetic energy barrier for the $H_2O$

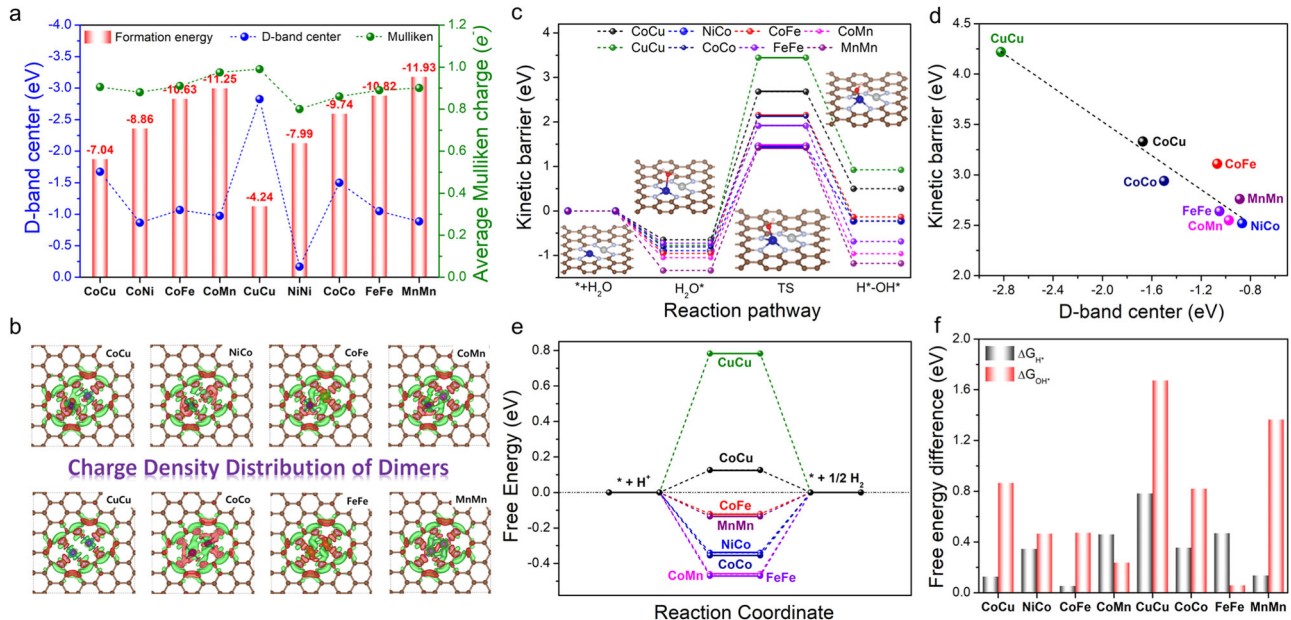

**Fig. 1 High-throughput screening of various transition metal-based SAD (TM-SAD) for HER. a** Formation energies corresponding to the charge depletion (Mulliken charge) and $d$-band center of various TM-SAD-$N_6$C. **b** Different charge density distribution of TM-SAD-$N_6$C. The charge depletion and accumulation are denoted by green and red colors, respectively. **c** Minimum energy paths of water splitting reactions on TM-SAD-$N_6$C (blue: Co and gray: Ni). **d** Linear correlation between the $d$-band center and kinetic energy barrier. **e** Free energy diagram of hydrogen adsorption and **f** free energy changes of the hydronium ($\Delta G_{H^*}$) and hydroxide ($\Delta G_{OH^*}$) desorption step for TM-SAD-$N_6$C.

dissociation (Fig. 1c). After the optimization of all the possible structures for the initial/final states in the $H_2O$ splitting process, we recognized that the NiNi-SAD-$N_6$C was inactive and incapable of $H_2O$ adsorption; therefore, further examination was ignored (Supplementary Discussion 1). Figure 1c revealed that the NiCo-SAD-$N_6$C exhibited the lowest transition state energy barrier for $H_2O$ dissociation compared to other TM-SADs, suggesting that the NiCo-SAD-$N_6$C with the highest $H_2O$ dissociation rate can be regarded as a promising candidate for alkaline HER (Supplementary Table 2a). Interestingly, we found that the $H_2O$ dissociation energy barrier for the TM-SADs decreased with the increase in the $3d$ band center, further corroborating the reactivity trends (Fig. 1d). In addition, we also extended our calculation for alkaline HER activity with other metal (Cr, Mo, and Zn)-based SAD, summarized in Supplementary Table 2b. As revealed in Supplementary Table 2b, the NiCo-SAD-$N_6$C still exhibited the best alkaline HER activity compared to other SAD based on the evaluated kinetic barrier of water dissociation and free energy of *H and *OH desorption. In addition, for the Co and Ni single-atom sites (Co-SA-$N_4$C and Ni-SA-$N_4$C), the calculation revealed that the hydronium (H*) and hydroxide (OH*) species auto-recombined to generate $H_2O$ molecule after the optimization process in the final state of $H_2O$ dissociation, suggesting that only Co and Ni single-atom sites were extremely sluggish for $H_2O$ dissociation, further validating the beneficial role of synergistic interaction at the Ni-Co atomic interface in the SAD configurations. To evaluate the potential-determining-step ($U_L$) for HER, the free energies for the desorption of H* and OH* were calculated, where the $|U_L|$ should be equal to the larger value among the $|\Delta G_{OH^*}|$ and $|\Delta G_{H^*}|$. Figure 1e, f revealed that for CoCu-SAD-$N_6$C, NiCo-SAD-$N_6$C, CoFe-SAD-$N_6$C, CuCu-SAD-$N_6$C, CoCo-SAD-$N_6$C, and MnMn-SAD-$N_6$C, the $|\Delta G_{OH^*}|$ was more uphill than that of $|\Delta G_{H^*}|$, suggesting the $|\Delta G_{OH^*}|$ as the $U_L$. In contrast, the $|U_L|$ for CoMn-SAD-$N_6$C and FeFe-SAD-$N_6$C were determined by the $|\Delta G_{H^*}|$. By comparing the $|U_L|$ of the TM-SADs, three potential candidates with the

lowest theoretical overpotentials were observed in the following order: NiCo-SAD-$N_6$C (0.460 eV) < CoMn-SAD-$N_6$C (0.465 eV) < CoFe-SAD-$N_6$C (0.472 eV). Furthermore, we calculated the kinetic energy barrier for $H_2O$ dissociation on NiCo-NP (111) surface and compared it with NiCo-SAD-$N_6$C (Supplementary Fig. 4). The NiCo-NP exhibited much higher barrier for $H_2O$ dissociation than NiCo-SAD-$N_6$C, suggesting that the Ni-Co atomic interface in the SAD exhibited much superior intrinsic activity compared to the NP counterpart. In a nutshell, the synergistic interaction at the Ni-Co atomic interface in the unique SAD structure can accelerate water dissociation and effectively adsorbs the proton, expected to achieve exceptional alkaline and reasonable acidic HER activity based on the DFT calculations.

**Synthesis and structural characterization.** As the single metal atoms in SACs are stabilized by their coordination with the N moieties and in the absence of N, the metal atoms tend to aggregate into cluster/NPs on the carbon matrix[15], we posited that by precisely controlling the amount of N, single metal–metal dimeric sites could be generated at an optimum amount of N moieties. Following our hypothesis and inspired by the theoretical predictions, we developed a facile synthesis strategy to obtain NiCo-SAD-NC by precisely controlling the N moieties (Fig. 2). Briefly, the synthetic strategy of NiCo-SAD-NC and NiCo-NP-NC could be divided into two steps, as schematically illustrated in Fig. 2. First, the targeted metal ions ($Ni^{2+}$ and $Co^{2+}$) were in situ trapped into the poly-dopamine sphere ($Ni^{2+}$-$Co^{2+}$@polydopamine) during the self-polymerization of dopamine under continuous stirring at alkaline pH value. The field-emission scanning electron microscopy (FESEM) image revealed the uniform spherical morphology of $Ni^{2+}$-$Co^{2+}$@polydopamine (average diameter: 150–200 nm) (Supplementary Fig. 5a). Subsequently, after annealing at 800 °C under vacuum conditions, the $Ni^{2+}$-$Co^{2+}$@polydopamine precursor was carbonized and transformed into NiCo-NP encapsulated into

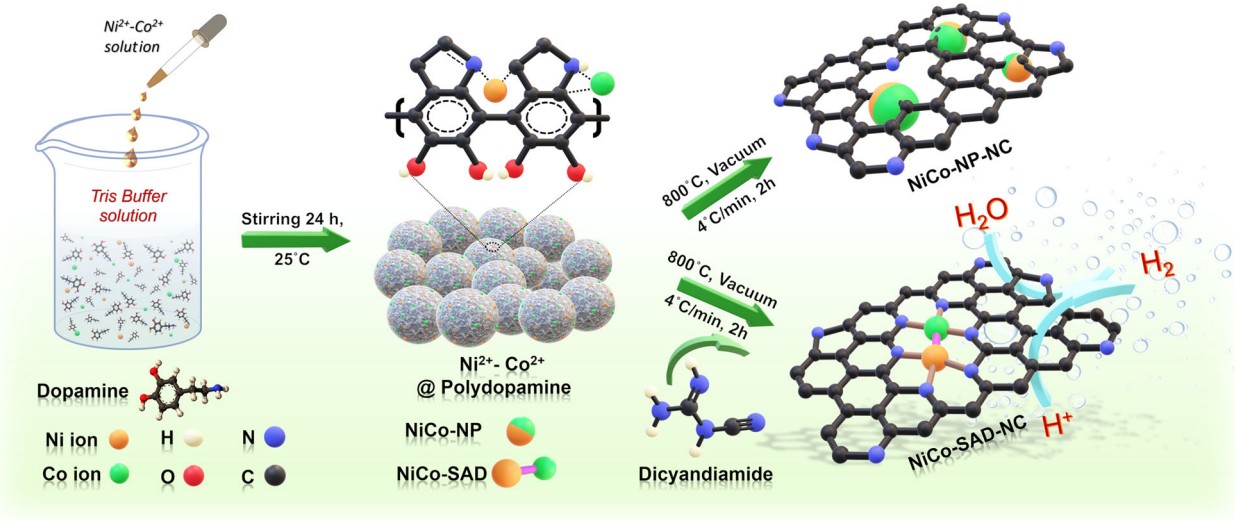

**Fig. 2 Synthesis strategy for single-atom dimer.** Schematic illustration of the synthetic strategy for the NiCo-SAD-NC and NiCo-NP-NC: (I) trapping metal ions (Ni and Co) into the polydopamine (PDA) via self-polymerization of dopamine; (II) generation of NiCo-NP-NC and NiCo-SAD-NC through calcination treatment upon modulating nitrogen containing precursor (dicyandiamide) amount.

N-carbon matrix (NiCo-NP-NC), while keeping the morphology intact (Supplementary Fig. 5b). The X-ray diffraction (XRD) pattern of the NiCo-NP-NC confirmed the aggregation of the metal ions to form NiCo-alloy NPs (JCPDS: 98-101-8308) (Fig. 3a)[26]. Energy-dispersive X-ray spectroscopy (EDS) and inductively coupled plasma atomic emission spectrometry (ICP-AES) analysis of NiCo-NP-NC revealed a 1 : 1 ratio of Ni to Co with total metal loading of 7.55 wt.%, whereas the N content of <1 at.% indicated that the amount of N moieties in the $Ni^{2+}$-$Co^{2+}$@polydopamine sphere were insufficient to trap the Ni/Co metal atoms (Supplementary Fig. 6a and Supplementary Tables 3 and 4). To obtain the NiCo-SAD-NC by trapping the dual-metal sites with N coordination, we blended the $Ni^{2+}$-$Co^{2+}$@polydopamine precursor with dicyandiamide (N containing organic molecule) in different ratios (1 : 1, 1 : 3, 1 : 5, and 1 : 7), followed by annealing at 800 °C under vacuum condition. The XRD pattern in Supplementary Fig. 7a revealed that upon mixing with dicyandiamide, the diffraction peaks attributed to NiCo-alloy gradually disappeared at a ratio of 1 : 7, matching well with that of NC along with fluffy porous morphology, denoted as NiCo-SAD-NC (Fig. 3a and Supplementary Fig. 5c). The slight shift of the G (002) peak for NiCo-SAD-NC compared to NC suggested the creation of carbon lattice defects due to the incorporation of N and metal atoms[27]. The NiCo-SAD-NC also exhibited a 1 : 1 ratio (Ni : Co) with total metal loading of 7.28 wt.%, while around 17.5 at.% of N content was the optimum amount to trap the Ni-Co atomic sites (Supplementary Fig. 6b and Supplementary Tables 3 and 4). The XRD pattern of Ni-SA-NC and Co-SA-NC were also similar to NC without any NPs peaks (Supplementary Fig. 7b).

Transmission electron microscopy (TEM) image of the NiCo-NP-NC revealed that the NiCo-alloy NPs (diameter: 15–20 nm) were uniformly encapsulated into the carbon matrix (Supplementary Fig. 8a). The high-resolution TEM image along with the corresponding selected area electron diffraction pattern confirmed the lattice spacing of 0.21 nm correspond to the (111) plane of NiCo-alloy (Supplementary Fig. 8b). The STEM high-angle annular dark-field (HAADF) image with EDS elemental map of the NiCo-NP-NC showed uniform distribution of the Ni, Co, and N (Supplementary Fig. 8c). Contrarily, no obvious NiCo-alloy NPs were spotted after the introduction of a sufficient amount of N, suggesting both Ni and Co species were atomically dispersed in the NiCo-SAD-NC (Supplementary Fig. 8d–f). In addition, the Raman spectra of both NC and NiCo-SAD-NC showed the characteristics D and G band, consistent with their corresponding XRD results (Supplementary Fig. 9)[21]. Aberration-corrected HAADF-STEM image in Fig. 3b clearly demonstrated the existence of isolated Ni-Co dimer sites (marked by the yellow square) with coordination between Ni and Co at atomic level along with some isolated Ni or Co atoms (marked by the orange circle). The homogeneously distributed bright dual dots marked by the yellow squares confirmed the existence of Ni-Co dual sites, verified using the intensity profile and corresponding electron energy loss (EEL) spectra (Fig. 3c, d). The bright Ni-Co dual dots were clearly identified in the intensity profiles and corresponding EEL spectrum, suggesting the possible formation of metal–metal bonds with an average dimer distance of 0.241 ± 0.024 nm, obtained from the statistical analysis over multiple dimer sites (Fig. 3e and Supplementary Fig. 10). The ratio of dimer structure was around 78%, indicating a significant amount of this type of structure in the prepared NiCo-SAD-NC material (Supplementary Fig. 11a). Meanwhile, HAADF-STEM and EDS elemental mapping revealed that N, Ni, and Co atoms were homogeneously dispersed in the NiCo-SAD-NC, rather than any possible aggregations in the form of NPs (Fig. 3f and Supplementary Fig. 11b).

**Spectroscopic characterizations.** We further employed XPS, XANES, and EXAFS measurements to investigate the electronic state and local coordination chemistry of Ni/Co atoms in the catalysts. The C 1$s$ high-resolution XPS spectra of NiCo-SAD-NC and NiCo-NP-NC were similar to that of NC, suggesting the absence of chemical bonding between metal atoms with C (Supplementary Fig. 12a–c). Contrarily, compared to NC and NiCo-NP-NC, the N 1$s$ XPS spectra of NiCo-SAD-NC were dominated by the pyridinic-N along with porphyrin-like moieties at 399.2 eV, corresponding to the metal-nitrogen (Ni/Co-N) coordination (Supplementary Figs. 12d and 13a)[22]. Both Ni and Co 2$p$ XPS spectrum of NiCo-SAD-NC, NiCo-NP-NC, Ni-SA-NC, and Co-SA-NC exhibited the characteristics 2$p_{3/2}$ and 2$p_{1/2}$ peaks (Supplementary Fig. 13b, c). Compared to NiCo-NP-NC, the binding energies for Ni-SA-NC and Co-SA-NC were positively shifted after introducing N to trap the single-atom sites,

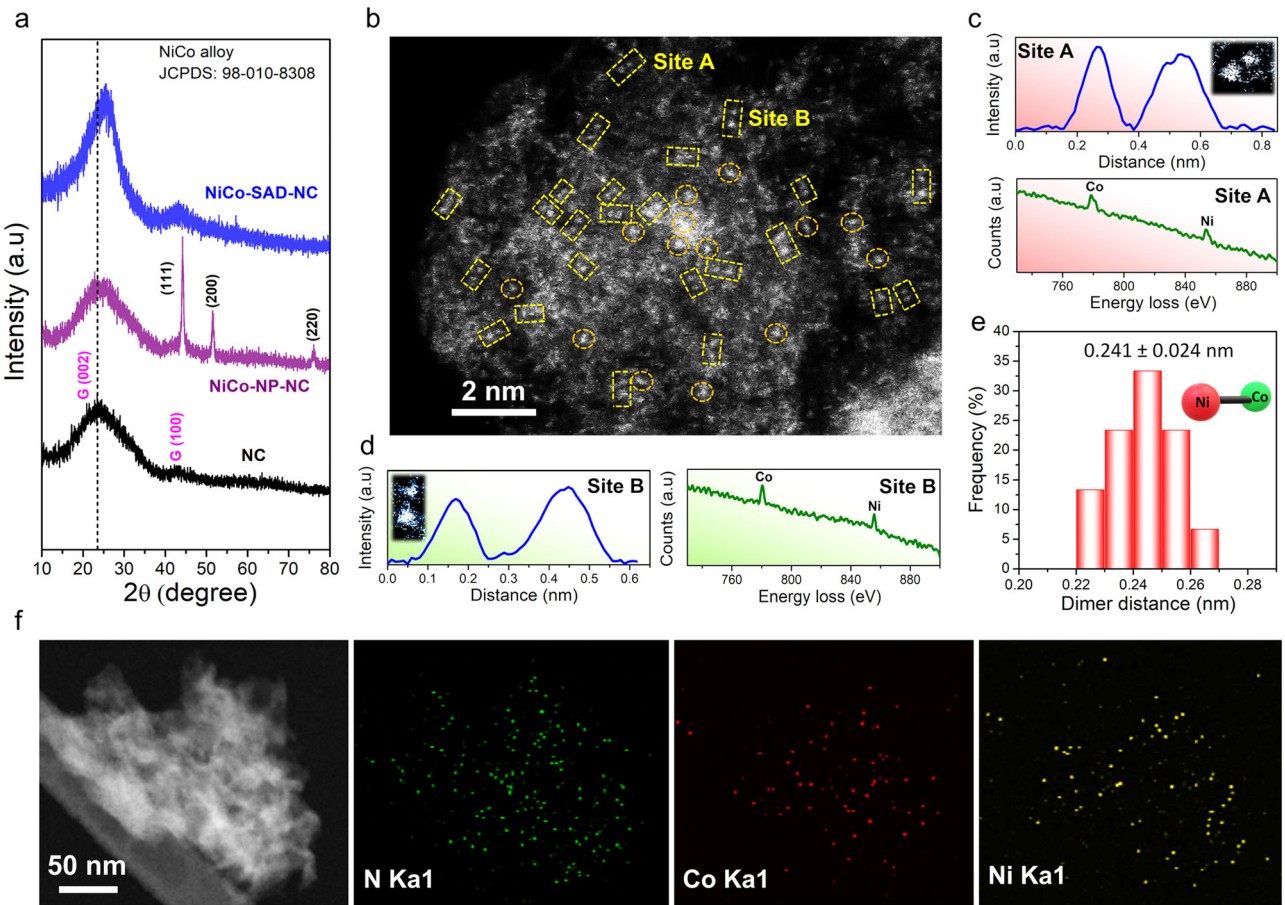

**Fig. 3 Structural analysis and electron microscopy of NiCo-SAD-NC. a** XRD spectra of NC, NiCo-NP-NC, and NiCo-SAD-NC. **b** Aberration-corrected HAADF-STEM image of the NiCo-SAD-NC. The yellow squares in **b** shows the dimer sites and some of single Ni/Co atom sites are highlighted by orange circles. **c, d** The intensity profile and corresponding EEL spectra obtained at site A (**c**) and site B (**d**) for NiCo-SAD-NC indicate Ni and Co are coordinated at the atomic scale. **e** Statistical Ni-Co distance in the observed dimers. **f** HAADF-STEM image and corresponding EDS maps of NiCo-SAD-NC showing the uniform dispersion of N (green), Co (red), and Ni (yellow).

revealing Ni-N and Co-N bond formation[28]. However, after forming the NiCo dimer sites, the Ni $2p_{3/2}$ XPS spectra of NiCo-SAD-NC showed a positive shift with Ni oxidation state of +1.73 eV compared to that of Ni in Ni-SA-NC (+1.57), whereas the Co $2p_{3/2}$ XPS spectra of NiCo-SAD-NC exhibited a negative shift with Co oxidation state of +1.39 compared to Co in Co-SA-NC (+1.67), suggesting that the electron transfer occurred from Ni to Co site at the atomic interface of NiCo-SAD-NC, probably due to single Ni-Co bond formation at the atomic level (Supplementary Fig. 14).

The Ni and Co K-edge XANES spectra of NiCo-SAD-NC, along with Ni-SA-NC, Co-SA-NC, NiCo-NP-NC, and Ni/Co foil as references exhibited a similar tendency to their corresponding XPS spectra (Fig. 4a, b). Compared to Ni foil and NiCo-NP-NC, a pre-edge peak around 8333.8 eV can be observed in the Ni K-edge XANES spectra of NiCo-SAD-NC, Ni-SA-NC, and standard nickel phthalocyanine (NiPC), due to a $1s$ to $4p_z$ shakedown transition of square-planar coordination with $D_{4h}$ local symmetry (Fig. 4a and Supplementary Fig. 15a)[27]. The weak pre-edge peak intensity of NiPC suggested its high $D_{4h}$ centro-symmetry. Compared to NiPC, the increased pre-edge peak intensity in NiCo-SAD-NC was ascribed to the distorted $D_{4h}$ local symmetry, further implying that the single Ni atom was coordinated with four nearest guest atoms (N or metal) with distorted $D_{4h}$ symmetry (Supplementary Fig. 15a)[28]. In contrast to Ni-SA-NC, the near edge and white line features in the Ni K-edge XANES spectra of NiCo-SAD-NC showed a positive shift, suggesting

a higher oxidation state of Ni in NiCo-SAD-NC compared to Ni-SA-NC, matched well with Ni $2p$ XPS results (insets of Fig. 4a). The precise oxidation state of Ni in NiCo-SAD-NC and Ni-SA-NC were +1.68 and +1.59, respectively, revealed from the further fitting of the Ni K-edge XANES energy at half-edge jump, in close agreement with the Ni oxidation states estimated from the XPS results (Supplementary Fig. 16a, b). Similar pre-edge peak around 7709.5 eV also appeared for NiCo-SAD-NC, Co-SA-NC, and standard cobalt phthalocyanine in the Co K-edge profile, suggesting the presence of X-ray absorbing Co centers with four coordination (N or metal) (Fig. 4b and Supplementary Fig. 15b). Here again, in agreement with the Co $2p$ XPS results, the near edge and white line feature in the Co K-edge XANES profile of NiCo-SAD-NC showed a negative shift compared to Co-SA-NC, indicating a lower oxidation state of Co in NiCo-SAD-NC (+1.35) compared to Co-SA-NC (+1.72) (insets of Fig. 4b and Supplementary Fig. 16c, d). Figure 4c, d displayed the Ni and Co K-edge Fourier-transformed (FT) $k^3$-weighted EXAFS spectra of NiCo-SAD-NC, with other control samples. In the Ni K-edge FT-EXAFS spectra, the predominant peaks in the R space at around 1.46 and 1.53 Å for Ni-SA-NC and NiCo-SAD-NC, respectively, can be assigned to Ni-N bonds[20]. Interestingly, the average Ni-N bond length was significantly shifted for NiCo-SAD-NC compared to Ni-SA-NC, indicating a distorted $D_{4h}$ local symmetry of Ni atom site with the simultaneous emergence of Ni-metal peak at 2.18 Å that cannot be found in Ni-SA-NC, corroborating the in situ formation of additional Ni-Co coordination

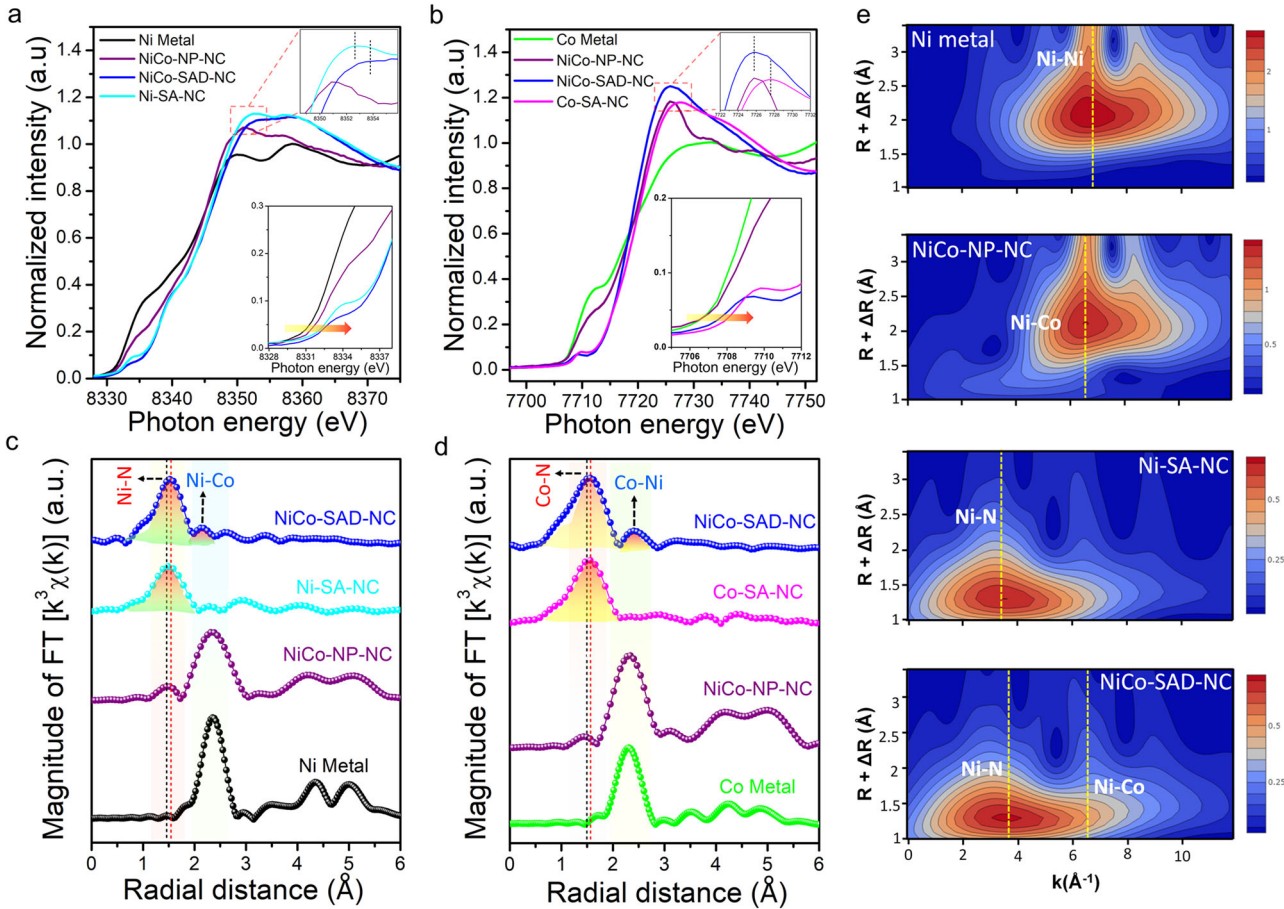

**Fig. 4 Structural characterization by X-ray absorption spectroscopy. a, b** Experimental Ni K-edge (**a**) and Co K-edge (**b**) XANES spectra of NiCo-SAD-NC with reference samples. The inset zooms in over the near edge (bottom) and white line (top) feature. **c, d** Experimental Ni K-edge (**c**) and Co K-edge (**d**) FT-EXAFS spectra of NiCo-SAD-NC along with reference samples. **e** WT-EXAFS of NiCo-SAD-NC along with reference samples at Ni K-edge.

along with Ni-N bonds, in agreement with the recently reported Zn-Co and Co-Fe dual sites (Fig. 4c)[21,22]. Supplementary Fig. 17 also confirmed the absence of dual Ni-Ni dots in Ni-SA-NC. Analogously, in the Co K-edge FT-EXAFS spectra, the elongation of Co-N bond from 1.48 Å (Co-SA-NC) to 1.56 Å (NiCo-SAD-NC) suggested the distorted $D_{4h}$ local symmetry of Co atom center with the formation of additional Co-Ni bond at 2.40 Å, which was missing in Co-SA-NC, suggesting the formation of dimer structure (Fig. 4d and Supplementary Fig. 18). The FT-EXAFS results were corroborated by the wavelet transform WT-EXAFS at the Ni K-edge (Fig. 4e). Furthermore, the structural information of NiCo-SAD-NC along with the Ni/Co-N and Ni-Co coordination numbers (CNs) was also affirmed by the EXAFS fitting results in R space oscillations (Supplementary Fig. 19 and Supplementary Table 5). The EXAFS fitting results clearly revealed that the CNs of Ni-N and Co-N were 3.3 and 3.1, respectively, suggesting that the NiCo-SAD-NC was dominated by the Ni-N$_3$ and Co-N$_3$ environment. Meanwhile, the Ni-Co/Co-Ni paths were fitted at a position of 2.55 Å with CNs of 0.7/0.6, directly indicating the existence of a significant amount of Ni-Co bonding dimer in the form of NiCo-N$_6$ structure, consistent with the HAADF-STEM results. The above results strongly demonstrate the successful formation of NiCo dimer sites with a unique coordination environment and electronic properties expected to synergistically facilitate the electrochemical activity.

**Electrocatalytic performance towards HER.** Motivated from the first-principle predictions and unique construction of Ni-Co

dimer structure, electrochemical HER performance was first accessed in 1 M KOH in a typical three-electrode setup. The Ag/AgCl reference electrode was calibrated in H$_2$ saturated electrolyte and all the potentials were converted to the reversible hydrogen electrode (RHE) (Supplementary Fig. 20)[7]. For comparison, Pt-SA was synthesized and extensively characterized using state-of-the-art techniques (Supplementary Fig. 21). The *iR*-compensated linear sweep voltammetry (LSV) polarization curves in Fig. 5a revealed an activity trend of NC < NiCo-NP-NC < Pt-SA < 20% Pt-C < NiCo-SAD-NC. The NiCo-SAD-NC delivered a remarkable alkaline HER activity, requiring only 61 and 189 mV overpotential ($\eta$) to reach −10 and −100 mA/cm$^2$, much superior to Ni-SA-NC and Co-SA-NC, corroborating the beneficial role of metal–metal synergistic effect (Fig. 5a, b and Supplementary Fig. 22a). Surprisingly, the HER performance of the NiCo-SAD-NC made solely of TMs was comparable and even superior to 20% Pt-C ($\eta_{10}$: 51.5 mV and $\eta_{100}$: 227 mV) and Pt-SA ($\eta_{10}$: 64 mV and $\eta_{100}$: 241 mV), respectively, which is indeed a rare occurrence. The superior activity of the NiCo-SAD-NC compared to NiCo-NP-NC with similar metal loading confirmed that it was indeed a pertinent step to move beyond the bimetallic-alloy NPs to SAD atomic interface, consistent with the DFT prediction (Fig. 5a, b). The Ni to Co ratio of 1 : 1 was the optimized metal ratio with the best HER performance (Supplementary Fig. 22b). We further analyzed the mass HER activity based on normalizing the HER catalytic current by their respective metal mass loading. The NiCo-SAD-NC exhibited much superior and comparable mass activity to that of 20% Pt-C and Pt-SA,

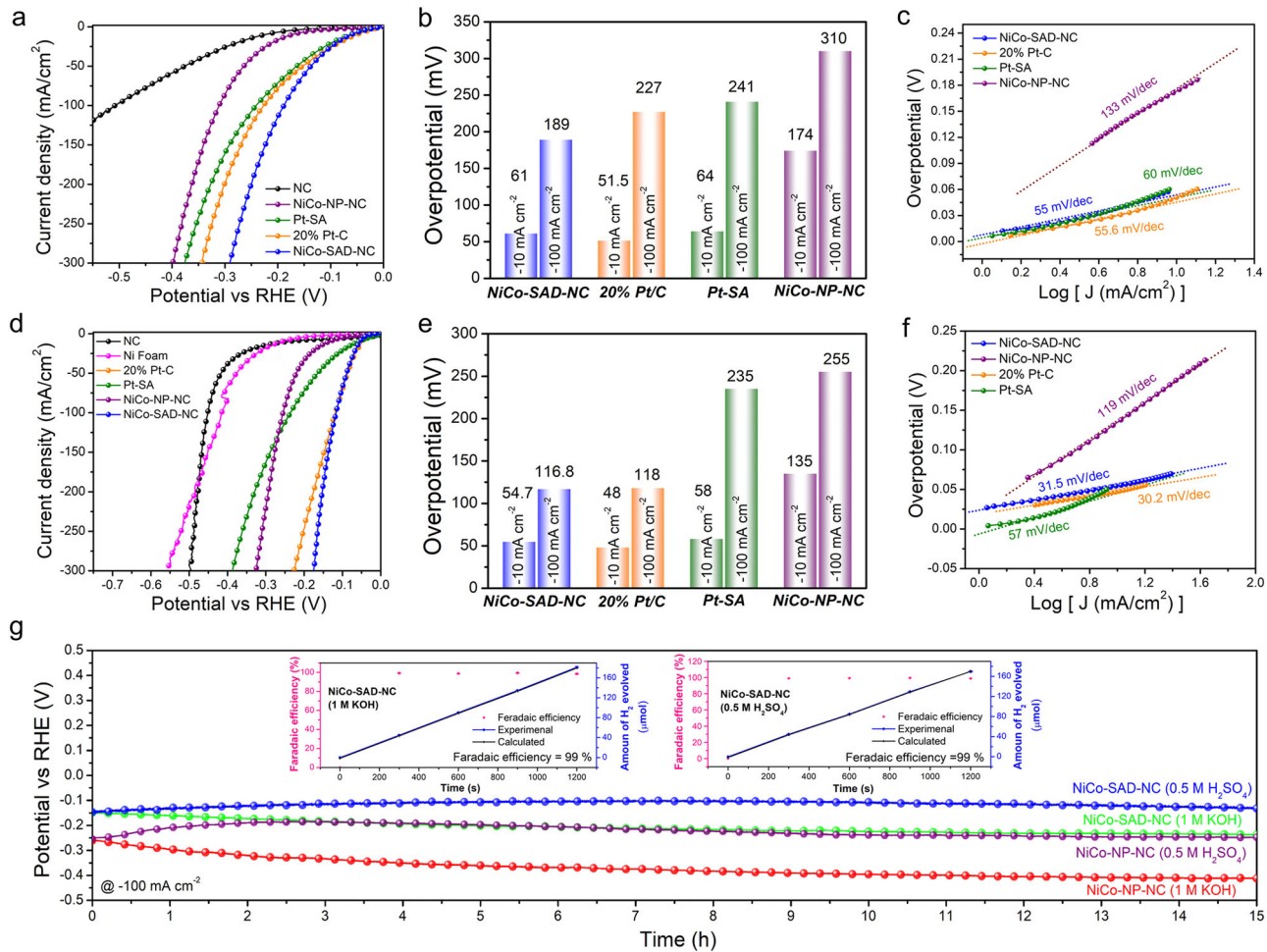

**Fig. 5 Electrochemical hydrogen generation performance in alkaline and acidic media. a–c** HER LSV polarization curves (iR-corrected) (**a**), the overpotentials required to reach −10 and −100 mA cm$^{-2}$ (**b**), and corresponding Tafel plots (**c**), in 1 M KOH. **d–f**, HER LSV polarization curves (iR-corrected) (**d**), the overpotentials required to reach −10 and −100 mA cm$^{-2}$ (**e**), and corresponding Tafel plots (**f**), in 0.5 M H$_2$SO$_4$. **g** Chronopotentiometric stability test for NiCo-SAD-NC and NiCo-NP-NC in 0.5 M H$_2$SO$_4$ and 1 M KOH at a current density of −100 mA cm$^{-2}$. The inset shows the faradaic efficiency of NiCo-SAD-NC for HER at −100 mA cm$^{-2}$ in 0.5 M H$_2$SO$_4$ (right) and 1 M KOH (left).

respectively, indicating that the SAD structures can effectively improve the performance compared to that of commercial Pt, only SACs and NPs (Supplementary Fig. 23a). The Tafel slope of NiCo-SAD-NC (55 mV/dec) was comparable to that of 20% Pt-C (55.6 mV/dec), suggesting that the HER followed the Volmer–Heyrovsky mechanism with electrochemical desorption as the rate-determining step (RDS)[7], whereas Pt-SA (60 mV/dec) and NiCo-NP-NC (133 mV/dec) exhibited higher Tafel slope, indicating slow reaction kinetics (Fig. 5c and Supplementary Discussion 2).

The NiCo-SAD-NC also exhibited promising HER performance under acidic media (0.5 M H$_2$SO$_4$). As depicted in Fig. 5d, e, the NiCo-SAD-NC required a low overpotential of only 54.7 and 116.8 mV to reach −10 and −100 mA/cm$^2$, comparable to those of 20% Pt-C ($\eta_{10}$: 48 mV and $\eta_{100}$: 118 mV) and surpassed all other control samples (Supplementary Fig. 22c). The mass activity of the NiCo-SAD-NC was also superior to that of Pt-SA and 20% Pt-C (Supplementary Fig. 23b). The Tafel slope of NiCo-SAD-NC (31.5 mV/dec) was comparable to that of 20% Pt-C (30.2 mV/dec), suggesting that both NiCo-SAD-NC and Pt-C exhibited similar RDS, where two hydrogen intermediates desorb and form molecular H$_2$ (Tafel step)[23] (Fig. 5f). The superior activity of the NiCo-SAD-NC under acidic and alkaline media

was further corroborated by the smallest charge transfer resistance ($R_{CT}$) with a higher electrochemically active surface area (ECSA) of 13.05 cm$^2$ compared to NiCo-NP-NC (6.8 cm$^2$), indicating rapid electron transfer kinetics with high abundance of active sites, boosting the HER performance (Supplementary Figs. 24 and 25)[29]. The excellent intrinsic activity of the NiCo-SAD-NC over NiCo-NP-NC was further corroborated by the ECSA-normalized HER curve (Supplementary Fig. 26). In addition, we also synthesized CoFe/CoMn-SAD-NC following a similar procedure and compared the HER performance with NiCo-SAD-NC in acidic/alkaline media (Supplementary Figs. 27 and 28). The HER activity of the NiCo-SAD-NC was superior to that of both CoFe/CoMn-SAD-NC in alkaline media and comparable to that of CoFe-SAD-NC in acidic media, consistent with the DFT prediction. In addition, the superior intrinsic HER performance of NiCo-SAD-NC was maintained under both alkaline and acidic media using Hg/HgO and Hg/Hg$_2$SO$_4$ as reference electrodes, respectively (Supplementary Fig. 29). Furthermore, to claim the pH universality, we tested the HER performance of NiCo-SAD-NC in phosphate buffer solution (PBS) (pH 7.4) and compared the HER activity with Pt/C and NiCo-NP-NC (Supplementary Fig. 30). Even in the PBS electrolyte, the NiCo-SAD-NC outperformed the NiCo-NP-NC

and was comparable with the benchmarking Pt/C. Since the Ni-foam substrate itself has some intrinsic HER activity, we further explored the HER performance of NiCo-SAD-NC along with Pt-C on a more inert carbon fiber paper (CFP) as the catalyst support[30,31]. Supplementary Fig. 31 revealed that the NiCo-SAD-NC also exhibited striking HER performance on CFP under both acidic/alkaline media, still comparable to 20% Pt-C and superior to that of NiCo-NP-NC, affirming the high intrinsic activity of the SAD atomic interface. However, the required overpotential was slightly larger than that on Ni-foam, probably due to less porous and thinner structure characteristics of CFPs with respect to Ni-foam hindering mass transport and increases $R_{CT}$[32].

The long-term durability test under both acidic/alkaline media was performed by chronopotentiometry analysis[33]. The best-performing NiCo-SAD-NC showed exceptional stability under both acidic and alkaline media for at least 15 h at $-100$ mA/cm$^2$ without any noticeable degradation, whereas the NiCo-NP-NC also showed reasonable durability (Fig. 5g). The NiCo-SAD-NC also exhibited long-term stability for at least 50 h at $-50$ mA/cm$^2$ in both acid/alkaline media (Supplementary Fig. 32). The overlapping LSV curve taken before and after the durability test further asserted the retention of active sites accompanied by high catalytic activity (Supplementary Fig. 33). Post-stability aberration-corrected HAADF-STEM images with corresponding intensity profiles and EEL spectrum along with nearly matching XANES and EXAFS spectra clearly revealed that the Ni-Co dimer sites in the NiCo-SAD-NC were well-preserved without any sign of metal agglomeration to form clusters/NPs (Supplementary Fig. 34). In addition, the crystal structure, morphology, and composition of the NiCo-SAD-NC were also maintained after the HER durability test (Supplementary Figs. 35 and 36). After stability test, the XPS spectra were matched with the pristine NiCo-SAD-NC along with well intact metal-N bonds. In addition, the NiCo-SAD-NC also demonstrated around 99% Faradaic efficiency for HER under both alkaline/acidic media at $-100$ mA cm$^{-2}$, indicating that almost all charge were utilized for hydrogen generation without any parasitic-side reactions (Fig. 5g, inset). The HER activity of NiCo-SAD-NC was highly reproducible as well as consistent (Supplementary Fig. 37). The pH-universal HER activity of the NiCo-SAD-NC was comparable and even better than most of the recently reported single-atom-based electrocatalysts (Supplementary Table 6).

## Discussion

In summary, we developed a facile strategy to obtain earth-abundant SAD sites via in situ trapping of the targeted metal ions (Ni, Co) followed by pyrolysis with precisely controlling the N-moieties for pH-universal HER. The detailed structural analysis of the obtained NiCo-SAD sites was carried out by XAS, AC-STEM, and XPS, which revealed that the NiCo-SAD-NC contains Ni-Co bond at atomic level stabilized by the N coordination. More notably, the synergistic interaction at the Ni-Co atomic interface in the SAD structure can significantly upshift the $d$-band center closer to the Fermi level and accelerated water dissociation, boosting pH-universal HER, as predicted by DFT calculations. Consistently, the obtained NiCo-SAD-NC delivered exceptional pH-universal catalytic kinetics towards HER, outperformed the NPs counterpart and comparable/superior to commercial Pt-C/Pt-SA, additionally surpassed most of the recently reported TM-based single-atom electrocatalysts. Our findings provide a rational design strategy for fabricating an earth-abundant metal-based SAD catalyst with atomic precision for both fundamental and practical research as well as for the deeper understanding of the bimetal synergistic effect for future energy-related applications.

## Methods

**Chemicals**. Tris-buffered Saline (Sigma-Aldrich), Dopamine hydrochloride ((HO)$_2$C$_6$H$_3$CH$_2$CH$_2$NH$_2$·HCl; Sigma-Aldrich), Nickel(II) nitrate hexahydrate (Ni(NO$_3$)$_2$·6H$_2$O; Sigma-Aldrich, ≥99%), Cobalt(II) nitrate hexahydrate

(Co(NO$_3$)$_2$·6H$_2$O; Sigma-Aldrich, ≥99%), Iron(III) nitrate nonahydrate (Fe(NO$_3$)$_3$.9H$_2$O; ≥99%), Manganese(II) nitrate tetrahydrate (Mn(NO$_3$)$_2$·4H$_2$O; Sigma-Aldrich, ≥99%), Dicyandiamide (NH$_2$C(=NH)NHCN; Sigma-Aldrich, ≥99%), Chloroplatinic acid hexahydrate (H$_2$PtCl$_6$·6H$_2$O; Sigma-Aldrich, ≥99%), potassium hydroxide pellets (KOH; Sigma-Aldrich, ≥85%), Sulfuric acid (H$_2$SO$_4$; Sigma-Aldrich, ≥99.99%), ethanol (C$_2$H$_5$OH; Sigma-Aldrich, ≥99.9%), Toray CFP/Ni foam (Alfa Aesar), and the nafion perfluorinated resin solution (5 wt.%, Sigma-Aldrich) were used without further purification.

**Synthesis of Ni$^{2+}$-Co$^{2+}$@Polydopamine (precursor)**. In a typical procedure, Tris-buffer (1.21 g) was dissolved in 135 mL of distilled water (DI water) followed by drop-wise addition of aqueous solution (5 mL) containing metal salts (2 mg/mL, Ni(NO$_3$)$_2$·6H$_2$O: Co(NO$_3$)$_2$·6H$_2$O = 1 : 1). Then dopamine hydrochloride (70 mg) was quickly added in the above suspension and the polymerization was kept under magnetic stirring for 24 h. The resultant precipitate was collected via filtration and washed two times with DI water and ethanol, respectively, and dried at 60 °C overnight. For control samples, Ni$^{2+}$-Co$^{2+}$@Polydopamine precursor with different ratio of Ni(NO$_3$)$_2$·6H$_2$O: Co(NO$_3$)$_2$·6H$_2$O (1 : 2 and 2 : 1) as well as only Ni$^{2+}$@Polydopamine (2 mg/mL, Ni(NO$_3$)$_2$·6H$_2$O solution), Co$^{2+}$@Polydopamine (2 mg/mL, Co(NO$_3$)$_2$·6H$_2$O solution), Pt$^{4+}$@Polydopamine (2 mg/mL, H$_2$PtCl$_6$·6H$_2$O solution), Co$^{2+}$-Fe$^{3+}$@Polydopamine (2 mg/mL, Co(NO$_3$)$_2$·6H$_2$O: Fe(NO$_3$)$_3$·9H$_2$O = 1 : 1), Co$^{2+}$-Mn$^{2+}$@Polydopamine (2 mg/mL, Co(NO$_3$)$_2$·6H$_2$O: Mn(NO$_3$)$_2$·4H$_2$O = 1 : 1), and polydopamine were also synthesized.

**Synthesis of NiCo-NP-NC**. For the synthesis of NiCo-NP-NC, a certain amount of Ni$^{2+}$-Co$^{2+}$@Polydopamine powder was placed in a vacuum furnace and heated at 800 °C for 2 h with a heating rate of 5 °C min$^{-1}$.

**Synthesis of NiCo-SAD-NC**. In a typical procedure, a certain amount of Ni$^{2+}$-Co$^{2+}$@Polydopamine (precursor) was mixed with dicyandiamide (organic molecule: OM) in a ratio of 1 : 7 by grinding in a mortar. The mixture was annealed in a vacuum furnace at 800 °C for 2 h with a heating rate of 5 °C min$^{-1}$ to yield NiCo-SAD-NC. Other control samples were also synthesized via the same procedure by varying the ratio of Ni$^{2+}$-Co$^{2+}$@Polydopamine (precursor) to OM and denoted as NiCo-NC (1 : 1), NiCo-NC (1 : 3), and NiCo-NC (1 : 5). For comparison, NiCo-SAD-NC (1 : 2 and 2 : 1), Ni-SA-NC, Co-SA-NC, CoFe-SAD-NC, and CoMn-SAD-NC were also synthesized following the similar procedure by only changing the starting precursor (Ni$^{2+}$-Co$^{2+}$@Polydopamine with different ratio of Ni(NO$_3$)$_2$·6H$_2$O: Co(NO$_3$)$_2$·6H$_2$O (1 : 2 and 2 : 1), Ni$^{2+}$@Polydopamine, Co$^{2+}$@Polydopamine, Co$^{2+}$-Fe$^{3+}$@Polydopamine, and Co$^{2+}$-Mn$^{2+}$@Polydopamine).

**Synthesis of NC**. For the synthesis of NC, a certain amount of polydopamine was mixed with dicyandiamide in a ratio of 1 : 7 by grinding in a mortar, followed by annealing in a vacuum furnace at 800 °C for 2 h with a heating rate of 5 °C min$^{-1}$.

**Synthesis of Pt-SA**. For the synthesis of Pt-SA, a certain amount of Pt$^{4+}$@Polydopamine precursor was mixed with dicyandiamide in a ratio of 1 : 20 by grinding in a mortar, followed by annealing in a vacuum furnace at 800 °C for 2 h with a heating rate of 5 °C min$^{-1}$.

**Material characterization**. The XRD measurements were carried out using a Rigaku Ultima IV powder X-ray diffractometer with Cu Kα radiation at $\lambda = 0.15405$ nm. FESEM images and EDS spectra were obtained using a JEOL 7500F FESEM. The Raman spectra were obtained using a Renishaw RM 1000-Invia micro-Raman system with excitation energy of 2.41 eV (514 nm). The XPS measurements were carried out on a Thermo VG Microtech ESCA 2000, with a monochromatic Al-Ka X-ray source at 100 W. The binding energy scale was calibrated by referencing C 1$s$ to 284.5 eV. The XPS data were background corrected by the Shirley method and the peaks were fitted using Fityk software, with Voigt peaks containing 80% Gaussian and 20% Lorentzian components to get the valence states. TEM images were recorded using a JEOL JEM-2100F with an accelerating voltage of 200 kV. The aberration-corrected HAADF-STEM was performed using a Thermo Fisher Themis Z TEM equipped with a double Cs corrector, an electron-beam monochromator, and Gatan Image Filter (GIF, model Quantum 965) at Seoul National University. The acceleration voltage was set to 200 kV. EEL spectra were all acquired with a 5 mm EELS aperture corresponding to a collection angle of 45 mrad, a probe with a convergence angle of 49 mrad, and a beam current of ~75 pA. The EELS spectrometer was set to 0.25 eV per channel dispersion. The ICP-AES measurements were done using OPTIMA 4300 DV. XANES and EXAFS data were collected on BL10C beamline at the Pohang light source (PLS-II) with top-up mode operation under a ring current of 250 mA at 3.0 GeV. The monochromatic X-ray beam can be obtained using liquid-nitrogen cooled Si (111) double crystal monochromator (Bruker ASC) using intense X-ray photons of multipole wiggler source. The X-ray absorption spectroscopic data were recorded for the uniformly dispersed powder samples with a proper thickness on the polyimide film, in fluorescence mode with N$_2$ gas-filled ionization chamber (IC-SPEC, FMB Oxford) for incident X-ray and passivated implanted planar silicon detector (PIPS, Canberra, Co.). Higher-order harmonic contaminations were

eliminated by detuning to reduce the incident X-ray intensity by ~30%. Energy calibration has been simultaneously carried out for each measurement with reference metal foils placed in front of the third ion chamber. The data reductions of the experimental spectra to normalized XANES and FT radial distribution function were performed through the standard XAFS procedure using IFEFFIT package. Also, Morlet wavelet-transformed EXAFS spectra have been obtained with proper values of $\eta$ and $\sigma$ in the equation as follows;

$$\varphi(t) = \frac{1}{\sqrt{2\pi\sigma}}(e^{i\eta t} - e^{-\eta^2\sigma^2/2})e^{-t^2/2\sigma^2} \qquad (1)$$

in which the $\eta$ is the frequency of the oscillation functions and the $\sigma$ is the half-width.

**Electrochemical measurements.** Electrochemical measurements were conducted using a VMP3 electrochemical workstation (Bio-logic Science Instruments, France) in a typical three-electrode configuration in 1 M KOH and 0.5 M $H_2SO_4$ as the electrolyte. Ag/AgCl (3 M KCl) and graphite rod were used as the reference and counter electrode, respectively. The catalyst ink-coated Ni foam or CFP was used as the working electrode. The reference electrode was calibrated in $H_2$-saturated 1 M KOH and all the potentials are converted to a RHE using the Nernst equation.

$$E(\text{RHE}) = E(\text{Ag/AgCl}) + E^0(\text{Ag/AgCl}) + 0.059 \times \text{pH} \qquad (2)$$

Then, 5 mg of catalyst powder was dispersed in 500 μL of ethanol containing 20 μL 5% Nafion and sonicated for 60 min to get a homogeneous ink. Afterward, a certain quantity of the ink was drop-cast onto Ni foam/CFP (loading: 0.8 mg cm$^{-2}$) and left to dry under ambient atmosphere. Before measurements, the catalysts were saturated via cyclic voltammetry (CV) scans at a scan rate of 100 mV s$^{-1}$. LSV was taken at a slow scan rate of 2 mV s$^{-1}$ to minimize the capacitive contribution[7]. Nyquist plot was obtained using electrochemical impedance spectroscopy measurements in the faradaic region to estimate the charge transfer resistance ($R_{CT}$). $C_{dl}$ was obtained by collecting CVs at various scan rates of 10, 20, 30, 40, and 50 mV s$^{-1}$ in the non-faradaic. ECSA was obtained from the $C_{dl}$ value using a specific capacitance of 0.04 mF/cm$^2$. The durability test was performed using chronopotentiometry. Faradaic efficiency was measured by using the eudiometric method in an air-tight vessel. All the potentials were 85% iR-corrected with respect to the ohmic resistance of the solution unless specified and calibrated to the RHE using the following equation[29].

$$E_{(\text{RHE})} = E_{(\text{Ag/AgCl})} + E^0_{(\text{Ag/AgCl})} + 0.059 \times \text{pH} - 85\% \, iR_s \qquad (3)$$

**Computational details.** All the DFT calculations were carried out in the VASP computational package[34]. The plane-wave was constructed with the projected augmented wave pseudopotentials[35] and the Perdew-Burke-Ernzerhof generalized gradient exchange approximation correlational functional are used for the treatment of the core electrons[36]. All geometric structures were fully optimized until forces and total energy are converged to 10$^{-5}$ eV/cell and $-0.01$ eV/Å, respectively. The vacuum space in the z-direction was set as 15 Å to eliminate interaction between two periodic images, and the cut-off energy was chosen at 450 eV. The Grimme-D3 level was used to describe the long-range van der Waals interactions[37,38]. The Brillouin zone of k-points is sampled by a $3 \times 3 \times 1$ Monkhorst–Pack grid. A $4 \times 4 \times 1$ supercell model of primitive graphene containing the TM-SAD-$N_6$C was adopted for the surface calculations. The minimum energy path of water dissociation on TM-SAD-$N_6$C surfaces was obtained by the nudged elastic band method with 5 intermediate images used to search for the transition states[39,40]. Vibrational free energy was calculated by zero-point energy and entropy contribution at room temperature (298 K).

## Data availability
All the data supporting this study are available in the paper and Supplementary Information. Additional data related to this study are available from the corresponding authors on reasonable request.

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

## Acknowledgements

This work was supported by the Institute for Basic Science (IBS-R011-D1). G.S.P. was supported by the National Research Foundation of Korea (NRF) grants funded by the Ministry of Science, ICT & Future Planning (MSIP) (NRF-2019R1A2C2090249). We also thank the High-performance Computing Collaboratory (HPC2), Mississippi State University, and the Center for Computational Materials Science, Institute for Materials Research, Tohoku University, Japan, for the use of MASAMUNE-IMR, Cray XC50-LC supercomputer facility.

## Author contributions

A.K. designed and carried out the experiments, as well as analyzed the data with support from H.L. V.Q.B., H.T.D.B., S.G.K. and Y.K. performed the theoretical calculations. J.L. helped to conduct the Raman analysis. A.R.J., L.W., J.Y., S.A., X.S., X.L. and Y.L. helped in the data discussion. Y.H. helped in collecting the FESEM images. G.S.P. helped in collecting the STEM data. M.G.K. performed the XAS measurements. A.K., V.Q.B. and H.L. co-wrote the paper. All authors have given approval to the final version.

## Competing interests

The authors declare no competing interests.
