## [Peer Review File · Nature Communications]

Moving Beyond Bimetallic-Alloy to Single-Atom Dimer Atomic-Interface for All-pH Hydrogen EvolutionREVIEWER COMMENTS

Reviewer #1 (Remarks to the Author):

The paper by Kumar et al. aims at demonstrating that dimeric heterometallic sites can be formed in graphitic-like materials, which are very efficient in alkaline and acidic HER. The topic is very interesting and the development of SACs and related catalysts very promising for HER and other energy-related reactions using earth abundant elements. The authors' approach for the catalyst preparation is interesting and the performances of the materials are very high with respect to the current standard. Overall, the paper is well written and the experiments are carefully performed. There are, however, important discrepancies in the identification of the active sites. We are not convinced by the heterometallic dimeric nature of the active sites described by the authors. In particular, the X-ray absorption spectroscopy falls short in describing the structure of the catalysts. Without any convincing evidence for the structure of the materials, the paper cannot be published in Nature Communications. Here are my comments about the paper and suggestions on how to improve it for a potential future publication:

Page 3: What does “atomic interface” mean?; What does “complex HER” mean?

Page 8: It is said that HRTEM and Raman shows an increased amount of carbon defects in the NiCo-SAD-NC material. Why is that? This conclusion is not obvious at all, looking either at the HRTEM or Raman figures.

- Electronic microscopy: One of the very strong evidences for the presence of dual Ni-Co sites is the observation by HAADF-STEM and EELS of Ni and Co intensities next to each other. This observation is, however, quite subjective. Looking at figure 2b, I could draw circle around bright spots that would certainly not correspond to the ones chosen by the authors. Also, the intensity of the Ni or Co peaks are not so evident on the EELS spectra. It would be more convincing is a series of spectra were shown in supplementary information, in order to see what the intensities look like for different spots. The same is true for the intensity profiles that show the presence of Co or Ni and their interatomic distances. It would be very beneficial to see a much longer line, in order to estimate the noise level and the sensitivity of the intensity profile measurements.

Both the EELS and intensity profiles are not so obvious to see, so it is important to have a statistical analysis to make sure that this is not just one event here and there.

The EDS elemental maps on figure 2f should be superimposed (at least the nickel and cobalt distribution) to make sure that they are colocalized.

- XPS: The signal-to-noise ratio of the Ni and Co XPS data (figure 10b, c) is quite poor and a significant smoothing was applied to obtain “nice” data. How reasonable is that? For example, the comparison of Co-SA-NC and NiCo-SAD-NC is not convincing when looking at the raw data. There is no convincing argument to attribute the shifts observed in the Ni and Co 2p_{3/2} data. The conclusions could be interesting, but there should be more comparison with reference samples and/or literature. If more reference spectra were measured, better comparison could be made and tentative oxidation states could be attributed, which could correlate with those observed in XAS (vide infra).

- XANES/EXAFS: The fact that a pre-edge can be observed for both the Ni and Co samples is not an evidence for a square planar geometry. It is one possibility, but many other are possible since pre-edge features arise from any geometry except from a perfect octahedron (which never exists). Unless a very good resolution is obtained (with a high-resolution spectrometer) and appropriate reference samples are used, the geometry of the metal site cannot be precisely determined. DFT calculations could also help if the pre-edge features were calculated.

- The determination of the oxidation state of Ni and Co is done by interpolating the edge energy position of the samples of interest between the positions of the corresponding metallic metal and their oxides. The edge position energy is used as a marker, but there is no mention as to how it is determined. It seems to be the first inflexion point, but it does not reflect the reality. Although there is a slight shift in the pre-edge position of the NiCo-SAD-NC and Ni-SA-NC it is clearly not equal to the 2 eV reported on figure 11b. The method by which the edge position is determined should be clearly mentioned.

Considering the nickel case, if the edge position is considered at half-edge jump, then the oxidation states of the Ni-SA and NiCo-SAD samples should be higher than +II. This would actually be more reasonable than a +0.5 or +1.0 oxidation state, which would be very unusual for a nickel ion in an N-coordinating environment.

Considering the cobalt case, the CoO reference sample looks to me like a Co₃O₄ sample (spinnelle), which is a mixed +II/+III oxidation state. Considering the reference samples used and an edge position at half-edge jump, then indeed, the oxidation state of cobalt in Co-SA and NiCo-SAD would be around +0.5-1.0. But again, this oxidation state is quite exotic for an N-coordinated cobalt complex.

Whether for nickel or cobalt, better references than oxides should be used to determine the oxidation state of the nickel and cobalt ions in the M-SA and NiCo-SAD samples. Metal porphyrins, phthalocyanines or other types of molecular, N-coordinating and delocalised systems would be more appropriate.

The determination state of nickel and cobalt determined by XAS should be compared to the oxidation states obtained by XPS, which are not specifically determined.

If the proposed oxidation states of +0.5 -1.0 are maintained despite the recommendations above, there should be additional evidences for their existence, since they are usually very unstable in coordination complexes. EPR spectroscopy could prove useful in confirming these oxidation states.

- The EXAFS analysis is very poor and only based on visual observation of the spectra. This is not acceptable and proper fits should be performed to determine the actual M-N distances and coordination numbers. The small peak at $R=2$ shown as a proof of the existence of a Ni-Co interaction looks more like a wish than actual data. In the nickel case, the peak said to correspond to the Ni-Co interaction is less intense than the noise level, so it is very difficult to consider it as a real feature. Again, fits should provide more solid information for a proper structural discussion.

It is not clear whether the data shown on figure S14 are just deconvolution of the spectra or actual EXAFS fits. If so, then the fitting parameters should be given.

Finally, there is not characterization of the material after electrochemical activity, even under ex situ conditions. Without such information, it is not reasonable to attribute the particular catalytic performances of this material to the dimeric heterometallic structure postulated.

Methods: The experimental X-ray absorption data collection conditions is very limited and far below the standards of any publication reporting such data. The monochromator type, detection mode, sample conditions and data analysis should be indicated.

Reviewer #2 (Remarks to the Author):

Kumar et al. report a metal@polymer strategy to fabricate the diatomic NiCo in N-doped carbon matrix for HER in H₂SO₄ and KOH. In this work, the promising candidates were firstly screened by DFT simulations and then characterized by ex situ XRD, STEM, XPS, and XAS. The overpotential of 54.7/61 mV in acid/base was measured in NiCo-SAD@Ni foam. The synthetic approach is nicely done to precisely tune single/dual atom catalyst and the synergistic effect of the diatomic system was well established. I believe this manuscript can be considered for publication after the following concerns are resolved.

DFT :

1. In line 109, the d-band center of the diatomic NiCo center (-0.87 eV) is claimed to be nearest to the Fermi level, but NiNi has a lower value of -0.17 eV (Table S1). Is it conflicting? The authors stated that NiNi was inactive and incapable of H₂O adsorption, which is quite the opposite to the experimental observations that Ni NPs or clusters show strong adsorption of H₂O. A more detailed explanation would be appreciated. Similarly, FeFe was also ignored with a low theoretical overpotential of ~0.470 eV comparable to other TM combinations. It is suggested to clearly justify the reasons why those combinations were ruled out for further considerations.

2. The fast screening of DFT simulation includes five 3d transition metals, Mn, Fe, Co, Ni, and Cu. It is suggested to provide rationale behind why these elements were selected but not the others, like Cr, Zn, and Mo.

3. What do the Bader charge and charge density distribution reflect (line 101 to 106)? Is the discussion related to the intrinsic property or the HER activity of materials?

Synthesis and characterization:

4. To prepare NiCo dimer site rather than alloy, dicyandiamide was added into NiCo@polydopamine precursor. How does the dicyandiamide affect the final composition since metal ions are already coordinated to polydopamine? The authors should give a more detailed explanation.

5. With the same synthetic procedure, the samples should have the similar coordination. Why does only NiCo dimer form in SAD while the counterparts are SA? Is it a mixed state of isolated Ni, Co centers, and NiCo centers in NiCo-SAD? Is it a mixed state of Ni and NiNi centers in Ni-SA?

6. According to the XPS and XANES features, the authors demonstrated the electron transfer between the NiCo interface. The essential reasons behind the interplay should be discussed.

7. What do the peaks at $\sim 1.3 \text{ \AA}$ (pink color) represent in the EXAFS fitting results in Figure S14? There is a difference of 0.22 \AA between Ni-Co and Co-Ni scattering. What is the reason for this difference?

HER:

8. Ag/AgCl (3 M KCl) electrode is only applicable in neutral medium. There is a consensus for electrochemist to use the suitable reference electrode (Hg/Hg₂SO₄ and Hg/HgO in acidic and basic media, respectively), especially for the long-term test. A suitable reference electrode should be used for each pH range.

9. To claim “all-pH” or “pH-universal”, the performance test in a neutral solution such as PBS is necessary. Otherwise, it is better to change the conclusion in a weaker tone.

10. The stability tests of SAD catalyst were compared by chronopotentiometric method. It shows negligible degradation in acid while $\sim 80 \text{ mV}$ decline in base. Could the authors explain the different behaviors? Also, the LSVs before and after the reaction should be provided.

11. Ambiguous discussion and errors:

(a) In the abstract, the authors state that the water-dissociation is facilitated, thus accelerate the HER kinetics in acid/alkaline. However, water dissociation does not take place in acid HER. It is better to make the statement more precise.

(b) There are two different rows of d-band center in Table S2, which looks confusing to me. Please correct or define them.

(c) The overpotentials measured on CFP should be labeled in Figure S25. It looks like $\sim 70/320 \text{ mV}$ in acid and $\sim 90/350 \text{ mV}$ in base at $10/100 \text{ mA cm}^{-2}$.

Reviewer #3 (Remarks to the Author):

The electrocatalytic hydrogen evolution reaction through the splitting of water has drawn great interests these years. This paper reports the formation of bimetallic single-atom-dimer catalysts for HER in both acidic and alkaline media. The authors have delivered detailed characterizations and thorough analysis

about those catalysts. The whole work is remarkable. However, I think the total novelty of this work is not enough to meet the high standards of the Nature Communications before revised.

1. Since the pioneering work by Tao Zhang on single-atom, a bulk of works has been reported to construct different single-atom systems. The synthesise process of this work is not novelty. A wide-pH-range HER catalyst or an overpotential lower than 50 mV could also been easily achieved, even in a none-noble-metal system (10.1002/adma.201807780, 10.1038/s41467-018-03358-x, 10.1002/anie.202011358). The wide-pH-range HER catalyzed by single-atom could also reported many times (10.1016/j.nanoen.2019.02.062, 10.1038/s41467-020-14848-2, 10.1021/acssuschemeng.0c04322). Thus, this paper is lack of novelty to meet the high standards of Nature Communications.
2. The authors should provide more clear evidence for the bimetallic single-atom-dimer. Neither the HAADF-STEM images nor EXAFS results is convincing, both of them seems to be subjective judgments. For example, more clear HAADF-STEM images of the dimers are needed (Angew. Chem. 2020, 132, 16147).
3. EELs spectra of Figure 2C is not convincing, due to the transmission property of TEM. The signals of atoms below the surface would be reckoned in. HAADF-STEM image with matched EELs mapping should be provided.
4. The authors also evaluated the energy barrier for H₂O dissociation of some selected TM-SAD structures, and get the conclusion that NiCo-SAD is a promising candidate for HER. According to Fig. 1c,d, CoMn is so closed to NiCo, while CoFe SAD is far from NiCo. However, the LSV lines of CoFe and CoMn are nearly overlaid in Fig. S24. The authors should also provide explanation about this.
5. In the electro catalytic performance part, the authors tested the HER performance of NiCo-SAD-NC and other counterparts. It was concluded that the HER followed the Volmer-Heyrovsky mechanism. The authors should provide detailed mechanism analysis about the roles of Ni site and Co site in Ni-Co pair. What's more, the existence of NiCo pair in NiCo-SAD-NC samples after stability test should be confirmed and need more discussion.

Point-by-point response for Nature Communications manuscript
(ID: NCOMMS-21-19692)

Manuscript Type: Article

Title: Moving Beyond Bimetallic-Alloy to Single-Atom Dimer Atomic-Interface for All-pH Hydrogen Evolution.

Author(s): Ashwani Kumar, Viet Q. Bui, Jinsun Lee, Lingling Wang, Amol R. Jadhav, Xinghui Liu, Xiaodong Shao, Yang Liu, Jianmin Yu, Yosep Hwang, Huong T. D. Bui, Sara Ajmal, Min Gyu Kim, Seong-Gon Kim, **Gyeong-Su Park**, Yoshiyuki Kawazoe, and Hyoyoung Lee.

We are grateful to the editor, editorial staff, and reviewers for their critical comments and valuable suggestions. The manuscript has been modified after addressing all the suggestions as listed below:

(The explanations to the comments from reviewers are shown with **yellow highlight**).

Author(s)' Points-by-points responses to Reviewer(s)':

Reviewer: 1

Comments:

The paper by Kumar et al. aims at demonstrating that dimeric heterometallic sites can be formed in graphitic-like materials, which are very efficient in alkaline and acidic HER. The topic is very interesting and the development of SACs and related catalysts very promising for HER and other energy-related reactions using earth abundant elements. The authors' approach for the catalyst preparation is interesting and the performances of the materials are very high with respect to the current standard. Overall, the paper is well written, and the experiments are carefully performed. There are, however, important discrepancies in the identification of the active sites. We are not convinced by the heterometallic dimeric nature of the active sites described by the authors. In particular, the X-ray absorption spectroscopy falls short in describing the structure of the catalysts. Without any convincing evidence for the structure of the materials, the paper cannot be published in Nature Communications. Here are my comments about the paper and suggestions on how to improve it for a potential future publication:

Response: We are grateful for the time and effort Reviewer 1 has spent in reviewing our manuscript. The review comments are constructive for further strengthening the manuscript.

1) Page 3: What does “atomic interface” mean?; What does “complex HER” mean?

Answer 1) We sincerely thank the referee for the comment, and we are pleased to clarify this issue.

“Atomic interface”

Here “atomic interface” indicates the interface region between the Ni and Co single atoms in the atomically isolated Ni-Co SAD structure. Since this interface is formed between two atoms at the atomic level, so we termed it as “atomic interface” (Rep. Fig. 1). This type of “atomic-interface” terminology has also been previously utilized to describe the atomically dispersed Pt-O-Ti³⁺ center [Angew. Chem. Int. Ed. **59**,1295–1301(2020)].

Representative Fig. 1 | Atomic interface region between Ni and Co atoms in the NiCo-SAD structure.

“Complex HER”

Generally, HER refers to the adsorption of the proton (H^{*}) on the catalyst surface followed by the H^{*} dimerization to molecular H₂. In the acidic media (pH=0), due to the higher abundance of the H^{*} source, the HER proceeds with the adsorption of H^{*} on the catalysts surface from the electrolyte and then undergoes dimerization to H₂, facilitated by the electrons from the catalysts (Rep. Fig. 2a).

However, in the alkaline media (pH=14), the source of H^{*} is water (H₂O). To initiate alkaline HER, the first step is water dissociation, which is necessary to generate sufficient H^{*} and then the generated H^{*} undergo HER to evolve H₂ (Rep. Fig. 2b) [Science, **334**, 1256-1260 (2011); ACS Energy Lett. **6**, 354-363 (2021)].

Hence, for pH universal HER (acidic and alkaline media), an additional sluggish water dissociation step needs to be considered along with the H^{*} adsorption and dimerization to molecular H₂, which makes pH universal HER relatively complex. Therefore, we termed pH universal HER as “complex HER”.

Representative Fig. 2 | a, HER in acidic media and b, HER in alkaline media.

2) Page 8: It is said that HRTEM and Raman shows an increased amount of carbon defects in the NiCo-SAD-NC material. Why is that? This conclusion is not obvious at all, looking either at the HRTEM or Raman figures.

Answer 2) We thank the referee for carefully reviewing our manuscript and we are pleased to clarify the raised issue.

“HRTEM image of NiCo-SAD-NC”

The enlarged-view of the HRTEM image of NiCo-SAD-NC in **Supplementary Fig. 8e** of the revised manuscript demonstrated the generation of lattice distortion defects characteristics, which might be attributed to the coordination of Ni/Co single atoms with the nitrogen of the carbon support, revealing the presence of carbon lattice defects in the NiCo-SAD-NC [*Angew. Chem. Int. Ed.* **131**, 2648–2652 (2019)].

“Raman spectra of NiCo-SAD-NC and NC”

The Raman spectra of NC and NiCo-SAD-NC showed a characteristic D band at 1338 cm^{-1} and G band at 1591 cm^{-1} , corresponding to carbon lattice defect and sp^2 -hybridized carbon atoms, respectively (**Supplementary Fig. 9**). Interestingly, after introducing Ni/Co single atoms, the ratio between D band and G band (I_d/I_g) of NiCo-SAD-NC (1.10) slightly increased compared to NC (0.96), possibly suggesting the formation of more defects created by the co-ordination between Ni and Co atoms with N doped carbon support [*Nano Energy* **71**, 104597 (2020)].

Therefore, combining both the HRTEM and Raman analysis, we concluded that the number of carbon defects in the NiCo-SAD-NC sample increased. The text is modified on **p. 9**, **S11**, and **S12** of the revised manuscript as

The HRTEM image and the Raman spectra of the NiCo-SAD-NC demonstrated the presence of lattice distortion defects and higher intensity of defective carbon (D-band), respectively, further revealed that more carbon lattice defects were produced due to the co-coordination between Ni and Co with N doped carbon support (Supplementary Fig. 8e and 9)²¹. (p. 9)

The enlarged view of the HRTEM image (Supplementary Fig. 8e) of NiCo-SAD-NC demonstrated the presence of lattice distortion defects characteristic, which might be attributed to the co-coordination between Ni and Co with N. (p. S11)

The Raman spectra of NC and NiCo-SAD-NC showed a characteristic D band at 1338 cm^{-1} and G band at 1591 cm^{-1} , corresponding to carbon lattice defect and sp^2 -hybridized carbon atoms, respectively (Supplementary Fig. 9). After introducing Ni and Co single atoms, the ratio between D band and G band (I_d/I_g) of NiCo-SAD-NC slightly increased from 0.96 to 1.10, possibly owing to the formation of more defects created by the co-coordination between Ni and Co with N doped carbon support. (p. S12)

Supplementary Figure 8 | **a**, **b**, and **c**, TEM, HRTEM and STEM-HAADF images with the corresponding element maps (N, Co, Ni), respectively, of NiCo-NP-NC. The inset in **b** shows the corresponding SAED pattern. **d**, and **e**, TEM, and HRTEM image, respectively, of NiCo-SAD-NC.

Supplementary Figure 9 | a, and b, Raman spectra of NC and NiCo-SAD-NC, respectively.

3) Electronic microscopy: One of the very strong evidence for the presence of dual Ni-Co sites is the observation by HAADF-STEM and EELs of Ni and Co intensities next to each other. This observation is, however, quite subjective. Looking at figure 2b, I could draw circle around bright spots that would certainly not correspond to the ones chosen by the authors. Also, the intensity of the Ni or Co peaks are not so evident on the EELS spectra. It would be more convincing if a series of spectra were shown in supplementary information, in order to see what the intensities look like for different spots. The same is true for the intensity profiles that show the presence of Co or Ni and their interatomic distances. It would be very beneficial to see a much longer line, in order to estimate the noise level and the sensitivity of the intensity profile measurements.

Answer 3) We thank the referee for carefully reviewing our manuscript and providing valuable suggestions for improving the quality of our manuscript. We agree with the referee that our previous HAADF-STEM image along with the intensity profile and EEL spectra for the NiCo-SAD-NC were not so evident. Following the referee's suggestions, we newly collected a much improved aberration-corrected HAADF-STEM image of the NiCo-SAD-NC, clearly showing the presence of multiple NiCo dimer sites and further being verified by using clear intensity profiles and EEL spectra in Figs. 2b-e and Supplementary Fig. 10 of the revised manuscript. The improved aberration-corrected HAADF-STEM image of NiCo-SAD-NC in Fig. 2b clearly demonstrated the existence of isolated Ni-Co dimer sites (marked by yellow square) with coordination between Ni and Co atoms at the atomic level along with some isolated Ni or Co atoms (marked by the orange circle). The intensity profiles along with the corresponding EEL spectra taken at multiple homogeneously distributed bright dual dots that was marked by yellow squares further confirmed the existence of Ni-Co dimer sites with an average dimer distance of 0.241 ± 0.024 nm (Figs. 2c,d,e, and Supplementary Fig. 10). Supplementary Fig. 10 showed a series of intensity profiles and EEL spectra taken at multiple dimer sites marked by yellow squares, clearly providing convincing evidence for the existence of multiple NiCo dimer sites in the NiCo-SAD-NC. The newly collected

EEL spectra of the NiCo dimer sites are better than the previously reported Co-Fe dual-atom pair catalysts [*Energy Environ. Sci.*, **11**, 3375—3379 (2018)] and clearly provided the evidence for the existence of NiCo dimer in the NiCo-SAD-NC.

In addition, to further provide the evidence for the existence of NiCo dimer sites in the NiCo-SAD-NC sample, we carried out proper Ni/Co K-edge EXAFS fitting in R space oscillations for determining the structural information of NiCo-SAD-NC such as Ni/Co-N and Ni-Co coordination number (CN) and bond distance, newly provided in **Supplementary Fig. 19** and **Supplementary Table S5** of the revised manuscript. The Ni/Co K-edge EXAFS fitting results clearly revealed that the CNs of Ni-N and Co-N were 3.3 and 3.1, respectively, with the bond distance of 1.86 Å (Ni-N) and 2.01 Å (Co-N), suggesting that the NiCo-SAD-NC was dominated by the Ni-N₃ and Co-N₃ environment. Meanwhile, the Ni-Co and Co-Ni path were fitted at a position of 2.55 Å with a CN of 0.7 and 0.6, respectively, directly indicating the existence of Ni-Co bonding dimer in the form of NiCo-N₆ structure, in consistent with the DFT calculated structure. The obtained CNs for Ni-N/Co-N and Ni-Co/Co-Ni were consistent with the previously reported Co-Fe and Co-Zn dual-atom pair catalysts [*Angew. Chem. Int. Ed.* **131**, 2648–2652 (2019); *Energy Environ. Sci.*, **11**, 3375—3379 (2018)].

Therefore, combining the much-improved aberration-corrected HAADF-STEM image, series of intensity profiles, and EEL spectra along with proper EXAFS fitting clearly provided the evidence for the existence of NiCo dimer sites in the NiCo-SAD-NC. The text is modified on **p. 9**, and **12** of the revised manuscript as

Aberration-corrected HAADF-STEM image in Fig. 2b clearly demonstrated the existence of isolated Ni-Co dimer sites (marked by yellow square) with coordination between Ni and Co at atomic level along with some isolated Ni or Co atoms (marked by the orange circle). The homogeneously distributed bright dual dots marked by yellow squares confirmed the existence of Ni-Co dual sites, verified using the intensity profile and corresponding electron energy loss (EEL) spectra (Figs. 2c,d). The bright Ni-Co dual dots were clearly identified in the intensity profiles and corresponding EEL spectrum, suggesting the possible formation of metal-metal bonds with an average dimer distance of 0.241 ± 0.024 nm, obtained from the statistical analysis over multiple dimer sites (Fig. 2e, and Supplementary Fig. 10). (p. 9)

Furthermore, the structural information of NiCo-SAD-NC along with the Ni/Co-N and Ni-Co coordination number (CN) was also affirmed by the EXAFS fitting results in R space oscillations (Supplementary Fig. 19 and Supplementary Table 5). The EXAFS fitting results clearly revealed that the CNs of Ni-N and Co-N were 3.3 and 3.1, respectively, suggesting that the NiCo-SAD-NC was dominated by the Ni-N₃ and Co-N₃ environment. Meanwhile, the Ni-Co/Co-Ni path was fitted at a position of 2.55 Å with a CN of 0.7/0.6, directly indicating the existence of Ni-Co bonding dimer in the form of NiCo-N₆ structure, consistent with the DFT calculated structure. (p. 12)

Fig. 2 | Structural analysis and electron microscopy of NiCo-SAD-NC. **a**, XRD spectra of NC, NiCo-NP-NC, and NiCo-SAD-NC. **b**, Aberration-corrected HAADF-STEM image of the NiCo-SAD-NC. The yellow squares in **(b)** show the dimer sites, and some of the single Ni/Co-atom sites are highlighted by orange circles. **c,d**, The intensity profile and corresponding EEL spectra obtained at site A **(c)** and site B **(d)** for NiCo-SAD-NC indicate that Ni and Co are coordinated at the atomic scale. **e**, Statistical Ni-Co distance in the observed dimers. **f**, HAADF-STEM image and corresponding EDS maps of NiCo-SAD-NC showing the uniform dispersion of N (green), Co (red) and Ni (yellow).

Supplementary Figure 10 | a, Aberration-corrected HAADF-STEM image of the NiCo-SAD-NC. The yellow squares in (a) show the dimer sites, and some of the single Ni/Co-atom sites are highlighted by orange circles. **b**, The intensity profile is obtained from sites 1-10 showing the distance between Ni and Co in the observed dimers at the atomic scale. **c**, The corresponding EEL spectra obtained from sites 1-7, 9, and site i, ii show the Ni and Co coordination in the dimer along with few Ni/Co single-atom sites.

Previous Fig. 2 | Structural analysis and electron microscopy of NiCo-SAD-NC. **a**, XRD spectra of NC, NiCo-NP-NC and NiCo-SAD-NC. **b,c**, Aberration-corrected HAADF-STEM image (**b**) and magnified view (**c**) of the NiCo-SAD-NC. The red circles in (**c**) shows the dimer sites and the inset shows the corresponding EEL spectra taken at site 2 indicate Ni and Co are coordinated at the atomic scale. **d**, The intensity profile obtained at NiCo dimer sites 1 and 2. **e**, Statistical Ni-Co distance in the observed dimers. **f**, HAADF-STEM image and corresponding EDS maps of NiCo-SAD-NC showing the uniform dispersion of N (green), Co (red) and Ni (yellow). (Previous Fig. 2 in our original manuscript for reference to the referee)

Supplementary Figure 19 | **a**, and **b**, The Ni K-edge, and Co K-edge FT-EXAFS, respectively, of NiCo-SAD-NC and Ni/Co metal with corresponding EXAFS fitting curves. **c**, The DFT simulated NiCo-SAD-NC model based on the FT-EXAFS fitting value.

Supplementary Table 5. EXAFS fitting parameters at the Ni and Co K-edge for NiCo-SAD-NC and Ni/Co metal ($S_0^2 = 0.780$ (Ni), 0.816 (Co)).

Sample	Shell	N^a	$R(\text{\AA})^b$	$\sigma^2(\text{\AA}^2)^c$	$\Delta E_0(\text{eV})^d$	R factor
Co K-edge						
Co Metal	Co-Co	12	2.49	0.0060	6.6	0.0002
NiCo-SAD-NC	Co-N	3.1	2.01	0.0018	-4.5	0.0001
	Co-Ni	0.6	2.55	0.0126		
Ni K-edge						
Ni Metal	Ni-Ni	12	2.48	0.0059	6.3	0.0001
NiCo-SAD-NC	Ni-N	3.3	1.86	0.0039	-4.8	0.0016
	Ni-Co	0.7	2.55	0.0121		

^a N : coordination numbers; ^b R : bond distance; ^c σ^2 : Debye-Waller factors; ^d ΔE_0 : the inner potential correction. R factor: goodness of fit. S_0^2 was set to 0.816 for Co and 0.780 for Ni, according to the experimental EXAFS fit of Co and Ni metal reference by fixing CN as the known crystallographic value.

4) Both the EELS and intensity profiles are not so obvious to see, so it is important to have a statistical analysis to make sure that this is not just one event here and there.

Answer 4) Following the referee's suggestions, we newly collected much improved aberration-corrected HAADF-STEM image of the NiCo-SAD-NC along with the series of intensity profiles and EEL spectra taken at multiple dimer sites marked by yellow squares, newly provided in **Supplementary Fig. 10** of the revised manuscript. As shown in **Supplementary Fig. 10**, a series of intensity profiles and EEL spectra taken at multiple dimer sites marked by yellow squares clearly provides obvious evidence for the existence of multiple NiCo dimer sites in the NiCo-SAD-NC. The text is modified on **p. 9** of the revised manuscript as

The bright Ni-Co dual dots were clearly identified in the intensity profiles and corresponding EEL spectrum, suggesting the possible formation of metal-metal bonds with an average dimer distance of 0.241 ± 0.024 nm, obtained from the statistical analysis over multiple dimer sites (Fig. 2e, and Supplementary Fig. 10).

Supplementary Figure 10 | **a**, Aberration-corrected HAADF-STEM image of the NiCo-SAD-NC. The yellow squares in **(a)** show the dimer sites and some of the single Ni/Co-atom sites are highlighted by orange circles. **b**, The intensity profile obtained from sites 1-10 showing the distance between Ni and Co in the observed dimers at the atomic scale. **c**, The corresponding EEL spectra obtained from sites 1-7, 9, and site i, ii show the Ni and Co coordination in the dimer along with few Ni/Co single-atom sites.

5) The EDS elemental maps on figure 2f should be superimposed (at least the nickel and cobalt distribution) to make sure that they are colocalized.

Answer 5) We thank the referee for the nice suggestion. Following the referee's suggestion, we newly provided the superimposed EDS elemental mapping of NiCo-SAD-NC in **Supplementary Fig. 11** of the revised manuscript, revealing the co-localization of Ni, Co, and N atoms. The text is modified on **p. 9** and **10** of the revised manuscript as

Meanwhile, HAADF-STEM and EDS elemental mapping revealed that N, Ni, and Co atoms were co-localized and homogeneously dispersed in the NiCo-SAD-NC (Fig. 2f and **Supplementary Fig. 11**). (p. 9,10)

Supplementary Figure 11 | Overlapping EDS map of NiCo-SAD-NC showing co-localized Ni, Co, and N atoms.

6) XPS: The signal-to-noise ratio of the Ni and Co XPS data (figure 10b, c) is quite poor, and a significant smoothing was applied to obtain “nice” data. How reasonable is that? For example, the comparison of Co-SA-NC and NiCo-SAD-NC is not convincing when looking at the raw data. There is no convincing argument to attribute the shifts observed in the Ni and Co 2p_{3/2} data. The conclusions could be interesting, but there should be more comparison with reference samples and/or literature. If more reference spectra were measured, better comparison could be made and tentative oxidation states could be attributed, which could correlate with those observed in XAS (vide infra).

Answer 6) We thank the referee for raising the concern regarding the XPS data and for the valuable suggestion. We agree that the signal-to-noise ratio of the Ni and Co XPS data was low. Since the XPS is a surface technique and the isolated single Ni/Co atoms in the NiCo-SAD-NC, and Ni/Co-SA-NC samples with a total metal loading of only ~ 7 wt% are uniformly dispersed throughout the carbon support, it is very difficult to probe all the Ni or Co atoms using XPS surface-technique. Therefore, the intensity of the signal-to-noise ratio for the Ni/Co XPS data was low as only a fraction of the metal atoms in the samples probed by the XPS. However, to observe the shift in the Ni/Co 2p_{3/2} XPS raw data, we plotted the Ni/Co 2p_{3/2} raw data of the NiCo-SAD-NC and compared it with that of Ni/Co-SA-NC (Rep. Fig. 3). As shown in Rep. Fig. 3, the intensities of signal-to-noise ratio for all the samples were around 10 or higher, which suggested that the signal can be distinguishable from the noise. From the raw data of Ni 2p_{3/2} XPS spectra (Rep. Fig. 3a), the XPS peak position of the NiCo-SAD-NC was positively shifted compared to the XPS peak position of Ni-SA-NC. Similarly, from the raw data of Co 2p_{3/2} XPS spectra (Rep. Fig. 3b), the XPS peak position of the NiCo-SAD-NC was negatively shifted compared to the XPS peak position of Co-SA-NC. Therefore, from the Ni/Co XPS raw data, we can confirm that the peaks were shifted between the NiCo-SAD-NC and Ni/Co-SA-NC samples.

Following the referee suggestions, we newly measured the XPS spectra of suitable reference samples such as nickel(II) phthalocyanine (NiPC), and cobalt(II) phthalocyanine (CoPC) and compared with the Ni/Co 2p_{3/2} fitted deconvoluted XPS spectra of NiCo-SAD-NC and Ni/Co-SA-NC along with additional reference samples from the literature such as Ni metal [*RSC Adv.*, **7**, 14152–14158 (2017)] and Co metal [*Journal of Electroanalytical Chemistry* **822**, 33–42 (2018)] to estimate the tentative oxidation states of the Ni and Co, newly provided in Supplementary Fig. 14 of the revised manuscript. As revealed in Supplementary Fig. 14a,b, the oxidation state of Ni in NiCo-SAD-NC is +1.73, which was higher than the oxidation state of Ni in Ni-SA-NC (+1.57). Contrarily, the oxidation state of Co in NiCo-SAD-NC (+1.39) was negatively shifted compared to Co in Co-SA-NC (+1.67), suggesting that the electron transfer occurred from Ni to Co site at the atomic interface of NiCo-SAD-NC, probably due to Ni-Co bond formation at the atomic level (Supplementary Fig. 14c,d).

To correlate the oxidation states of Ni/Co estimated from the XPS with those from XANES analysis, we newly measured the XANES spectra of NiPC and CoPC following the referee’s suggestions and compared with the Ni/Co K-edge XANES spectra of NiCo-SAD-NC, Ni/Co-SA-

NC, and Ni/Co metal to estimate the oxidation state of Ni/Co from their corresponding XANES energy at half-edge jump, newly provided in Supplementary Fig. 16 of the revised manuscript. As shown in Supplementary Fig. 16a,b, the oxidation state of Ni in NiCo-SAD-NC is +1.68, which was positively shifted compared to that of the oxidation state of Ni in Ni-SA-NC (+1.59), in close agreement with the Ni $2p_{3/2}$ XPS results. Similarly, from the Co K-edge XANES spectra (Supplementary Fig. 16c,d), the oxidation state of Co in NiCo-SAD-NC (+1.35) was negatively shifted compared to Co in Co-SA-NC (+1.72), here again in close agreement with the oxidation state of Co estimated from the Co $2p_{3/2}$ XPS results.

Combining both XPS and XANES results, we can conclude that there was electron transfer occurred from Ni to Co site at the atomic interface of NiCo-SAD-NC.

Also, we noticed that the oxidation state of Ni/Co in the NiCo-NP-NC estimated from the XPS (Supplementary Fig. 13) was slightly higher than the literature value, which may be probably due to the surface oxidation of the NiCo nanoparticles probed by the XPS surface-technique. However, the XAS bulk-analysis using the hard-X-ray clearly confirmed the metallic nature of NiCo nanoparticles matching well with that of Ni/Co metal foam (Fig. 3a-d).

The text is modified on p. 10 and 11 of the revised manuscript as

Compared to NiCo-NP-NC, the binding energies for Ni-SA-NC and Co-SA-NC were positively shifted after introducing N to trap the single-atom sites, revealing Ni-N and Co-N bond formation²⁸. However, after forming the NiCo dimer sites, the Ni $2p_{3/2}$ XPS spectra of NiCo-SAD-NC showed a positive shift with Ni oxidation state of +1.73 eV compared to that of Ni in Ni-SA-NC (+1.57), while the Co $2p_{3/2}$ XPS spectra of NiCo-SAD-NC exhibited a negative shift with Co oxidation state of +1.39 compared to Co in Co-SA-NC (+1.67), suggesting that the electron transfer occurred from Ni to Co site at the atomic interface of NiCo-SAD-NC, probably due to single Ni-Co bond formation at the atomic level (Supplementary Fig. 14). (p. 10)

The precise oxidation state of Ni in NiCo-SAD-NC and Ni-SA-NC were +1.68 and +1.59, respectively, revealed from the further fitting of the Ni K-edge XANES energy at half-edge jump, in close agreement with the Ni oxidation states estimated from the XPS results (Supplementary Figs. 16a,b). (p. 11)

Here too, in agreement with the Co 2p XPS results, the near edge and white line feature in the Co K-edge XANES profile of NiCo-SAD-NC showed a negative shift compared to Co-SA-NC, indicating a lower oxidation state of Co in NiCo-SAD-NC (+1.35) compared to Co-SA-NC (+1.72) (insets of Fig. 3b and Supplementary Figs. 16c,d). (p. 11)

Representative Fig. 3 | a, Ni 2p_{3/2} XPS raw data of NiCo-SAD-NC, and Ni-SA-NC. **b,** Co 2p_{3/2} XPS raw data of NiCo-SAD-NC, and Co-SA-NC.

Supplementary Figure 14 | **a**, and **b**, Fitted deconvoluted Ni $2p_{3/2}$ XPS spectra and the Ni oxidation states analyzed by the XPS peak position, respectively, of NiCo-SAD-NC, and Ni-SA-NC with reference samples. **c**, and **d**, Fitted deconvoluted Co $2p_{3/2}$ XPS spectra and the Co oxidation states analyzed by the XPS peak position, respectively, of NiCo-SAD-NC, and Co-SA-NC with reference samples.

Supplementary Figure 16 | **a**, and **b**, Experimental Ni K-edge XANES spectra of NiCo-SAD-NC with reference samples and the Ni oxidation state analysis by corresponding XANES energy at half-edge jump, respectively. **c**, and **d**, Experimental Co K-edge XANES spectra of NiCo-SAD-NC with reference samples and the Co oxidation state analysis by corresponding XANES energy at half-edge jump, respectively.

Supplementary Figure 13 | **a**, Fitted deconvoluted N $1s$ XPS spectra of NiCo-SAD-NC and NiCo-NP-NC. **b**, and **c**, Fitted deconvoluted Ni $2p$ and Co $2p$, respectively, XPS spectra of NiCo-SAD-NC, NiCo-NP-NC, Co-SA-NC, and Ni-SA-NC.

Fig. 3 | **Structural characterization by X-ray absorption spectroscopy.** **a,b**, Experimental Ni K-edge (**a**) and Co K-edge (**b**) XANES spectra of NiCo-SAD-NC with reference samples. The inset zooms in over the near edge (bottom) and white line (top) feature. **c,d**, Experimental Ni K-edge (**c**) and Co K-edge (**d**) FT-EXAFS spectra of NiCo-SAD-NC along with reference samples.

7) XANES/EXAFS: The fact that a pre-edge can be observed for both the Ni and Co samples is not an evidence for a square planar geometry. It is one possibility, but many other are possible since pre-edge features arise from any geometry except from a perfect octahedron (which never exists). Unless a very good resolution is obtained (with a high-resolution spectrometer) and appropriate reference samples are used, the geometry of the metal site cannot be precisely determined. DFT calculations could also help if the pre-edge features were calculated.

Answer 7) Following the referee's suggestion, we newly measured the XANES spectra of appropriate reference samples such as nickel(II) phthalocyanine (NiPC) and cobalt(II) phthalocyanine (CoPC) and compared them with the Ni/Co K-edge XANES spectra of NiCo-SAD-NC to precisely determine the geometry of metal sites, newly provided in **Supplementary Fig. 15** of the revised manuscript. In the Ni and Co K-edge XANES spectra, a pre-edge peak was observed for NiCo-SAD-NC and standard NiPC/CoPC, due to a $1s$ to $4p_z$ shakedown transition of square-planar coordination with D_{4h} local symmetry (**Supplementary Fig. 15a,b**) [*Nano Energy* **71**, 104597 (2020)]. The weak pre-edge peak intensity of NiPC and CoPC suggested its high D_{4h} centro-symmetry [*Nat. Energy* **3**, 140–147 (2018)]. Compared to NiPC and CoPC, the increased pre-edge peak intensity in NiCo-SAD-NC was ascribed to the distorted D_{4h} local symmetry, further implying that the Ni and Co atom centers were coordinated with four nearest guest atoms (N or metal), but the D_{4h} symmetry was distorted due to another coordination such as metal-metal (**Supplementary Fig. 15a,b**) [*Angew. Chem. Int. Ed.* **58**, 6972–6976 (2019)]. Therefore, comparing the pre-edge features in the XANES spectra of the NiCo-SAD-NC with the standard NiPC/CoPC, we configured the D_{4h} local geometry of the metal centers. The text is modified on **p.10, 11, and S14** of the revised manuscript as

Compared to Ni foil and NiCo-NP-NC, a pre-edge peak around 8333.8 eV can be observed in the Ni K-edge XANES spectra of NiCo-SAD-NC, Ni-SA-NC, and **standard nickel phthalocyanine (NiPC)**, due to a $1s$ to $4p_z$ shakedown transition of square-planar coordination with D_{4h} local symmetry (Fig. 3a and **Supplementary Fig. 15a**)²⁷. **The weak pre-edge peak intensity of NiPC suggested its high D_{4h} centro-symmetry. Compared to NiPC, the increased pre-edge peak intensity in NiCo-SAD-NC was ascribed to the distorted D_{4h} local symmetry, further implying that the single Ni atom was coordinated with four nearest guest atoms (N or metal) with distorted D_{4h} symmetry due to another possible coordination such as metal-metal (Supplementary Fig. 15a)**^{22,28}. (p. 10, 11)

A similar pre-edge peak around 7709.5 eV also appeared for NiCo-SAD-NC, Co-SA-NC, and **standard cobalt phthalocyanine (CoPC)** in the Co K-edge profile, suggesting the presence of X-ray absorbing Co centers with four coordination (N or metal) (Fig. 3b and **Supplementary Fig. 15b**). (p. 11)

The increased pre-edge peak intensity in NiCo-SAD-NC compared to NiPC and CoPC in the Ni and Co K-edge XANES spectra, respectively, are attributed to the distorted D_{4h} local symmetry, suggesting that the Ni and Co single atom centers are coordinated with four nearest guest atoms (N or metal), but the D_{4h} symmetry was distorted by another coordination such as metal-metal (Supplementary Fig. 15). (p. S17)

Supplementary Figure 15 | **a**, Experimental Ni K-edge XANES spectra, of NiCo-SAD-NC with Ni-SA-NC and NiPC (Inset in **a** shows the pre-edge feature). **b**, Experimental Co K-edge XANES spectra of NiCo-SAD-NC with Co-SA-NC and CoPC (Inset in **b** shows the pre-edge feature).

8) The determination of the oxidation state of Ni and Co is done by interpolating the edge energy position of the samples of interest between the positions of the corresponding metallic metal and their oxides. The edge position energy is used as a marker, but there is no mention as to how it is determined. It seems to be the first inflexion point, but it does not reflect the reality. Although there is a slight shift in the pre-edge position of the NiCo-SAD-NC and Ni-SA-NC it is clearly not equal to the 2 eV reported on figure 11b. The method by which the edge position is determined should be clearly mentioned.

Answer 8) We sincerely apologize for not clearly mentioning the determination of the oxidation state of Ni and Co from the XANES edge energy. Following the referee's suggestions, we newly measured the XANES spectra of suitable reference samples (NiPC and CoPC) and compared them with the Ni/Co K-edge XANES spectra of NiCo-SAD-NC, Ni/Co-SA-NC, and Ni/Co metal to estimate the oxidation state of Ni/Co from their corresponding XANES energy **at half-edge jump**, newly provided in **Supplementary Fig. 16** of the revised manuscript. Now, we considered the XANES edge energy at half-edge jump to determine the tentative oxidation states of Ni and Co in the NiCo-SAD-NC and Ni/Co-SA-NC. As shown in **Supplementary Fig. 16a,b**, the oxidation state of Ni in NiCo-SAD-NC is +1.68, which was positively shifted compared to that of the oxidation

state of Ni in Ni-SA-NC (+1.59). Similarly, from the Co K-edge XANES spectra (Supplementary Fig. 16c,d), the oxidation state of Co in NiCo-SAD-NC (+1.35) was negatively shifted compared to Co in Co-SA-NC (+1.72), both in close agreement with the oxidation state of Ni and Co estimated from the Ni and Co 2p_{3/2} XPS results. The text is modified on p. 11 of the revised manuscript as

The precise oxidation state of Ni in NiCo-SAD-NC and Ni-SA-NC were +1.68 and +1.59, respectively, revealed from the further fitting of the Ni K-edge XANES energy at half-edge jump, in close agreement with the Ni oxidation states estimated from the XPS results (Supplementary Figs. 16a,b) (p. 11)

Here too, in agreement with the Co 2p XPS results, the near edge and white line feature in the Co K-edge XANES profile of NiCo-SAD-NC showed a negative shift compared to Co-SA-NC, indicating a lower oxidation state of Co in NiCo-SAD-NC (+1.35) compared to Co-SA-NC (+1.72) (insets of Fig. 3b and Supplementary Figs. 16c,d). (p. 11)

Supplementary Figure 16 | **a**, and **b**, Experimental Ni K-edge XANES spectra of NiCo-SAD-NC with reference samples and the Ni oxidation state analysis by corresponding XANES energy at half-edge jump, respectively. **c**, and **d**, Experimental Co K-edge XANES spectra of NiCo-SAD-NC with reference samples and the Co oxidation state analysis by corresponding XANES energy at half-edge jump, respectively.

9) Considering the nickel case, if the edge position is considered at half-edge jump, then the oxidation states of the Ni-SA and NiCo-SAD samples should be higher than +II. This would actually be more reasonable than a +0.5 or +1.0 oxidation state, which would be very unusual for a nickel ion in an N-coordinating environment.

Answer 9) Following the referee's suggestion, we newly measured and compared the Ni K-edge XANES spectra of suitable reference samples (NiPC) with NiCo-SAD-NC and Ni-SA-NC and considered the XANES energy at half-edge jump for determining the Ni oxidation states, newly provided in **Supplementary Fig. 16a,b** of the revised manuscript. As shown in **Supplementary Fig. 16a,b**, the oxidation state of Ni in NiCo-SAD-NC was +1.68 and the oxidation state of Ni in Ni-SA-NC was +1.59, both were close to that of NiPC, which is quite reasonable for metal ion in N-coordinating environment [*Angew. Chem. Int. Ed.* **58**, 6972–6976 (2019)]. The text is modified on **p. 11** of the revised manuscript as

The precise oxidation state of Ni in NiCo-SAD-NC and Ni-SA-NC were +1.68 and +1.59, respectively, revealed from the further fitting of the Ni K-edge XANES energy at half-edge jump, in close agreement with the Ni oxidation states estimated from the XPS results (Supplementary Figs. 16a,b). (p. 11)

Supplementary Figure 16 | **a**, and **b**, Experimental Ni K-edge XANES spectra of NiCo-SAD-NC with reference samples and the Ni oxidation state analysis by corresponding XANES energy at half-edge jump, respectively. **c**, and **d**, Experimental Co K-edge XANES spectra of NiCo-SAD-NC with reference samples and the Co oxidation state analysis by corresponding XANES energy at half-edge jump, respectively.

10) Considering the cobalt case, the CoO reference sample looks to me like a Co₃O₄ sample (spinel), which is a mixed +II/+III oxidation state. Considering the reference samples used and an edge position at half-edge jump, then indeed, the oxidation state of coal in Co-SA and NiCo-SAD would be around +0.5-1.0. But again, this oxidation state is quite exotic for an N-coordinated cobalt complex.

Answer 10) We are very grateful to the referee for carefully reviewing and pointing out that the CoO reference sample looks like Co₃O₄, which is a mixed +II/+III oxidation state. We confirmed that there were some problems with the previous oxide reference sample. Now, we replaced the metal oxide reference with a more suitable reference sample (CoPC) and compared the Co K-edge XANES spectra of CoPC with NiCo-SAD-NC and Co-SA-NC for determining the Co oxidation

states, considering the XANES energy at half-edge jump, newly provided in Supplementary Fig. 16c,d of the revised manuscript. As shown in Supplementary Fig. 16c,d, the oxidation state of Co in Co-SA-NC was +1.72 and the oxidation state of Co in NiCo-SAD-NC was +1.35, which is quite reasonable for metal ions in N-coordinating environment [*Angew. Chem. Int. Ed.* **58**, 6972–6976 (2019)] and in close agreement with the Co oxidation states estimated from the XPS results. The text is modified on p. 11 of the revised manuscript as

Here again, in agreement with the Co 2p XPS results, the near edge and white line feature in the Co K-edge XANES profile of NiCo-SAD-NC showed a negative shift compared to Co-SA-NC, indicating a lower oxidation state of Co in NiCo-SAD-NC (+1.35) compared to Co-SA-NC (+1.72) (insets of Fig. 3b and Supplementary Figs. 16c,d).

Supplementary Figure 16 | **a**, and **b**, Experimental Ni K-edge XANES spectra of NiCo-SAD-NC with reference samples and the Ni oxidation state analysis by corresponding XANES energy at half-edge jump, respectively. **c**, and **d**, Experimental Co K-edge XANES spectra of NiCo-SAD-NC with reference samples and the Co oxidation state analysis by corresponding XANES energy at half-edge jump, respectively.

11) Whether for nickel or cobalt, better references than oxides should be used to determine the oxidation state of the nickel and cobalt ions in the M-SA and NiCo-SAD samples. Metal porphyrins, phthalocyanines or other types of molecular, N-coordinating and delocalized systems would be more appropriate.

Answer 11) We thank the referee for the nice suggestion. Following the referee's suggestions, we newly measured the XANES spectra of suitable references such as nickel phthalocyanine (NiPC), and cobalt phthalocyanine (CoPC) and compared with the Ni/Co K-edge XANES spectra of NiCo-SAD-NC, Ni/Co-SA-NC, and Ni/Co metal to estimate the oxidation state of Ni/Co from their corresponding XANES energy at half-edge jump, newly provided in **Supplementary Fig. 16** of the revised manuscript. As shown in **Supplementary Fig. 16a,b**, the oxidation state of Ni in NiCo-SAD-NC was +1.68, which was positively shifted compared to that of the oxidation state of Ni in Ni-SA-NC (+1.59), in close agreement with the Ni 2p_{3/2} XPS results. Similarly, from the Co K-edge XANES spectra (**Supplementary Fig. 16c,d**), the oxidation state of Co in NiCo-SAD-NC (+1.35) was negatively shifted compared to Co in Co-SA-NC (+1.72), again in close agreement with the oxidation state of Co estimated from the Co 2p_{3/2} XPS results. Now, the oxidation states of the Ni/Co metal atoms were closer to their respective M-PC reference samples, which is quite reasonable for metal ions in N-coordinating environment [*Angew. Chem. Int. Ed.* **58**, 6972–6976 (2019)]. The text is modified on **p. 11** of the revised manuscript as

The precise oxidation state of Ni in NiCo-SAD-NC and Ni-SA-NC were +1.68 and +1.59, respectively, revealed from the further fitting of the Ni K-edge XANES energy at half-edge jump, in close agreement with the Ni oxidation states estimated from the XPS results (**Supplementary Figs. 16a,b**). (p. 11)

Here too, in agreement with the Co 2p XPS results, the near edge and white line feature in the Co K-edge XANES profile of NiCo-SAD-NC showed a negative shift compared to Co-SA-NC, indicating a lower oxidation state of Co in NiCo-SAD-NC (+1.35) compared to Co-SA-NC (+1.72) (insets of Fig. 3b and **Supplementary Figs. 16c,d**). (p. 11)

Supplementary Figure 16 | a, and b, Experimental Ni K-edge XANES spectra of NiCo-SAD-NC with reference samples and the Ni oxidation state analysis by corresponding XANES energy at half-edge jump, respectively. c, and d, Experimental Co K-edge XANES spectra of NiCo-SAD-NC with reference samples and the Co oxidation state analysis by corresponding XANES energy at half-edge jump, respectively.

12) The determination state of nickel and cobalt determined by XAS should be compared to the oxidation states obtained by XPS, which are not specifically determined.

Answer 12) Following the referee's suggestions, we newly determined the oxidation states of the Ni/Co in the NiCo-SAD-NC and Ni/Co-SA-NC from the XPS and XAS by comparing with suitable references (NiPC/CoPC) and newly provided in **Supplementary Fig. 14 and 16** of the revised manuscript.

Oxidation states from the XPS:

We newly measured the XPS spectra of suitable reference samples such as nickel(II) phthalocyanine (NiPC), and cobalt(II) phthalocyanine (CoPC) and compared with the Ni/Co 2p_{3/2}

fitted deconvoluted XPS spectra of NiCo-SAD-NC and Ni/Co-SA-NC along with additional reference samples from the literature such as Ni metal (*RSC Adv.*, **7**, 14152–14158 (2017)) and Co metal (*Journal of Electroanalytical Chemistry* **822**, 33–42 (2018)) to estimate the tentative oxidation states of the Ni and Co, newly provided in **Supplementary Fig. 14** of the revised manuscript. As revealed in **Supplementary Fig. 14a,b**, the oxidation state of Ni in NiCo-SAD-NC was +1.73, which was higher than the oxidation state of Ni in Ni-SA-NC (+1.57). Contrarily, the oxidation state of Co in NiCo-SAD-NC (+1.39) was negatively shifted compared to Co in Co-SA-NC (+1.67), suggesting that the electron transfer occurred from Ni to Co site at the atomic interface of NiCo-SAD-NC, probably due to Ni-Co bond formation at the atomic level (**Supplementary Fig. 14c,d**).

Oxidation states from the XAS:

To correlate the oxidation states of Ni/Co estimated from the XPS with those from XANES analysis, we newly measured the XANES spectra of NiPC and CoPC following the referee's suggestions and compared with the Ni/Co K-edge XANES spectra of NiCo-SAD-NC, Ni/Co-SA-NC, and Ni/Co metal to estimate the oxidation state of Ni/Co from their corresponding XANES energy at half-edge jump, newly provided in **Supplementary Fig. 16** of the revised manuscript. As shown in **Supplementary Fig. 16a,b**, the oxidation state of Ni in NiCo-SAD-NC was +1.68, which was positively shifted compared to that of the oxidation state of Ni in Ni-SA-NC (+1.59), in close agreement with the Ni 2p_{3/2} XPS results. Similarly, from the Co K-edge XANES spectra (**Supplementary Fig. 16c,d**), the oxidation state of Co in NiCo-SAD-NC (+1.35) was negatively shifted compared to Co in Co-SA-NC (+1.72), again in close agreement with the oxidation state of Co estimated from the Co 2p_{3/2} XPS results.

From the above discussion, we conclude that the oxidation states of Ni/Co determined from the XPS and XAS were in close agreement, and both suggested that there was electron transfer occurred from Ni to Co sites at the atomic interface of NiCo-SAD-NC. The text is modified on **p. 10, and 11** of the revised manuscript as

Compared to NiCo-NP-NC, the binding energies for Ni-SA-NC and Co-SA-NC were positively shifted after introducing N to trap the single-atom sites, revealing Ni-N and Co-N bond formation²⁸. However, after forming the NiCo dimer sites, the Ni 2p_{3/2} XPS spectra of NiCo-SAD-NC showed a positive shift with Ni oxidation state of +1.73 eV compared to that of Ni in Ni-SA-NC (+1.57), while the Co 2p_{3/2} XPS spectra of NiCo-SAD-NC exhibited a negative shift with Co oxidation state of +1.39 compared to Co in Co-SA-NC (+1.67), suggesting that the electron transfer occurred from Ni to Co sites at the atomic interface of NiCo-SAD-NC, probably due to single Ni-Co bond formation at the atomic level (**Supplementary Fig. 14**). (p. 10)

The precise oxidation state of Ni in NiCo-SAD-NC and Ni-SA-NC were +1.68 and +1.59, respectively, revealed from the further fitting of the Ni K-edge XANES energy at half-edge jump, in close agreement with the Ni oxidation states estimated from the XPS results (Supplementary Figs. 16a,b). (p. 11)

Here again, in agreement with the Co 2p XPS results, the near edge and white line feature in the Co K-edge XANES profile of NiCo-SAD-NC showed a negative shift compared to Co-SA-NC, indicating a lower oxidation state of Co in NiCo-SAD-NC (+1.35) compared to Co-SA-NC (+1.72) (insets of Fig. 3b and Supplementary Figs. 16c,d). (p. 11)

Supplementary Figure 14 | a, and b, Fitted deconvoluted Ni $2p_{3/2}$ XPS spectra and the Ni oxidation states analyzed by the XPS peak position, respectively, of NiCo-SAD-NC, and Ni-SA-NC with reference samples. c, and d, Fitted deconvoluted Co $2p_{3/2}$ XPS spectra and the Co oxidation states analyzed by the XPS peak position, respectively, of NiCo-SAD-NC, and Co-SA-NC with reference samples.

Supplementary Figure 16 | a, and b, Experimental Ni K-edge XANES spectra of NiCo-SAD-NC with reference samples and the Ni oxidation state analysis by corresponding XANES energy at half-edge jump, respectively. c, and d, Experimental Co K-edge XANES spectra of NiCo-SAD-NC with reference samples and the Co oxidation state analysis by corresponding XANES energy at half-edge jump, respectively.

13) If the proposed oxidation states of +0.5 -1.0 are maintained despite the recommendations above, there should be additional evidence for their existence, since there are usually very unstable in coordination complexes. EPR spectroscopy could prove useful in confirming these oxidation states.

Answer 13) After following the referee's above-mentioned recommendations of using a suitable reference sample (NiPC/CoPC) for determining the Ni/Co oxidation states in the NiCo-SAD-NC and Ni/Co-SA-NC from the XAS and XPS, the newly estimated the oxidation states of Ni/Co were no more within the range of +0.5-1.0. Now, the Ni/Co oxidation states in the NiCo-SAD-NC and Ni/Co-SA-NC measured from the XAS and XPS are closer to their M-PC reference samples, which is quite reasonable for metal ions in N-coordinating environment [*Angew. Chem. Int. Ed.* **58**, 6972–6976 (2019)].

14) The EXAFS analysis is very poor and only based on visual observation of the spectra. This is not acceptable and proper fits should be performed to determine the actual M-N distances and coordination numbers. The small peak at R'=2 shown as a proof of the existence of a Ni-Co interaction looks more like a wish than actual data. In the nickel case, the peak said to correspond to the Ni-Co interaction is less intense than the noise level, so it is very difficult to consider it as a real feature. Again, fits should provide more solid information for a proper structural discussion.

It is not clear whether the data shown on figure S14 are just deconvolution of the spectra or actual EXAFS fits. If so, then the fitting parameters should be given.

Answer 14) We are very grateful to the referee for carefully reviewing our manuscript and providing valuable suggestions for further strengthening the quality of our manuscript. Following the referee's suggestion, we carried out proper Ni/Co K-edge EXAFS fitting in R space oscillations for determining the structural information of NiCo-SAD-NC such as Ni/Co-N and Ni-Co coordination number (CN) and bond distance, newly provided in **Supplementary Fig. 19** and **Supplementary Table S5** of the revised manuscript. The Ni/Co K-edge EXAFS fitting results clearly revealed that the CN of Ni-N and Co-N were 3.3 and 3.1, respectively, with the bond distance of 1.86 Å (Ni-N) and 2.01 Å (Co-N), suggesting that the NiCo-SAD-NC was dominated by the Ni-N₃ and Co-N₃ environment. Meanwhile, the Ni-Co and Co-Ni paths were fitted at a position of 2.55 Å with a CN of 0.7 and 0.6, respectively, directly indicating the existence of Ni-Co bonding dimer in the form of NiCo-N₆ structure, in consistent with the DFT calculated structure. The obtained CN for Ni-N/Co-N and Ni-Co/Co-Ni were consistent with the previous Co-Fe and Co-Zn dual-atom pair catalysts [*Angew. Chem. Int. Ed.* **131**, 2648–2652 (2019); *Energy Environ. Sci.*, **11**, 3375—3379 (2018)].

Therefore, the proper EXAFS fitting clearly provided the structural information of NiCo-SAD-NC, indicating the existence of NiCo dimer coordinated to N.

The data previously shown in **Supplementary Fig. 14** of our original manuscript was just the deconvolution of the EXAFS spectra. However, we now replaced the previous FT-EXAFS peaks deconvolution (**previous Supplementary Fig. 14**) with proper EXAFS fitting at the Ni/Co K-edge for determining the structural information of NiCo-SAD-NC such as Ni/Co-N and Ni-Co

coordination number (CN) and bond distance, newly provided in Supplementary Fig. 19 and Supplementary Table S5 of the revised manuscript.

The text is modified on p. 12 of the revised manuscript as

Furthermore, the structural information of NiCo-SAD-NC along with the Ni/Co-N and Ni-Co coordination number (CN) was also affirmed by the EXAFS fitting results in R space oscillations (Supplementary Fig. 19 and Supplementary Table 5). The EXAFS fitting results clearly revealed that the CNs of Ni-N and Co-N were 3.3 and 3.1, respectively, suggesting that the NiCo-SAD-NC was dominated by the Ni-N₃ and Co-N₃ environment. Meanwhile, the Ni-Co/Co-Ni paths were fitted at a position of 2.55 Å with a CN of 0.7/0.6, directly indicating the existence of Ni-Co bonding dimer in the form of NiCo-N₆ structure, consistent with the DFT calculated structure.

Supplementary Figure 19 | **a**, and **b**, The Ni K-edge, and Co K-edge FT-EXAFS, respectively, of NiCo-SAD-NC and Ni/Co metal with corresponding EXAFS fitting curves. **c**, The DFT simulated NiCo-SAD-NC model based on the FT-EXAFS fitting value.

Supplementary Table 5. EXAFS fitting parameters at the Ni and Co K-edge for NiCo-SAD-NC and Ni/Co metal ($S_0^2 = 0.780$ (Ni), 0.816 (Co)).

Sample	Shell	N^a	$R(\text{\AA})^b$	$\sigma^2(\text{\AA}^2)^c$	$\Delta E_0(\text{eV})^d$	R factor
Co K-edge						
Co Metal	Co-Co	12	2.49	0.0060	6.6	0.0002
NiCo-SAD-NC	Co-N	3.1	2.01	0.0018	-4.5	0.0001
	Co-Ni	0.6	2.55	0.0126		
Ni K-edge						
Ni Metal	Ni-Ni	12	2.48	0.0059	6.3	0.0001
NiCo-SAD-NC	Ni-N	3.3	1.86	0.0039	-4.8	0.0016
	Ni-Co	0.7	2.55	0.0121		

$^a N$: coordination numbers; $^b R$: bond distance; $^c \sigma^2$: Debye-Waller factors; $^d \Delta E_0$: the inner potential correction. R factor: goodness of fit. S_0^2 was set to 0.816 for Co and 0.780 for Ni, according to the experimental EXAFS fit of Co and Ni metal reference by fixing CN as the known crystallographic value.

Previous Supplementary Figure 14 | The FT-EXAFS peaks deconvolution of NiCo-SAD-NC at **a** Ni K-edge and **b** Co K-edge (**Previous Supplementary Figure 14** in our original manuscript for reference to the referee).

15) Finally, there is not characterization of the material after electrochemical activity, even under ex situ conditions. Without such information, it is not reasonable to attribute the particular catalytic performances of this material to the dimeric heterometallic structure postulated.

Answer 15) We agree with the referee that the characterizations of the catalyst after the stability test are very necessary. Following the referee's suggestion, we newly collected post-stability aberration-corrected HAADF-STEM image, EEL spectra, intensity profile, XANES, and EXAFS spectra for the NiCo-SAD-NC after the stability test in both alkaline/acidic media, newly provided in **Supplementary Fig. 34** of the revised manuscript. As revealed in **Supplementary Fig. 34a-d**, post-stability aberration-corrected HAADF-STEM images along with intensity profiles and EEL spectra taken at site A confirmed the existence of NiCo dimer sites in the NiCo-SAD-NC after the stability test in both alkaline and acidic media. The nearly matching Ni/Co K-edge XANES spectra of NiCo-SAD-NC taken before and after the alkaline/acidic stability test further confirmed that the Ni/Co oxidation states along with pre-edge features also remained intact (**Supplementary Fig. 34e,g**). Similarly, the Ni/Co K-edge FT-EXAFS spectra confirmed that the Ni/Co-N and Ni-Co bonding in the NiCo-SAD-NC also remained intact after the stability test (**Supplementary Fig. 34f,h**).

Additionally, in our original manuscript, we already provided the post-stability characterizations such as XRD, SEM, TEM, EDX, and XPS of the NiCo-SAD-NC to investigate any structural and electronic changes after the long-term stability test in alkaline and acidic media (**Supplementary Figs. 35 and 36**). As revealed in **Supplementary Figs. 35a and 36a**, the post-stability XRD spectra of NiCo-SAD-NC were similar to the XRD spectra of the pristine NiCo-SAD-NC, suggesting that crystal structure remained intact without any aggregation of Ni/Co to form nanoparticles. Similarly, the post-stability electron microscopy (SEM, TEM, and HAADF-STEM EDS mapping) of the NiCo-SAD-NC further revealed that the morphology remained intact without any visible aggregates of the Ni/Co, and the Ni and Co atoms remained uniformly distributed (**Supplementary Figs. 35b,c,d and 36b,c,d**). The post-stability EDS spectra confirmed that the Ni/Co composition remained intact after the stability test (**Supplementary Figs. 35e and 36e**). After the stability test, the XPS spectra were well matched with the pristine NiCo-SAD-NC along with well intact metal-N bonds and Ni/Co oxidation states, confirming that the NiCo-SAD structure remained well preserved (**Supplementary Figs. 35f,g,h and 36f,g,h**).

Therefore, combining HAADF-STEM, EELs, XANES/EXAFS, XRD, XPS, and EDX analysis, we conclude that the NiCo-SAD-NC was stable after the stability test in both acidic/alkaline media will well preserved NiCo dimer sites. The text is present on **p. 15 and S34** of the revised manuscript as

Post-stability aberration-corrected HAADF-STEM images with corresponding intensity profiles and EEL spectrum along with nearly matching XANES and EXAFS spectra clearly revealed that the Ni-Co dimer sites in the NiCo-SAD-NC were well-preserved without any sign of metal

agglomeration to form clusters/NPs (Supplementary Fig. 34). In addition, the crystal structure, morphology, and composition of the NiCo-SAD-NC were also maintained after the HER durability test (Supplementary Figs. 35 and 36). After the stability test, the XPS spectra were well matched with the pristine NiCo-SAD-NC along with well intact metal-N bonds. (p. 15)

Supplementary Fig. 34a-d showed the STEM images along with intensity profiles and EEL spectra taken at site A confirmed the existence of NiCo dimer sites after the stability test in both alkaline and acidic media. The overlapping Ni/Co K-edge XANES spectra of NiCo-SAD-NC taken before and after the alkaline/acidic stability test further revealed that the Ni/Co oxidation states and pre-edge features remained intact (Supplementary Fig. 34e,g). Similarly, the Ni/Co K-edge FT-EXAFS spectra confirmed that the Ni/Co-N and Ni-Co bondings in the NiCo-SAD-NC also remained intact after the stability test (Supplementary Fig. 34f,h). (p. S34)

Supplementary Figure 34 | **a,b**, Aberration-corrected HAADF-STEM image with yellow squares showing the dimer sites (**a**), and corresponding intensity profile and EEL spectrum obtained at site A (**b**) for NiCo-SAD-NC after stability test in alkaline media. **c,d**, Aberration-corrected HAADF-STEM image with yellow squares showing the dimer sites (**c**), and corresponding intensity profile and EEL spectrum obtained at site A (**d**) for NiCo-SAD-NC after stability test in acidic media. **e,f,g,h**, experimental Ni K-edge XANES (**e**), Ni K-edge FT-EXAFS (**f**), Co K-edge XANES (**g**), and Co K-edge FT-EXAFS (**h**) spectra of NiCo-SAD-NC after stability test in alkaline and acidic media.

Supplementary Figure 35 |a, b, c, d, and e, XRD pattern, FESEM image, TEM image, HAADF-STEM images and corresponding EDS elemental mapping, and EDS pattern, respectively, of NiCo-SAD-NC after stability test in alkaline media. f, g, and h Fitted deconvoluted N 1s, Ni 2p and Co 2p, respectively, XPS spectra of NiCo-SAD-NC after stability test in alkaline media.

Supplementary Figure 36 |a, b, c, d, and e, XRD pattern, FESEM image, TEM image, HAADF-STEM images and corresponding EDS elemental mapping, and EDS pattern, respectively, of NiCo-SAD-NC after stability test in acidic media. f, g, and h Fitted deconvoluted N 1s, Ni 2p and Co 2p, respectively, XPS spectra of NiCo-SAD-NC after stability test in acidic media.

16) Methods: The experimental X-ray absorption data collection conditions is very limited and far below the standards of any publication reporting such data. The monochromator type, detection mode, sample conditions and data analysis should be indicated.

Answer 16) We apologize for the limited information regarding the XAS and thank the referee for the suggestion. We now provided all the necessary information about the XAS measurements. The text is modified on p. 18 and 19 of the revised manuscript as

The XANES and EXAFS data were collected on BL10C beamline at the Pohang light source (PLS-II) with top-up mode operation under a ring current of 250 mA at 3.0 GeV. The monochromatic X-ray beam could be obtained using liquid-nitrogen cooled Si (111) double crystal monochromator (Bruker ASC) using intense X-ray photons of multipole wiggler source. The X-ray absorption spectroscopic data were recorded for the uniformly dispersed powder samples with a proper thickness on the polyimide film, in fluorescence mode with N₂ gas-filled ionization chamber (IC-SPEC, FMB Oxford) for incident X-ray and passivated implanted planar silicon detector (PIPS, Canberra Co.). Higher-order harmonic contaminations were eliminated by detuning to reduce the incident X-ray intensity by ~30%. Energy calibration has been simultaneously carried out for each measurement with reference metal foils placed in front of the third ion chamber. The data reductions of the experimental spectra to normalized XANES and Fourier-transformed radial distribution function (RDF) were performed through the standard XAFS procedure using IFEFFIT package. Also, Morlet wavelet-transformed EXAFS spectra have been obtained with proper values of η and σ in the equation as follows;

$$\psi(t) = \frac{1}{\sqrt{2\pi\sigma}} (e^{i\eta t} - e^{-\eta^2\sigma^2/2}) e^{-t^2/2\sigma^2}$$

in which the η is the frequency of the oscillation functions and the σ is the half-width.

We truly thank Reviewer 1 for the insightful comments and kind suggestions.

Reviewer: 2

Comments:

Kumar et al. report a metal@polymer strategy to fabricate the diatomic NiCo in N-doped carbon matrix for HER in H₂SO₄ and KOH. In this work, the promising candidates were firstly screened by DFT simulations and then characterized by ex situ XRD, STEM, XPS, and XAS. The overpotential of 54.7/61 mV in acid/base was measured in NiCo-SAD@Ni foam. The synthetic approach is nicely done to precisely tune single/dual atom catalyst and the synergistic effect of the diatomic system was well established. I believe this manuscript can be considered for publication after the following concerns are resolved.

Response: We are grateful for the time and effort Reviewer 2 has spent in reviewing our manuscript. The review comments are constructive for further strengthening the manuscript.

1) In line 109, the d-band center of the diatomic NiCo center (−0.87 eV) is claimed to be nearest to the Fermi level, but NiNi has a lower value of −0.17 eV (Table S1). Is it conflicting? The authors stated that NiNi was inactive and incapable of H₂O adsorption, which is quite the opposite to the experimental observations that Ni NPs or clusters show strong adsorption of H₂O. A more detailed explanation would be appreciated. Similarly, FeFe was also ignored with a low theoretical overpotential of ~0.470 eV comparable to other TM combinations. It is suggested to clearly justify the reasons why those combinations were ruled out for further considerations.

Answer 1) “In line 109, the d-band center of the diatomic NiCo center (−0.87 eV) is claimed to be nearest to the Fermi level, but NiNi has a lower value of −0.17 eV (Table S1). Is it conflicting?”

We are very thankful to the referee for carefully reviewing our manuscript and pleased to clarify the issue. Among the considered single atom dimers (SAD), the d-band center of the NiCo-SAD center (−0.87 eV) was observed to be nearest to the Fermi level compared to the other SADs, excluding the case of NiNi-SAD (Fig. 1a). As revealed in Fig. 1d, there was linear relationship between d-band center and water dissociation kinetic barrier, excluding the case of NiNi-SAD although the d-band center of NiNi-SAD was only -0.17 eV because the NiNi-SAD exhibited the lowest Mulliken charge transfer process in which the Ni-active atom loses 0.80 e⁻, impeding water adsorption ($E_{ads}(*H_2O) = -0.28$ eV) too weak to justify its water adsorption behavior compared to other SADs (Supplementary Fig. 1b). Therefore, we concluded that NiNi-SAD was unfavorable and inactive for water dissociation, so we ignored NiNi-SAD for further examination.

Fig. 1 | High-throughput screening of various transition metal-based SAD (TM-SAD) for HER. **a**, Formation energies corresponding to the charge depletion (Mulliken charge) and d-band center of various TM-SAD-N₆C. **b**, different charge density distribution of TM-SAD-N₆C. The charge depletion and accumulation are denoted by green and red colors, respectively. **c**, Minimum energy paths of water splitting reactions on TM-SAD-N₆C (blue: Co and grey: Ni). **d**, linear correlation between the d-band center and kinetic energy barrier. **e**, Free energy diagram of hydrogen adsorption and **f**, free energy changes of the hydronium and hydroxide desorption step for TM-SAD-N₆C.

Supplementary Figure 1 | a, Average Mulliken (Δq (e⁻)), Lowdin and Bader charges of TM-SAD center. **b**, Calculated correlation between water adsorption energy and charge transfer from metal-active atom in the TM-SAD-N₆C. $\Delta q'$ (e⁻) is computed as the difference of Mulliken charges on the metal atom at the active site when isolated and when on the support (SAD-N₆C), with a positive value indicating electron withdrawal from active-metal atom.

“The authors stated that NiNi was inactive and incapable of H₂O adsorption, which is quite the opposite to the experimental observations that Ni NPs or clusters show strong adsorption of H₂O. A more detailed explanation would be appreciated.”

We agree with the referee that the Ni NPs or clusters show stronger adsorption of H₂O. To clarify the difference in H₂O adsorption of NiNi-SAD and NiNi-NPs, we calculated the crystal orbital Hamilton population (COHP). The integrated COHP (ICOHP) was calculated to examine the interaction between Ni–H₂O bonds quantitatively (see below table). It should be noted that more positive ICOHP value closer to the Fermi level indicates weaker covalent interaction between atoms. As shown in Supplementary Discussion 1, $E_{ads>(*H_2O)}$ increased as ICOHP shifts away from the Fermi level, suggesting that Ni-NPs exhibited stronger adsorption of H₂O molecule compared to NiNi-SAD. The weak Ni–H₂O bond in NiNi-SAD-N₆C could be attributed to fewer electrons in the valance state of Ni bonding with H₂O, and therefore the strength of adsorption with H₂O is weak.

	d_{H_2O-Ni} (Å)	Δq (*H ₂ O)	ICOHP	E_{ads} (*H ₂ O)
H₂O@Ni NPs	2.16	-0.19	-0.34	-1.00
H₂O@NiNi-SAD-N₆C	3.32	-0.07	-0.03	-0.28

Distance between adsorbed H₂O and Ni active site (d_{H_2O-Ni}), calculated adsorption energy (E_{ads}) of H₂O, integrated crystal orbital Hamilton population (ICOHP) and Mulliken charge transfer (Δq) from an active atom into the adsorbed H₂O on NiNi-SAD-N₆C and Ni-NPs.

In fact, among all the considered SADs, the NiNi-SAD exhibited the lowest Mulliken charge transfer process in which the Ni-active atom loses 0.80 e⁻, impeding water adsorption ($E_{ads}(*H_2O) = -0.28$ eV) too weak to justify its water adsorption behavior compared to other SADs (Supplementary Fig. 1b). Therefore, we concluded that NiNi-SAD was unfavorable for water dissociation.

Supplementary Figure 1 | a, Average Mulliken (Δq (e⁻)), Lowdin and Bader charges of TM-SAD center. **b**, Calculated correlation between water adsorption energy and charge transfer from metal-active atom in the TM-SAD-N₆C. $\Delta q'$ (e⁻) is computed as the difference of Mulliken charges on the metal atom at the active site when isolated and when on the support (SAD-N₆C), with a positive value indicating electron withdrawal from active-metal atom.

“Similarly, FeFe was also ignored with a low theoretical overpotential of ~ 0.470 eV comparable to other TM combinations. It is suggested to clearly justify the reasons why those combinations were ruled out for further considerations.”

The overall alkaline HER activity of a catalyst depends on both the water dissociation energy barrier and $*H$ (and $*OH$) desorption energy on the catalyst surface. As revealed in Fig. 1e,f, although the theoretical overpotential for the FeFe-SAD (~ 0.47 eV) was very close to that of NiCo-SAD (~ 0.46 eV), however, the water dissociation kinetic barrier for the FeFe-SAD (2.64eV) was higher than that of NiCo-SAD (2.52eV) (Fig. 1c). In addition, the H^* adsorption energy for the FeFe-SAD (-0.47 eV) was much far from the thermoneutral point of 0 eV compared to that of NiCo-SAD (-0.34 eV) (Fig. 1e). Therefore, driving H_2 generation by water splitting on the FeFe-SAD was more sluggish although they facilitate OH^* desorption step. Hence, by considering the H_2O dissociation and hydrogen evolution process, we expected that NiCo-SAD catalyst was better than of FeFe-SAD.

The text is modified on p. 5, 6 and S5 of the revised manuscript as

In addition, the water adsorption energy exhibited a linear correlation with the Mulliken charge transfer from the metal active site, suggesting that higher charge transfer from the active site exhibits stronger water adsorption strength (Supplementary Fig. 1b). (p. 5)

We further investigated whether there was a correlation between the d-band center and formation energy, in fact, our results indicated that there was a poor linear relationship between them (Fig. 1a), however, 3d-band center of SAD displayed a linear correlation with kinetic barrier of H₂O dissociation (Fig. 1d). Especially, the comprehensive d-band center of Co and Ni atoms in the NiCo-SAD-N₆C (-0.87 eV) was nearest to the Fermi level compared to the other TM-SADs except for NiNi-SAD-N₆C, demonstrated its superior ability to facilitate water dissociation and enhanced proton adsorption, beneficial for HER²⁵. (p. 5, 6)

Supplementary Discussion 1. (p. S5)

	$d_{\text{H}_2\text{O-Ni}}$ (Å)	Δq (*H ₂ O)	ICOHP	E_{ads} (*H ₂ O)
H₂O@Ni NPs	2.16	-0.19	-0.34	-1.00
H₂O@NiNi-SAD-N₆C	3.32	-0.07	-0.03	-0.28

Distance between adsorbed H₂O and Ni active site ($d_{\text{H}_2\text{O-Ni}}$), calculated adsorption energy (E_{ads}) of H₂O, integrated crystal orbital Hamilton population (ICOHP) and Mulliken charge transfer (Δq) from an active atom into the adsorbed H₂O on NiNi-SAD-N₆C and Ni-NPs.

The integrated crystal orbital Hamilton population COHP (ICOHP) was calculated to examine interaction in Ni–H₂O bond quantitatively. It should be noted that more positive ICOHP value closer to the Fermi level indicated weaker covalent interaction between atoms. As shown in Supplementary Discussion 1, $E_{\text{ads}(*\text{H}_2\text{O})}$ increased as ICOHP shifts away from the Fermi level, suggesting that Ni-NPs exhibited stronger adsorption of H₂O molecule compared to NiNi-SAD. The weak Ni–H₂O bond in NiNi-SAD-N₆C could be attributed to fewer electrons in the valance state of Ni bonding with H₂O, and therefore the strength of adsorption with H₂O was weak. (p. S5)

Fig. 1 | High-throughput screening of various transition metal-based SAD (TM-SAD) for HER. **a**, Formation energies corresponding to the charge depletion (Mulliken charge) and d-band center of various TM-SAD-N₆C. **b**, different charge density distribution of TM-SAD-N₆C. The charge depletion and accumulation are denoted by green and red colors, respectively. **c**, Minimum energy paths of water splitting reactions on TM-SAD-N₆C (blue: Co and grey: Ni). **d**, linear correlation between the d-band center and kinetic energy barrier. **e**, Free energy diagram of hydrogen adsorption and **f**, free energy changes of the hydronium and hydroxide desorption step for TM-SAD-N₆C.

2) The fast screening of DFT simulation includes five 3d transition metals, Mn, Fe, Co, Ni, and Cu. It is suggested to provide rationale behind why these elements were selected but not the others, like Cr, Zn, and Mo.

Answer 2) We thank the referee for the suggestion. Following the referee's suggestion, we newly extended our calculation for HER activity with suggested metal (Cr, Mo, and Zn) based SAD, newly provided in **Supplementary Table 2b** of the revised manuscript [note: there was no specific reason for not including these metals before]. We newly created these six SAD (CoCr/CoMo/CoZn/CrCr/MoMo/ZnZn-SAD) and investigated the water dissociation process. As summarized in **Supplementary Table 2b**, all five dimers showed favorable H₂O adsorption with negative $E_{ads}(*H_2O)$ values (exception for CoMo-SAD-N₆C) and rather easy H₂O dissociation with a relatively low kinetic barrier. Remarkably, CrCr-SAD-N₆C proved to be one of the most potential candidates with the lowest H₂O adsorption (-1.388 eV) and kinetic energy barrier for H₂O dissociation (2.480 eV). However, its theoretical overpotential was very high due to the higher free

energy of OH* desorption (1.06 eV) compared to NiCo-SAD (0.46 eV). Therefore, NiCo-SAD-N₆C is still the most promising potential candidate for alkaline-HER.

The text is modified on p. 6 and S6 of the revised manuscript as

In addition, we also extended our calculation for alkaline HER activity with other metals (Cr, Mo, and Zn) based SAD, summarized in Supplementary Table 2b. As revealed in Supplementary Table 2b, the NiCo-SAD-N₆C still exhibited the best alkaline HER activity compared to other SAD based on the evaluated kinetic barrier of water dissociation and free energy of *H and *OH desorption. (p.6)

As summarized in Supplementary Table 2b, all five dimers showed favorable H₂O adsorption with negative $E_{ads>(*H_2O)}$ values (exception for CoMo-SAD-N₆C) and rather easy H₂O dissociation with a relatively low kinetic barrier. Remarkably, CrCr-SAD-N₆C proved to be one of the most potential candidates with the lowest H₂O adsorption (-1.388 eV) and kinetic energy barrier for H₂O dissociation (2.480 eV). However, its theoretical overpotential was very high due to the higher free energy of OH* desorption (1.06 eV) compared to NiCo-SAD (0.46 eV). Therefore, NiCo-SAD-N₆C is still the most promising potential candidate for alkaline-HER. (p. S6)

Supplementary Table 2. b. The water adsorption energy and kinetic barrier of water splitting on six TM-SAD-N₆C (CoCr, CoMo, CoZn, CrCr, MoMo, and ZnZn) catalysts and free energy of *H and *OH.

b

	$E_{ads>(*H_2O)}$ (eV)	Kinetic barrier (eV)	$ \Delta G^{*OH} $	$ \Delta G^{*H} $
CoCr	-0.515	3.854	0.801	0.240
CoMo	-0.256	inactive	--	--
CoZn	-0.532	2.832	0.649	0.057
CrCr	-1.388	2.480	1.058	0.252
MoMo	-0.515	3.100	0.102	0.444
ZnZn	-0.464	2.827	0.104	0.249

3) What do the Bader charge and charge density distribution reflect (line 101 to 106)? Is the discussion related to the intrinsic property or the HER activity of materials?

Answer 3) We thank the referee for the comment, and we are pleased to clarify the issue. We apologize for the unclear explanation regarding the Bader charge and the charge density difference. The Bader charge and charge density difference just provides the information for the charge depletion and accumulation region on the catalysts. It does not provide any information regarding the intrinsic HER activity of the catalysts. Now, we newly added more information regarding the analysis of the atomic charge by two more methods such as Mulliken and Lowdin along with the Bader charge analysis (Supplementary Fig. 1a and Fig. 1a). By the thorough analysis to carefully evaluate the charge values in these large varieties of catalysis, we found that the Bader charge analysis was useless as this method does not show a similar trend compared to the Mulliken and Lowdin method (Supplementary Fig. 1a). The Mulliken and lowdin charges followed a similar trend and are recommended because they provide meaningful chemical bonding explanations for these SAD catalysts. Hence, we now replaced the Bader charge analysis with the Mulliken charge analysis.

The text is modified on p. 5 of the revised manuscript as

Interestingly, we found that there was a consistent trend for the average Mulliken charges distribution (Δq) of TM-SAD center with the formation energy of TM-SAD-N₆G structures, except for the CoCu/CuCu-SAD-N₆C (Fig 1a and Supplementary Fig. 1a). The formation energy increased with the increase of Δq in the homo/heterostructures of SAD, suggesting that higher Δq confirmed the higher thermodynamically stable SAD structure²⁴.

Supplementary Figure 1 | a, Average Mulliken (Δq (e⁻)), Lowdin and Bader charges of TM-SAD center. **b**, Calculated correlation between water adsorption energy and charge transfer from metal-active atom in the TM-SAD-N₆C. $\Delta q'$ (e⁻) is computed as the difference of

Mulliken charges on the metal atom at active site when isolated and when on the support (SAD-N₆C), with a positive value indicating electron withdrawal from active-metal atom.

Fig. 1 | High-throughput screening of various transition metal-based SAD (TM-SAD) for HER. **a**, Formation energies corresponding to the charge depletion (Mulliken charge) and d-band center of various TM-SAD-N₆C. **b**, different charge density distribution of TM-SAD-N₆C. The charge depletion and accumulation are denoted by green and red colors, respectively. **c**, Minimum energy paths of water splitting reactions on TM-SAD-N₆C (blue: Co and grey: Ni). **d**, linear correlation between the d-band center and kinetic energy barrier. **e**, Free energy diagram of hydrogen adsorption and **f**, free energy changes of the hydronium and hydroxide desorption step for TM-SAD-N₆C.

4) To prepare NiCo dimer site rather than alloy, dicyandiamide was added into NiCo@polydopamine precursor. How does the dicyandiamide affect the final composition since metal ions are already coordinated to polydopamine? The authors should give a more detailed explanation.

Answer 4) We sincerely thank the referee for carefully reviewing our manuscript and we are pleased to clarify the issue. As shown in **scheme 1** of our original manuscript, when the Ni²⁺-Co²⁺@polydopamine precursor was annealed in the absence of dicyandiamide (N containing organic molecule), the Ni and Co ions aggregated to form NiCo alloys encapsulated in the carbon support (NiCo-NP-NC), suggesting that the content of N moieties in the Ni²⁺-Co²⁺@polydopamine were insufficient to trap the Ni/Co metal atoms. The final content of total metal was 7.55 wt% with

only 0.95 at% of N in the NiCo-NP-NC confirmed by EDAX and ICP-AES (Supplementary Table 3, 4). Whereas, after introducing dicyandiamide (N containing organic molecule) for trapping the dual-metal sites with N coordination, the content of N in the NiCo-SAD-NC increased to 17.41 at% while the total metal loading was slightly reduced to 7.28 wt% (Supplementary Table 3, 4). Thus, due to the introduction of additional N using dicyandiamide to trap the metal atoms, the total metal loading was slightly reduced while the N content increased, slightly affecting the final composition. The text is modified on p. 8, and S10 of the revised manuscript as

Energy-dispersive X-ray spectroscopy (EDS) and inductively coupled plasma atomic emission spectrometry (ICP-AES) analysis of NiCo-NP-NC revealed a 1:1 ratio of Ni to Co with total metal loading of 7.55 wt.%, while the N content of less than 1 at.% indicated that the amount of N moieties in the Ni²⁺-Co²⁺@polydopamine sphere were insufficient to trap the Ni/Co metal atoms (Supplementary Fig. 6a and Supplementary Table 3, 4). (p. 8)

The NiCo-SAD-NC also exhibited a 1:1 ratio (Ni:Co) with total metal loading of 7.28 wt.%, while around 17.5 at.% of N content was the optimum amount to trap the Ni-Co atomic sites (Supplementary Fig. 6b and Supplementary Tables 3, 4). (p. 8)

Due to the introduction of additional N using dicyandiamide to trap the metal atoms, the total metal loading was slightly reduced while the N content increased for the NiCo-SAD-NC, slightly tuning the final composition. (p. S10)

Scheme 1 | Synthesis strategy for single atom dimer. Schematic illustration of the synthetic strategy for the NiCo-SAD-NC and NiCo-NP-NC: I) trapping metal ions (Ni and Co) into the

polydopamine (PDA) via self-polymerization of dopamine; II) generation of NiCo-NP-NC and NiCo-SAD-NC through calcination treatment upon modulating nitrogen containing precursor (dicyandiamide) amount.

Supplementary Table 3. Atomic % of elements and Ni : Co ratio obtained from EDS and XPS analysis.

Sample	EDS			Ni : Co (EDS)		XPS			Ni : Co (XPS)	
	Ni (at%)	Co (at%)	N (at%)	Expt.	Obser.	Ni (at%)	Co (at%)	N (at%)	Expt.	Obser.
NiCo-SAD-NC	0.79	0.88	17.41	1	0.9	0.55	0.529	18.1	1	1.03
NiCo-NP-NC	0.84	0.82	0.95	1	1.02	0.87	0.89	0.93	1	0.98
Ni-SA-NC	1.59	-	18.63	-	-	1.78	-	17.09	-	-
Co-SA-NC	-	1.64	16.68	-	-	-	1.57	16.96	-	-
NiCo-SAD-NC (1:2)	0.58	1.25	14.67	0.5	0.46					
NiCo-SAD-NC (2:1)	1.28	0.6	17.3	2	2.1					
NC	-	-	15.9	-	-	-	-	17.52	-	-

Supplementary Table 4. Weight % of metal loading obtained from EDS and ICP-AES analysis.

Sample	EDS	ICP-AES
--------	-----	---------

	Ni (wt%)	Co (wt%)	Pt (wt%)	Ni + Co (wt%)	Ni (wt%)	Co (wt%)	Ni + Co (wt%)	Average (wt%) from EDS and ICP-AES
NiCo-SAD-NC	3.94	3.56	-	7.5	3.719	3.331	7.05	7.28
NiCo-NP-NC	4.18	4.09	-	8.27	3.675	3.558	6.83	7.55
Pt-SA	-	-	4.5	-	-	-	-	-

5) With the same synthetic procedure, the samples should have the similar coordination. Why does only NiCo dimer form in SAD while the counterparts are SA? Is it a mixed state of isolated Ni, Co centers, and NiCo centers in NiCo-SAD? Is it a mixed state of Ni and NiNi centers in Ni-SA?

Answer 5) We sincerely thank the referee for raising this issue and we are pleased to clarify the issue. Generally, to trap the isolated single metal atomic site, higher electronegative N atoms were introduced which can strongly coordinate and stabilize the metal site in the carbon support [*Nat Commun.* **11**, 1576 (2020)]. While, for generating the single atomic dimer sites with M_1 - M_2 bond (M_1 , M_2 : metals), the bonding dimer needs favorable electron-negativity and enthalpy for M_1 - M_2 dimer and alloys synthesis, along with optimum N for trapping the dimer site [*Adv. Mater.* **33**, 2003327 (2021)]. In the case of NiCo-SAD-NC, since the electronegativities of Ni and Co atoms are different, the formation of Ni-Co bonding dimer stabilized by N coordination is preferred over Ni-Ni or Co-Co bonding because the electronegativity difference between Ni-Ni and Co-Co is unfavorable. Also, because of the favorable electronegativity difference between Ni and Co, the NiCo alloy NPs were formed in the absence of dicyandiamide (N containing organic molecule) rather than separate Ni or Co NPs in the NiCo-NP-NC sample, additionally corroborating the Ni-Co bonding preference over Ni-Ni or Co-Co. As revealed in the newly collected improved aberration-corrected HAADF-STEM image in Fig. 2b, the NiCo-SAD-NC contained multiple isolated Ni-Co dimer sites (marked by yellow square) with coordination between Ni and Co atoms at the atomic level along with few isolated Ni or Co single atom centers coordinated to electronegative N (marked by the orange circle). Although few Ni and Co isolated atoms also existed in the NiCo-SAD-NC, they were not the active site for HER, confirmed by DFT and this result do not affect the conclusion of our work.

In the case of Ni-SA-NC, since there is no electronegativity difference between Ni and Ni, the Ni-Ni dimer bonding is unfavorable. Therefore, in the case Ni-SA-NC, mostly isolated Ni centers were present. Supplementary Fig. 17 in the revised manuscript clearly showed the presence of isolated Ni atoms in the Ni-SA-NC. The text is modified on p. 9 of the revised manuscript as

Aberration-corrected HAADF-STEM image in Fig. 2b clearly demonstrated the existence of isolated Ni-Co dimer sites (marked by yellow square) with coordination between Ni and Co at atomic level along with some isolated Ni or Co atoms (marked by the orange circle).

Fig. 2 | Structural analysis and electron microscopy of NiCo-SAD-NC. **a**, XRD spectra of NC, NiCo-NP-NC, and NiCo-SAD-NC. **b**, Aberration-corrected HAADF-STEM image of the NiCo-SAD-NC. The yellow squares in (b) shows the dimer sites and some of single Ni/Co-atom sites are highlighted by orange circles. **c,d**, The intensity profile and corresponding EEL spectra obtained at site A (c) and site B (d) for NiCo-SAD-NC, indicate Ni and Co are coordinated at the atomic scale. **e**, Statistical Ni-Co distance in the observed dimers. **f**, HAADF-STEM image and corresponding EDS maps of NiCo-SAD-NC showing the uniform dispersion of N (green), Co (red) and Ni (yellow).

Supplementary Figure 17 | **a**, and **b**, Aberration-corrected HAADF-STEM image of the Ni-SA-NC. The white circles in (**b**) shows the isolated Ni-SA uniformly dispersed. **c**, The intensity profile obtained at Ni-SA site 1, 2, and 3.

6) According to the XPS and XANES features, the authors demonstrated the electron transfer between the NiCo interface. The essential reasons behind the interplay should be discussed.

Answer 6) We sincerely thank the referee for the comment, and we are pleased to clarify the issue. In many previous studies, it has been reported that the intimate contact at the interface junction can result in strong electronic interactions and electron transfer at the junction, which usually provided solid proof for the generation or formation of the interface. XPS measurements have been utilized in previous studies to show the electron transfer between different components at the interface [ACS Energy Lett. **6**, 354–363 (2021); ACS Energy Lett. **2**, 2257–2263 (2017); Adv. Mater. **31**, 1901174 (2019); Small **16**, 2001642 (2020)]. The high-resolution XPS and XANES K-edge spectra of Ni and Co for NiCo-SAD-NC and Ni/Co-SA-NC (Supplementary Fig. 14 and 16) clearly showed the strong electronic interaction and electron transfer between the Ni and Co, confirming the formation of the atomic interface in NiCo-SAD-NC. Compared to the difference in the electronegativity between Ni and N in Ni-SA-NC, due to the higher difference in the electronegativity between Co and N in the Co-SA-NC, the Co atom donates more electrons to the coordinated N, as a result, the Co oxidation state in Co-SA-NC was higher than the Ni in Ni-SA-NC, revealed from their respective XPS and XANES spectra. In the case of NiCo-SAD-NC, when the Co at higher oxidation state underwent strong electronic interaction with the Ni at a relatively lower oxidation state at the interface, then the electron transfer occurred from the Ni to Co site and the final oxidation state of Ni in NiCo-SAD-NC became higher than the Ni in Ni-SA-NC.

Contrarily, the oxidation state of Co in NiCo-SAD-NC became lower than the Co in Co-SA-NC. Therefore, the electron transfer between the Ni and Co suggested the successful generation of the atomic interface in NiCo-SAD-NC. The text is modified on p. S18 of the revised manuscript as

Compared to the difference in the electronegativity between Ni and N in Ni-SA-NC, due to the larger difference in the electronegativity between Co and N in the Co-SA-NC, the Co oxidation state in Co-SA-NC was higher than the Ni in Ni-SA-NC, revealed from their respective XPS and XANES spectra. In the case of NiCo-SAD-NC, when Co at a higher oxidation state underwent strong electronic interaction with the Ni at a relatively lower oxidation state at the interface, then the electron transfer occurred from the Ni to Co site and the final oxidation state of Ni in NiCo-SAD-NC became higher than the Ni in Ni-SA-NC. Contrarily, the oxidation state of Co in NiCo-SAD-NC became lower than the Co in Co-SA-NC. Therefore, the electron transfer between the Ni and Co suggested the successful generation of the atomic interface in NiCo-SAD-NC.

Supplementary Figure 14 | a, and b, Fitted deconvoluted Ni $2p_{3/2}$ XPS spectra and the Ni oxidation states analyzed by the XPS peak position, respectively, of NiCo-SAD-NC, and Ni-SA-NC with reference samples. c, and d, Fitted deconvoluted Co $2p_{3/2}$ XPS spectra and the Co oxidation states analyzed by the XPS peak position, respectively, of NiCo-SAD-NC, and Co-SA-NC with reference samples.

Supplementary Figure 16 | **a**, and **b**, Experimental Ni K-edge XANES spectra of NiCo-SAD-NC with reference samples and the Ni oxidation state analysis by corresponding XANES energy at half-edge jump, respectively. **c**, and **d**, Experimental Co K-edge XANES spectra of NiCo-SAD-NC with reference samples and the Co oxidation state analysis by corresponding XANES energy at half-edge jump, respectively.

7) What do the peaks at $\sim 1.3 \text{ \AA}$ (pink color) represent in the EXAFS fitting results in Figure S14? There is a difference of 0.22 \AA between Ni-Co and Co-Ni scattering. What is the reason for this difference?

Answer 7) We sincerely thank the referee for carefully reviewing our manuscript, and we are pleased to clarify the issue. As revealed in the newly added improved aberration-corrected HAADF-STEM image (Fig. 2b), the NiCo-SAD-NC contains multiple isolated Ni-Co dimer sites (marked by yellow square) with coordination between Ni and Co atoms at the atomic level along with few isolated Ni or Co single atom centers coordinated to electronegative N (marked by the orange circle). Therefore, the deconvoluted peaks at $\sim 1.3 \text{ \AA}$ in the Ni and Co K-edge FT-EXAFS peaks deconvolution of NiCo-SAD-NC referred to the Ni/Co-N coordination originating from the isolated Ni/Co-SA sites in the previous Supplementary Fig. 14 of our original manuscript.

However, we now replaced the previous FT-EXAFS peaks deconvolution (previous Supplementary Fig. 14) with proper EXAFS fitting at the Ni/Co K-edge for determining the structural information of NiCo-SAD-NC such as Ni/Co-N and Ni-Co coordination numbers (CNs), newly provided in Supplementary Fig. 19 and Supplementary Table S5 of the revised manuscript. The Ni/Co K-edge EXAFS fitting results clearly revealed that the CNs of Ni-N and Co-N were 3.3 and 3.1, respectively, suggesting that the NiCo-SAD-NC was dominated by the Ni-N₃ and Co-N₃ environment. Meanwhile, the Ni-Co and Co-Ni paths were fitted with CNs of 0.7 and 0.6, respectively, directly indicating the existence of Ni-Co bonding dimer in the form of NiCo-N₆ structure, consistent with the DFT calculated structure. The obtained CNs for Ni-N/Co-N and Ni-Co/Co-Ni were consistent with the previous Co-Fe and Co-Zn dual-atom pair catalysts [*Angew. Chem. Int. Ed.* **131**, 2648–2652 (2019); *Energy Environ. Sci.*, **11**, 3375—3379 (2018)].

We agree with the referee that there was a difference of ~0.2 Å between Ni-Co and Co-Ni scattering observed from their respective FT-EXAFS spectra. Such differences of around 0.2 Å between the M₁-M₂ and M₂-M₁ scattering have also been previously observed in Fe-NiNC-50 [*Nano Energy* **71**, 104597 (2020)] and Pt-Ru dimer [*Nat Commun* **10**, 4936 (2019)] dual single atoms catalysts. The possible reason behind such small deviation is that since the k dependence of the backscattering amplitude in these heavy atoms exhibits oscillations in k space, the Fourier transform (FT) of the EXAFS is no longer a simple symmetric peak as in low z scattering atoms like C or N where the k dependence is monotonic [*Nat Commun* **10**, 4936 (2019)]. As the FT of the EXAFS is no longer a simple symmetric peak, therefore, the Ni-Co distance observed from the Ni K-edge side is slightly different from the Co-Ni distance observed from the Co K-edge side.

The text is modified on p. 12 of the revised manuscript as

Furthermore, the structural information of NiCo-SAD-NC along with the Ni/Co-N and Ni-Co coordination number (CN) was also affirmed by the EXAFS fitting results in R space oscillations (Supplementary Fig. 19 and Supplementary Table 5). The EXAFS fitting results clearly revealed that the CNs of Ni-N and Co-N were 3.3 and 3.1, respectively, suggesting that the NiCo-SAD-NC was dominated by the Ni-N₃ and Co-N₃ environment. Meanwhile, the Ni-Co/Co-Ni paths were fitted at a position of 2.55 Å with a CN of 0.7/0.6, directly indicating the existence of Ni-Co bonding dimer in the form of NiCo-N₆ structure, consistent with the DFT calculation structure.

Previous Supplementary Figure 14 | The FT-EXAFS peaks deconvolution of NiCo-SAD-NC at **a** Ni K-edge and **b** Co K-edge (**Previous Supplementary Figure 14** in our original manuscript for reference to the refereed).

Supplementary Figure 19 | **a**, and **b**, The Ni K-edge, and Co K-edge FT-EXAFS, respectively, of NiCo-SAD-NC and Ni/Co metal with corresponding EXAFS fitting curves. **c**, The DFT simulated NiCo-SAD-NC model based on the FT-EXAFS fitting value.

Supplementary Table 5. EXAFS fitting parameters at the Ni and Co K-edge for NiCo-SAD-NC and Ni/Co metal ($S_0^2 = 0.780$ (Ni), 0.816 (Co)).

Sample	Shell	N^a	$R(\text{\AA})^b$	$\sigma^2(\text{\AA}^2)^c$	$\Delta E_0(\text{eV})^d$	R factor
Co K-edge						
Co Metal	Co-Co	12	2.49	0.0060	6.6	0.0002
NiCo-SAD-NC	Co-N	3.1	2.01	0.0018	-4.5	0.0001
	Co-Ni	0.6	2.55	0.0126		
Ni K-edge						
Ni Metal	Ni-Ni	12	2.48	0.0059	6.3	0.0001
NiCo-SAD-NC	Ni-N	3.3	1.86	0.0039	-4.8	0.0016
	Ni-Co	0.7	2.55	0.0121		

^a N : coordination numbers; ^b R : bond distance; ^c σ^2 : Debye-Waller factors; ^d ΔE_0 : the inner potential correction. R factor: goodness of fit. S_0^2 was set to 0.816 for Co and 0.780 for Ni, according to the experimental EXAFS fit of Co and Ni metal reference by fixing CN as the known crystallographic value.

8) Ag/AgCl (3 M KCl) electrode is only applicable in neutral medium. There is a consensus for electrochemist to use the suitable reference electrode (Hg/Hg₂SO₄ and Hg/HgO in acidic and basic media, respectively), especially for the long-term test. A suitable reference electrode should be used for each pH range.

Answer 8) Following the referee's suggestion, we newly measured the HER LSV polarization curve and long-term stability of NiCo-SAD-NC in alkaline and acidic media using Hg/HgO, and Hg/Hg₂SO₄ as reference electrodes, respectively, newly provided in Supplementary Fig. 29 of the revised manuscript. As revealed in Supplementary Fig. 29a-d, the HER activity of NiCo-SAD-NC was still superior to NiCo-NP-NC and comparable to that of 20% Pt/C in alkaline and acidic media using Hg/HgO and Hg/Hg₂SO₄ as reference electrodes, respectively, certifying the superior intrinsic activity of NiCo-SAD-NC. In addition, the NiCo-SAD-NC also demonstrated a superior long-term stability for 25 h at -60 mA/cm^2 under both alkaline and acidic media using Hg/HgO and Hg/Hg₂SO₄ as reference electrodes, respectively, without any noticeable degradation (Supplementary Fig. 29e). The overlapping LSV polarization curves before and after the durability

test confirmed the retention of the active sites and superior HER activity (Supplementary Fig. 29e, insets). The text is modified on p. 14, S30, and S31 of the revised manuscript as

Additionally, the superior intrinsic HER performance of NiCo-SAD-NC was also maintained under both alkaline and acidic media using Hg/HgO and Hg/Hg₂SO₄ as reference electrodes, respectively (Supplementary Fig. 29). (p. 14)

As revealed in Supplementary Fig. 29a-d, the HER activity of NiCo-SAD-NC was still superior to NiCo-NP-NC and comparable to that of 20% Pt/C in alkaline and acidic media using Hg/HgO and Hg/Hg₂SO₄ as reference electrodes, respectively, certifying the superior intrinsic activity of NiCo-SAD-NC. In addition, the NiCo-SAD-NC also demonstrated a superior long-term stability for 25 h at -60 mA/cm² under both alkaline and acidic media using Hg/HgO and Hg/Hg₂SO₄ as reference electrodes, respectively, without any noticeable degradation (Supplementary Fig. 29e). The overlapping LSV polarization curves before and after the durability test confirmed the retention of the active sites and superior HER activity (Supplementary Fig. 29e, insets). (p. S30, S31)

Supplementary Figure 29 | **a**, and **b**, HER LSV polarization curve and the overpotentials required to reach -10 mA cm^{-2} in 1 M KOH using Hg/HgO as reference electrode. **c**, and **d**, HER LSV polarization and the overpotentials required to reach -10 mA cm^{-2} in 0.5 M H₂SO₄ using Hg/Hg₂SO₄ as reference electrode. **e**, long-term chronopotentiometric stability test for NiCo-SAD-NC in 0.5 M H₂SO₄ (ref. electrode: Hg/Hg₂SO₄) and 1 M KOH (ref. electrode: Hg/HgO) at a current density of -60 mA cm^{-2} . The inset shows the LSV curves before and after the durability test.

9) To claim “all-pH” or “pH-universal”, the performance test in a neutral solution such as PBS is necessary. Otherwise, it is better to change the conclusion in a weaker tone.

Answer 9) We sincerely thank the referee for the valuable suggestion. Following the referee’s suggestion, we newly measured the HER activity of NiCo-SAD-NC in phosphate buffer solution (PBS) (pH 7.4) and compared the HER activity with Pt/C and NiCo-NP-NC, newly provided in **Supplementary Fig. 30** of the revised manuscript. Even in the PBS solution, the HER activity of the NiCo-SAD-NC was superior to that of NiCo-NP-NC and comparable to that of benchmarking Pt/C, certifying the pH-universal HER activity of the NiCo-SAD-NC. The text is modified on **p. 14 and 15** of the revised manuscript as

Furthermore, to claim the pH-universality, we also tested the HER performance of NiCo-SAD-NC in phosphate buffer solution (PBS) (pH: 7.4) and compared the HER activity with Pt/C and NiCo-NP-NC (Supplementary Fig. 30). Even in the PBS electrolyte, the NiCo-SAD-NC outperformed the NiCo-NP-NC comparable with the benchmarking Pt/C. (p. 14, 15)

Supplementary Figure 30| a, and b, HER LSV polarization curve and the overpotentials required to reach -10 mA cm⁻², respectively, in PBS solution (pH = 7.4).

10) The stability tests of SAD catalyst were compared by chronopotentiometric method. It shows negligible degradation in acid while ~80 mV decline in base. Could the authors explain the different behaviors? Also, the LSVs before and after the reaction should be provided.

Answer 10) We sincerely thank the referee for carefully reviewing our manuscript, and we are pleased to clarify the issue. According to the bar plots in Fig. 4b,e, the overpotentials required at -100 mA/cm² for NiCo-SAD-NC were ~190 and ~117 mV in alkaline and acidic media, respectively. After the durability test for NiCo-SAD-NC (Fig. 4g), the final overpotentials were 215.3 and 125.5 mV at -100 mA/cm² in alkaline and acidic media, respectively, suggesting that only around 25 and 8 mV decline in the overpotential in alkaline and acidic media, respectively. Such small degradation generally happened owing to the blocking of the active sites by the hydrogen gas bubbles or small leaching of the catalysts from the electrode surface due to vigorous H₂ bubble generation at high current density [*J. Electrochem. Soc.* **166**, F458–F464 (2019)]. Although in Fig. 4g, the stability test in alkaline media started at around 150 mV, within 2 h of operation the overpotential becomes ~190 mV, which was expected at a current density of -100 mA/cm² and after the stability test, the final overpotential was 215.3 mV.

Following the referee's suggestion, we newly provided the LSV polarization curve before and after the durability test in alkaline and acidic media in Supplementary Fig. 33 of the revised manuscript. The overlapping LSV curve taken before and after the durability test further confirmed the retention of active sites along with the high catalytic activity. The text is modified on p. 15 of the revised manuscript as

The overlapping LSV curve taken before and after the durability test further asserted the retention of active sites along with high catalytic activity (Supplementary Fig. 33).

Fig. 4 | Electrochemical hydrogen generation performance in alkaline and acidic media. a,b,c, HER LSV polarization curves (iR-corrected) (a), the overpotentials required to reach -10 and -100 mA cm^{-2} (b), and corresponding Tafel plots (c), in 1 M KOH . **d,e,f,** HER LSV polarization curves (iR-corrected) (d), the overpotentials required to reach -10 and -100 mA cm^{-2} (e), and corresponding Tafel plots (f), in $0.5 \text{ M H}_2\text{SO}_4$. **g,** Chronopotentiometric stability test for NiCo-SAD-NC and NiCo-NP-NC in $0.5 \text{ M H}_2\text{SO}_4$ and 1 M KOH at a current density of -100 mA cm^{-2} . The inset shows the faradaic efficiency of NiCo-SAD-NC for HER at -100 mA cm^{-2} in $0.5 \text{ M H}_2\text{SO}_4$ (right) and 1 M KOH (left).

Supplementary Figure 33 | a, and b, HER LSV polarization curve before and after the stability test in alkaline and acidic media, respectively.

11) Ambiguous discussion and errors:

(a) In the abstract, the authors state that the water-dissociation is facilitated, thus accelerate the HER kinetics in acid/alkaline. However, water dissociation does not take place in acid HER. It is better to make the statement more precise.

(b) There are two different rows of d-band center in Table S2, which looks confusing to me. Please correct or define them.

(c) The overpotentials measured on CFP should be labeled in Figure S25. It looks like ~70/320 mV in acid and ~90/350 mV in base at 10/100 mA cm⁻².

Answer 11)

(a) We apologize for the mistake. Following the referee's suggestion, we corrected the statement in the abstract to make it more appropriate for pH-universal HER. The text is modified on p. 2 of the revised manuscript as

Herein, systematic first-principle screening revealed that the synergistic interaction at the NiCo-SAD atomic-interface can significantly upshift the d-band center, thereby, facilitate rapid water-dissociation and optimal proton adsorption, accelerating alkaline/acidic HER kinetics.

(b) We apologize for the mistake. We now corrected the caption of Supplementary Table 2.

Supplementary Table 2. a, D-band center of the TM-SAD-N₆C systems with H* and H₂O* corresponding to the Gibbs adsorption free energy of the H* and energy barrier of water dissociation, respectively.

a

	D-band center of SAD systems with H* (eV)	ΔG_{H^*} (eV)	D-band center of SAD systems with H ₂ O* (eV)	E_{split} (eV)
CoCu	-2.16	0.126	-1.67	3.33
NiCo	-1.28	-0.34	-0.87	2.52
CoFe	-0.88	-0.121	-1.07	3.11
CoMn	-0.78	-0.46	-0.97	2.55
CuCu	-2.30	0.783	-2.82	4.22
CoCo	-0.86	-0.355	-1.50	2.94
FeFe	-0.90	-0.47	-1.05	2.64
MnMn	-0.16	-0.135	-0.89	2.76

(c) Following the referee's suggestion, we newly provided the HER overpotentials at 10/100 mA/cm² measured on the CFP in **Supplementary Fig. 31b,d** of the revised manuscript. The NiCo-SAD-NC precisely required 75.4/371 mV overpotential in alkaline media and 61.2/338 mV overpotential in acidic media at 10/100 mA/cm² which is still superior to NiCo-NP-NC and comparable to Pt/C.

Supplementary Figure 31 | **a**, and **b**, HER LSV polarization curve and the overpotentials required to reach -10 mA cm^{-2} , respectively, on carbon fiber paper (CFP) current collector in 1 M KOH. **c**, and **d**, HER LSV polarization curve and the overpotentials required to reach -10 mA cm^{-2} , respectively, on CFP current collector in 0.5 M H₂SO₄.

We truly thank the Reviewer 2 for the insightful comments and kind suggestions.

Reviewer: 3

Comments:

The electrocatalytic hydrogen evolution reaction through the splitting of water has drawn great interests these years. This paper reports the formation of bimetallic single-atom-dimer catalysts for HER in both acidic and alkaline media. The authors have delivered detailed characterizations and thorough analysis about those catalysts. The whole work is remarkable. However, I think the total novelty of this work is not enough to meet the high standards of the Nature Communications before revised.

Response: We are grateful for the time and effort Reviewer 3 has spent in reviewing our manuscript. The review comments are constructive for further strengthening the manuscript.

1) Since the pioneering work by Tao Zhang on single-atom, a bulk of works has been reported to construct different single-atom systems. The synthesise process of this work is not novelty. A wide-pH-range HER catalyst or an overpotential lower than 50 mV could also be easily achieved, even in a none-noble-metal system (10.1002/adma.201807780, 10.1038/s41467-018-03358-x, 10.1002/anie.202011358). The wide-pH-range HER catalyzed by single-atom could also reported many times (10.1016/j.nanoen.2019.02.062, 10.1038/s41467-020-14848-2, 10.1021/acssuschemeng.0c04322). Thus, this paper is lack of novelty to meet the high standards of Nature Communications.

Answer 1) We thank the referee's comments on the novelty of our work, and we are pleased to clarify this issue. We agree that quite a number of nanomaterials/single atom-based HER catalysts were being published in the last few years. However, evaluating the impact/novelty of work should reply on what it has done to solve the remaining scientific or engineering challenges/questions of the studied materials or reactions, not only on the number of published papers. As we have mentioned in our introduction, the single-atom catalysts (SACs) have shown captivating activity (especially mass activity) over regular nanomaterial-based counterparts for HER due to their near 100% atom economy. However, the main issue related to SACs is that the isolated single metal sites coordinated with the neighboring nitrogen atoms in the carbon matrix (M-NC) are only capable of catalyzing simple elementary reactions, mainly due to the simplicity of the single-atom center. Because of the simplicity of the M-NC single-atom center, the possibilities for further modification of the active site in SACs are extremely limited, hindering their wide range of applications. Therefore, simple M-NC SACs are insufficient for catalyzing HER in both alkaline/acidic media, since the pH-universal HER is a complex process involving additional water dissociation steps along with proton adsorption and dimerization to molecular H₂. Aiming at this issue, our approach of introducing secondary metal atoms to form a single atom dimer (SAD) atomic interface can further modulate the coordination environment and electronic structure of the SACs, boosting the intrinsic HER kinetics beyond SACs. Additionally, the synthesis and identification of SAD are extremely challenging. Our DFT calculations indicated that the

conventional Ni and Co SACs (Ni-SA-NC and Co-SA-NC) were extremely sluggish for water dissociation and exhibited poor performance for pH-universal HER. However, after successfully generating NiCo-SAD atomic interface via our synthetic approach, we found that the electronic d-band center of the SAD was significantly upshifted nearest to the fermi level compared to Ni/Co-SA-NC, which facilitated rapid water dissociation and optimal proton adsorption, accelerating pH-universal HER kinetics. Our NiCo-SAD exhibited comparable pH-universal HER activity to that of state-of-the-art Pt/C and much superior to that of conventional Ni-SA-NC, Co-SA-NC, and NiCo-NPs. The HER mass activity of the NiCo-SAD in acidic/alkaline media was much superior to that Pt/C and the NiCo-NPs counterpart. Thus, our idea or approach to obtain SAD atomic interface for modulating the electronic structure beyond SACs for achieving pH universal HER catalysis is conceptually novel and can also be extended to the other fields of materials science and electrocatalysis.

For the reference papers mentioned by the referee [i-10.1002/adma.201807780: *Pt-Ni nanowire array for alkaline HER*, ii-10.1038/s41467-018-03358-x: *MnO_x catalysts for aerobic ammoxidation (not HER)*, iii-10.1002/anie.202011358: *single-crystal Mo₂C hexagonal nanosheet on carbon sheet for acidic HER*]. First, the materials are different from ours, and the scientific questions faced by each material are also different. Moreover, the i and iii reference talk about the nanomaterials-based catalysts for HER rather than SACs, suggesting that the targeted research motivations and scientific questions are different. Secondly, although the reported HER overpotentials are below 50 mV, the major drawback in such nanomaterials-based systems is that the mass activity is very poor as only the surface atoms take part in the catalysis and the core atoms remain dormant, significantly affecting the atom economy and not suitable from practical applications. In our case, the atomically isolated NiCo-SAD made solely of earth-abundant elements exhibited comparable pH-universal activity to that of commercial Pt and much superior mass activity to that Pt/C and the NiCo-NPs counterpart.

For the reference papers mentioned by the referee [i-10.1016/j.nanoen.2019.02.062: *metal clusters (Ru, Pt, Pd etc.) combining single cobalt atoms anchored on N-doped carbon (Ru/Pt/Pd@Co-SAs/N-C) for all-pH HER*, ii-10.1038/s41467-020-14848-2: *Pt single atom for all-pH HER*, iii-10.1021/acssuschemeng.0c04322: *W-CoP for all-pH HER*]. Here again, the materials are different from ours. None of the references mentioned further modulating the intrinsic HER activity beyond SACs via introducing secondary metal to form SAD structure, suggesting that the targeted research motivations and scientific questions are different. In the i and ii references, the authors used noble metal-based clusters (Ru/Pt/Pd) and a single atom of Pt for achieving high HER activity, even knowing the fact that the noble metals are well known for superior HER performance and are highly expensive, limiting their practical applications. In our case, the NiCo-SAD-NC made solely of earth-abundant elements exhibited comparable pH-universal activity to that of commercial Pt, which is a rare occurrence. In the case of reference iii, the W single atom was grown on CoP substrate rather than carbon support. The CoP support itself exhibited a superior activity in wide-pH range and hence achieved high HER activity with additional single atoms on

it. Whereas, in our case, the NiCo-SAD is supported on the HER inactive carbon support. So, the comparable pH-universal HER activity to that of commercial Pt is solely from the NiCo-SAD rather than any substrate effect.

Taken together, we reasonably believe our work on SAD atomic interface is conceptually novel and the existence of the above-mentioned reference paper for HER could not affect the novelty of our work.

In addition, as the referee suggested these reference papers and for better comparison, we now included these suggested papers in **Supplementary Table 6** of the revised manuscript.

Supplementary Table 6. Comparison table of the HER performance of NiCo-SAD-NC with reported HER electrocatalysts.

Catalyst	$\eta@-10 \text{ mA/cm}^2$ in acidic media (mV)	$\eta@-10 \text{ mA/cm}^2$ in alkaline media (mV)	Reference
Comparison with single atom catalyst			
NiCo-SAD-NC	54.7	61	This work
Pt-Ru dimer	50		Nat. Commun. 10 , 4936 (2019)
Pt-SA/S-C	53		Nat. Commun. 10 , 4977 (2019)
Ru@Co-SAs/N-C	57	7	Nano Energy 59, 472-480 (2019)
Pt₁/N-C	19	46	Nat. Commun. 11, 1029 (2020)
W-CoP	48	40	ACS Sustainable Chem. Eng. 8, 14825-14832 (2020)
Pt/np-Co _{0.85} Se	58	58	Nat. Commun. 10 , 1743 (2019)
Fe/GD	66		Nat. Commun. 9 , 1460 (2018)
CoN ₄ -SAC		111	Adv. Funct. Mater. 31 , 2100547 (2021)

Ru _{SA} CoFe ₂ /G		164	Energy Environ. Sci. , 13 , 5152-5164 (2020)
Co ₁ /PCN		89	Nat. Catal. 2 , 134–141 (2019)
Co-SA@NCA		78.6	Chem. Eng. J. , 410 , 128359 (2021)
Pt@PCM	105	139	Sci. Adv. 4 , eaao6657 (2018)
Mo-SAC	154	132	Angew. Chem. Int. Ed. 56 , 16086–16090 (2017)
Mo SAs/ML-MoS ₂	107	209	ACS Nano 14 , 767–776 (2020)
Pt-SA decorated VS ₂	77		ACS Nano 14 , 5600-5608 (2020)
Ni NP Ni-N-C		147	Energy Environ. Sci. 12 , 149-156 (2019)
Co-C-N	138	178	J. Am. Chem. Soc. 137 , 15070–15073 (2015)
Co-NG-MW film	127		Adv. Mater. 30 , 1802146 (2018)
Ni _{SA} -MoS ₂	110	98	Nano Energy 53 , 458-467 (2018)
Co-SAS/HOPNC	137		Proc. Natl. Acad. Sci. U.S.A. 115 , 12692–12697 (2018)

Comparison with nanomaterials-based catalyst

Pt ₃ Ni ₂ NWs-S/C	70		Nat. Commun. 8 , 14580 (2017)
Pt-Ni nanowires		13	Adv. Mater. 31 , 1807780 (2019)
SAP-Mo ₂ C-CS	36		Angew. Chem. Int. Ed. , 59 , 23791-23799 (2020)
PtNi-O/C	70		J. Am. Chem. Soc. 140 , 9046–9050 (2018)

Mn-doped FeP/Co ₃ (PO ₄) ₂	27	85	ChemSusChem 12 , 1334–1341 (2019)
3D-NiCoP	80	105	Nano Res. 12 , 375–380 (2019)
CoNi@NC	142		Angew. Chem. Int. Ed. , 54 , 2100-2104 (2015)
CP/CTs/Co-S		190	ACS Nano 10 , 2342- 2348 (2016)
Co-Ni-P-300		150	Chem. Commun. 52 , 1633-1636 (2016)
Co-NiS ₂ NSs		80	Angew. Chem. Int. Ed. 58 , 18676 –18682 (2019)
NiCoN/C		103	J. Am. Chem. Soc. 140 , 610-617 (2018)
NiCo ₂ S ₄		65	Nano Energy 24 , 139-147 (2016)
NiO/Ni/CNT		86	Nat Commun. 5 , 4695 (2014)

2) The authors should provide more clear evidence for the bimetallic single-atom-dimer. Neither the HAADF-STEM images nor EXAFS results is convincing, both of them seems to be subjective judgments. For example, more clear HAADF-STEM images of the dimers are needed (*Angew. Chem.* 2020, 132, 16147).

Answer 2) We thank the referee for carefully reviewing our manuscript and providing valuable suggestions for improving the quality of our manuscript. We agree with the referee that our previous HAADF-STEM image and the EXAFS results along with the intensity profile and EEL spectra for the NiCo-SAD-NC were not so evident. Following the referee’s suggestions, we newly collected a much improved aberration-corrected HAADF-STEM image of the NiCo-SAD-NC clearly showing the presence of multiple NiCo dimer sites, further verified using clear intensity profiles and EEL spectra, newly provided in **Figs. 2b-e** and **Supplementary Fig. 10** of the revised manuscript. The newly collected aberration-corrected HAADF-STEM image of NiCo-SAD-NC in **Fig. 2b** clearly demonstrated the existence of isolated Ni-Co dimer sites (marked by yellow square) with coordination between Ni and Co atoms at the atomic level along with some isolated Ni or Co atoms (marked by the orange circle). The intensity profiles along with the corresponding EEL spectra taken at multiple homogeneously distributed bright dual dots marked by yellow

squares further confirmed the existence of Ni-Co dimer sites with an average dimer distance of 0.241 ± 0.024 nm (Figs. 2c,d,e, and Supplementary Fig. 10). Supplementary Fig. 10 showed a series of intensity profiles and EEL spectra taken at multiple dimer sites marked by yellow squares, clearly providing convincing evidence for the existence of multiple NiCo dimer sites in the NiCo-SAD-NC. The newly collected EEL spectra of the NiCo dimer sites are better than the previously reported Co-Fe dual-atom pair catalysts [*Energy Environ. Sci.*, **11**, 3375—3379 (2018)] and clearly provides the evidence for the existence of NiCo dimer in the NiCo-SAD-NC.

In addition, to further provide the evidence for the existence of NiCo dimer sites in the NiCo-SAD-NC sample, we carried out proper Ni/Co K-edge EXAFS fitting in R space oscillations for determining the structural information of NiCo-SAD-NC such as Ni/Co-N and Ni-Co coordination number (CN) and bond distance, newly provided in Supplementary Fig. 19 and Supplementary Table S5 of the revised manuscript. The Ni/Co K-edge EXAFS fitting results clearly revealed that the CNs of Ni-N and Co-N were 3.3 and 3.1, respectively, with the bond distance of 1.86 Å (Ni-N) and 2.01 Å (Co-N), suggesting that the NiCo-SAD-NC was dominated by the Ni-N₃ and Co-N₃ environment. Meanwhile, the Ni-Co and Co-Ni paths were fitted at a position of 2.55 Å with a CN of 0.7 and 0.6, respectively, directly indicating the existence of Ni-Co bonding dimer in the form of NiCo-N₆ structure, consistent with the DFT calculated structure. The obtained CNs for Ni-N/Co-N and Ni-Co/Co-Ni were consistent with the previously reported Co-Fe and Co-Zn dual-atom pair catalysts [*Angew. Chem. Int. Ed.* **131**, 2648–2652 (2019); *Energy Environ. Sci.*, **11**, 3375—3379 (2018)].

Therefore, combining the much-improved aberration-corrected HAADF-STEM image, series of intensity profiles, and EEL spectra along with proper EXAFS fitting clearly provided the evidence for the existence of NiCo dimer sites in the NiCo-SAD-NC. The text is modified on p. 9, and 12 of the revised manuscript as

Aberration-corrected HAADF-STEM image in Fig. 2b clearly demonstrated the existence of isolated Ni-Co dimer sites (marked by yellow square) with coordination between Ni and Co at atomic level along with some isolated Ni or Co atoms (marked by the orange circle). The homogeneously distributed bright dual dots marked by yellow squares confirmed the existence of Ni-Co dual sites, verified using the intensity profile and corresponding electron energy loss (EEL) spectra (Figs. 2c,d). The bright Ni-Co dual dots were clearly identified in the intensity profiles and corresponding EEL spectrum, suggesting the possible formation of metal-metal bonds with an average dimer distance of 0.241 ± 0.024 nm, obtained from the statistical analysis over multiple dimer sites (Fig. 2e, and Supplementary Fig. 10). (p. 9)

Furthermore, the structural information of NiCo-SAD-NC along with the Ni/Co-N and Ni-Co coordination numbers (CNs) were also affirmed by the EXAFS fitting results in R space oscillations (Supplementary Fig. 19 and Supplementary Table 5). The EXAFS fitting results clearly revealed that the CNs of Ni-N and Co-N were 3.3 and 3.1, respectively, suggesting that the

NiCo-SAD-NC was dominated by the Ni-N₃ and Co-N₃ environment. Meanwhile, the Ni-Co/Co-Ni paths were fitted at a position of 2.55 Å with a CN of 0.7/0.6, directly indicating the existence of Ni-Co bonding dimer in the form of NiCo-N₆ structure, in consistent with the DFT calculation structure. (p. 12)

Fig. 2 | Structural analysis and electron microscopy of NiCo-SAD-NC. **a**, XRD spectra of NC, NiCo-NP-NC, and NiCo-SAD-NC. **b**, Aberration-corrected HAADF-STEM image of the NiCo-SAD-NC. The yellow squares in (b) shows the dimer sites and some of single Ni/Co-atom sites are highlighted by orange circles. **c,d**, The intensity profile and corresponding EEL spectra obtained at site A (c) and site B (d) for NiCo-SAD-NC, indicate that Ni and Co are coordinated at the atomic scale. **e**, Statistical Ni-Co distance in the observed dimers. **f**, HAADF-STEM image and corresponding EDS maps of NiCo-SAD-NC showing the uniform dispersion of N (green), Co (red) and Ni (yellow).

Supplementary Figure 10 | a, Aberration-corrected HAADF-STEM image of the NiCo-SAD-NC. The yellow squares in **(a)** show the dimer sites and some of single Ni/Co-atom sites are highlighted by the orange circles. **b**, The intensity profile obtained from sites 1-10 showing the distance between Ni and Co in the observed dimers at the atomic scale. **c**, The corresponding EEL spectra obtained from site 1-7, 9, and site i, ii showing the Ni and Co coordination in the dimer along with few Ni/Co single-atom sites.

Previous Fig. 2 | Structural analysis and electron microscopy of NiCo-SAD-NC. **a**, XRD spectra of NC, NiCo-NP-NC and NiCo-SAD-NC. **b,c**, Aberration-corrected HAADF-STEM image (**b**) and magnified view (**c**) of the NiCo-SAD-NC. The red circles in (**c**) shows the dimer sites and the inset shows the corresponding EEL spectra taken at site 2 indicate Ni and Co are coordinated at the atomic scale. **d**, The intensity profile obtained at NiCo dimer site 1 and 2. **e**, Statistical Ni-Co distance in the observed dimers. **f**, HAADF-STEM image and corresponding EDS maps of NiCo-SAD-NC showing the uniform dispersion of N (green), Co (red) and Ni (yellow). (Previous Fig. 2 in our original manuscript for reference to the referee)

Supplementary Figure 19 | **a**, and **b**, The Ni K-edge, and Co K-edge FT-EXAFS, respectively, of NiCo-SAD-NC and Ni/Co metal with corresponding EXAFS fitting curves. **c**, The DFT simulated NiCo-SAD-NC model based on the FT-EXAFS fitting value.

Supplementary Table 5. EXAFS fitting parameters at the Ni and Co K-edge for NiCo-SAD-NC and Ni/Co metal ($S_0^2 = 0.780$ (Ni), 0.816 (Co)).

Sample	Shell	N^a	$R(\text{\AA})^b$	$\sigma^2(\text{\AA}^2)^c$	$\Delta E_0(\text{eV})^d$	R factor
Co K-edge						
Co Metal	Co-Co	12	2.49	0.0060	6.6	0.0002
NiCo-SAD-NC	Co-N	3.1	2.01	0.0018	-4.5	0.0001
	Co-Ni	0.6	2.55	0.0126		
Ni K-edge						
Ni Metal	Ni-Ni	12	2.48	0.0059	6.3	0.0001
NiCo-SAD-NC	Ni-N	3.3	1.86	0.0039	-4.8	0.0016
	Ni-Co	0.7	2.55	0.0121		

^a N : coordination numbers; ^b R : bond distance; ^c σ^2 : Debye-Waller factors; ^d ΔE_0 : the inner potential correction. R factor: goodness of fit. S_0^2 was set to 0.816 for Co and 0.780 for Ni, according to the experimental EXAFS fit of Co and Ni metal reference by fixing CN as the known crystallographic value.

3) EELs spectra of Figure 2C is not convincing, due to the transmission property of TEM. The signals of atoms below the surface would be reckoned in. HAADF-STEM image with matched EELs mapping should be provided.

Answer 3) Following the referee's suggestions, we newly collected much improved aberration-corrected HAADF-STEM image of the NiCo-SAD-NC along with the series of intensity profiles and EEL spectra taken at multiple dimer sites marked by yellow squares, newly provided in **Supplementary Fig. 10** of the revised manuscript. As shown in **Supplementary Fig. 10**, a series of intensity profiles and EEL spectra taken at multiple dimer sites marked by yellow squares clearly provided convincing evidence for the existence of multiple NiCo dimer sites in the NiCo-SAD-NC. The text is modified on **p. 9** of the revised manuscript as

The bright Ni-Co dual dots were clearly identified in the intensity profiles and corresponding EEL spectrum, suggesting the possible formation of metal-metal bonds with an average dimer distance of 0.241 ± 0.024 nm, obtained from the statistical analysis over multiple dimer sites (Fig. 2e, and Supplementary Fig. 10).

Supplementary Figure 10 | **a**, Aberration-corrected HAADF-STEM image of the NiCo-SAD-NC. The yellow squares in **(a)** shows the dimer sites and some of single Ni/Co-atom sites are highlighted by orange circles. **b**, The intensity profile obtained from site 1-10 showing the distance between Ni and Co in the observed dimers at the atomic scale. **c**, The corresponding EEL spectrum obtained from site 1-7, 9, and site i, ii showing the Ni and Co coordination in the dimer along with few Ni/Co single-atom sites.

4) The authors also evaluated the energy barrier for H₂O dissociation of some selected TM-SAD structures, and get the conclusion that NiCo-SAD is a promising candidate for HER. According to Fig. 1c,d, CoMn is so closed to NiCo, while CoFe SAD is far from NiCo. However, the LSV lines of CoFe and CoMn are nearly overlaid in Fig. S24. The authors should also provide explanation about this.

Answer 4) We sincerely thank the referee for carefully reviewing our manuscript, and we are pleased to clarify the issue. In the alkaline media (pH=14), the HER starts with water dissociation (Fig. 1c,d), which is necessary to generate sufficient H*, followed by H* adsorption on the catalysts surface (Fig. 1e) and subsequent dimerization to molecular H₂. The overall HER activity of a catalyst in the alkaline media depends on both the water dissociation energy barrier and H* adsorption energy on the surface of the catalyst. As revealed in Fig. 1c,d,e, although the water dissociation kinetic barrier for the CoMn-SAD (2.55 eV) was close to that of NiCo-SAD (2.52 eV) and better than that of CoFe-SAD (3.11 eV), the H* adsorption energy for the CoMn-SAD (-0.46 eV) was much far from the thermoneutral point of 0 eV compared to that of CoFe-SAD (-0.121 eV) and NiCo-SAD (-0.34 eV) (Supplementary Table 2a). Therefore, for CoMn-SAD, the overall alkaline HER activity was hampered by the poor H* adsorption energy, making H₂ desorption more sluggish despite reasonable water dissociation step. However, for CoFe-SAD, despite relatively sluggish water dissociation step, the overall alkaline HER activity was improved by the optimal H* adsorption energy, facilitating the proper desorption of generated H₂. Hence, considering together the water dissociation and H* adsorption step, we expect that the alkaline HER activities of CoFe-SAD and CoMn-SAD can be close to each other. Consistent with the DFT prediction, we experimentally found that the HER LSV polarization curves for the CoFe-SAD-NC and CoMn-SAD-NC in alkaline media were close to each other with CoMn-SAD-NC requiring an overpotential of 116.5 mV at -10 mA/cm² slightly better than CoFe-SAD-NC (124.9 mV) (Supplementary Fig. 28). Additionally, in our original manuscript, based on our DFT results, we already mentioned that “*by comparing the |U_L| of the TM-SADs, three potential candidates with the lowest theoretical overpotentials were observed in the following order: NiCo-SAD-N6C (0.460 eV) < CoMn-SAD-N6C (0.465 eV) < CoFe-SAD-N6C (0.472 eV)*”. The text is modified on p. 7 and S29 of the revised manuscript as

By comparing the $|U_L|$ of the TM-SADs, three potential candidates with the lowest theoretical overpotentials were observed in the following order: NiCo-SAD-N₆C (0.460 eV) < CoMn-SAD-N₆C (0.465 eV) < CoFe-SAD-N₆C (0.472 eV). (p. 7)

Consistent with the DFT prediction, we experimentally found that the HER LSV polarization curves for the CoFe-SAD-NC and CoMn-SAD-NC in alkaline media were close to each other with CoMn-SAD-NC requiring an overpotential of 116.5 mV at -10 mA/cm² slightly better than CoFe-SAD-NC (124.9 mV) (Supplementary Fig. 28). (p. S29)

Fig. 1 | High-throughput screening of various transition metal-based SAD (TM-SAD) for HER. **a**, Formation energies corresponding to the charge depletion and d-band center of various TM-SAD-N₆C. **b**, different charge density distribution of TM-SAD-N₆C. The charge depletion and accumulation are denoted by green and red colors, respectively. **c**, Minimum energy paths of water splitting reactions on TM-SAD-N₆C (blue: Co and grey: Ni). **d**, linear correlation between the d-band center and kinetic energy barrier. **e**, Free energy diagram of hydrogen adsorption and **f**, free energy changes of the hydronium and hydroxide desorption step for TM-SAD-N₆C.

Supplementary Table 2. a, D-band center of the TM-SAD-N₆C systems with H* and H₂O* corresponding to the Gibbs adsorption free energy of the H* and energy barrier of water dissociation, respectively.

a

	D-band center of SAD systems with H* (eV)	ΔG_{H^*} (eV)	D-band center of SAD systems with H ₂ O* (eV)	E_{split} (eV)
CoCu	-2.16	0.126	-1.67	3.33
NiCo	-1.28	-0.34	-0.87	2.52
CoFe	-0.88	-0.121	-1.07	3.11
CoMn	-0.78	-0.46	-0.97	2.55
CuCu	-2.30	0.783	-2.82	4.22
CoCo	-0.86	-0.355	-1.50	2.94
FeFe	-0.90	-0.47	-1.05	2.64
MnMn	-0.16	-0.135	-0.89	2.76

Supplementary Figure 28 | a, HER LSV polarization curve of NiCo-SAD-NC with CoMn-SAD-NC and CoFe-SAD-NC, in 1 M KOH. **b**, HER LSV polarization curve of NiCo-SAD-NC with CoFe-SAD-NC in 0.5 M H₂SO₄.

5) In the electro catalytic performance part, the authors tested the HER performance of NiCo-SAD-NC and other counterparts. It was concluded that the HER followed the Volmer-

Heyrovsky mechanism. The authors should provide detailed mechanism analysis about the roles of Ni site and Co site in Ni-Co pair. What's more, the existence of NiCo pair in NiCo-SAD-NC samples after stability test should be confirmed and need more discussion.

Answer 5) *“In the electro catalytic performance part, the authors tested the HER performance of NiCo-SAD-NC and other counterparts. It was concluded that the HER followed the Volmer-Heyrovsky mechanism. The authors should provide detailed mechanism analysis about the roles of Ni site and Co site in Ni-Co pair.”*

We thank the referee for the comment, and we are pleased to clarify the issue. We now provided the detailed discussion regarding the roles of Ni and Co sites in Ni-Co pair, newly provided in **Supplementary Discussion 2** of the revised manuscript. Notably, H₂O tends to be adsorbed on the top site of the metal atom with lower electronegativity corresponding to a higher positive charge. Hence, it is adsorbed on the Fe/Mn top sites in the case of MnCo/FeCo-SAD-NC, and on the Co top site in the NiCo/CuCo-SAD-NC case. According to our results, NiCo-SAD-NC had the smallest water dissociation barrier of 2.52 eV. This means that the Co site in the NiCo-SAD-NC adsorbed the H₂O and facilitated the H-OH cleaving of the water molecules. After water dissociation, H* was preferably adsorbed on the higher electronegative metal sites (Ni) (Volmer step), and *OH remained on the lower electronegativity Co site, followed by proton dimerization to molecular H₂ from the Ni site (Heyrovsky step).

The text is present on **p. S24** of the revised manuscript as

Supplementary Discussion 2:

The most stable water absorption geometries are illustrated in the below table.

	$\Delta\mu$	$\Delta\delta$ (e^-)
CuCo	0.02	-0.036
FeCo	-0.05	0.922
MnCo	-0.33	0.446
NiCo	0.03	-0.158

The difference in electronegativity ($\Delta\mu=M_{\mu}-Co_{\mu}$) and Mulliken charge difference ($\Delta\delta=M_{\delta}-Co_{\delta}$) on MCo-SAD-NC.

Notably, H₂O tends to be adsorbed on the top site of the metal atom with lower electronegativity corresponding to a higher positive charge. Hence, it is adsorbed on the Fe/Mn top sites in the case of MnCo/FeCo-SAD-NC, and on the Co top site in the NiCo/CuCo-SAD-NC case. According to

our results, NiCo-SAD-NC had the smallest water dissociation barrier of 2.52 eV. This means that the Co site in the NiCo-SAD-NC adsorbed the H₂O and facilitates the H-OH cleaving of the water molecules. After water dissociation, H* was preferably adsorbed on the higher electronegative metal sites (Ni) (Volmer step) and *OH remained on the lower electronegativity Co site, followed by proton dimerization to molecular H₂ from the Ni site (Heyrovsky step).

“What’s more, the existence of NiCo pair in NiCo-SAD-NC samples after stability test should be confirmed and need more discussion.”

We agree with the referee that the characterizations of the catalyst after the stability test are very necessary. Following the referee’s suggestion, we newly collected post-stability aberration-corrected HAADF-STEM image, EEL spectra, intensity profile, XANES, and EXAFS spectra for the NiCo-SAD-NC after the stability test in both alkaline/acidic media, newly provided in Supplementary Fig. 34 of the revised manuscript. As revealed in Supplementary Fig. 34a-d, post-stability aberration-corrected HAADF-STEM images along with intensity profiles and EEL spectra were taken at site A confirmed the existence of NiCo dimer sites in the NiCo-SAD-NC after the stability test in both alkaline and acidic media. The nearly matching Ni/Co K-edge XANES spectra of NiCo-SAD-NC taken before and after the alkaline/acidic stability test further confirmed that the Ni/Co oxidation states along with pre-edge features also remained intact (Supplementary Fig. 34e,g). Similarly, the Ni/Co K-edge FT-EXAFS spectra confirmed that the Ni/Co-N and Ni-Co bonding in the NiCo-SAD-NC also remained intact after the stability test (Supplementary Fig. 34f,h).

Additionally, in our original manuscript, we already provided the post-stability characterizations such as XRD, SEM, TEM, EDX, and XPS of the NiCo-SAD-NC to investigate any structural and electronic changes after the long-term stability test in alkaline and acidic media (Supplementary Figs. 35 and 36). As revealed in Supplementary Figs. 35a and 36a, the post-stability XRD spectra of NiCo-SAD-NC were similar to the XRD spectra of the pristine NiCo-SAD-NC, suggesting that crystal structure remained intact without any aggregation of Ni/Co to form nanoparticles. Similarly, the post-stability electron microscopy (SEM, TEM, and HAADF-STEM EDS mapping) of the NiCo-SAD-NC further revealed that the morphology remains intact without any visible aggregates of the Ni/Co, and the Ni and Co atoms remained uniformly distributed (Supplementary Figs. 35b,c,d and 36b,c,d). The post-stability EDS spectra confirmed that the Ni/Co composition remained intact after the stability test (Supplementary Figs. 35e and 36e). After the stability test, the XPS spectra were well matched with the pristine NiCo-SAD-NC along with well intact metal-N bonds and Ni/Co oxidation states, confirming that the NiCo-SAD structure remained well preserved (Supplementary Figs. 35f,g,h and 36f,g,h).

Therefore, combining HAADF-STEM, EELs, XANES/EXAFS, XRD, XPS, and EDX analysis, we concluded that the NiCo-SAD-NC was stable after the stability test in both acidic/alkaline media will well-preserved NiCo dimer sites.

The text is present on p. 15 and S34 of the revised manuscript as

Post-stability aberration-corrected HAADF-STEM images with corresponding intensity profiles and EEL spectrum along with nearly matching XANES and EXAFS spectra clearly revealed that the Ni-Co dimer sites in the NiCo-SAD-NC were well-preserved without any sign of metal agglomeration to form clusters/NPs (Supplementary Fig. 34). In addition, the crystal structure, morphology, and composition of the NiCo-SAD-NC were also maintained after the HER durability test (Supplementary Figs. 35 and 36). After the stability test, the XPS spectra were matched with the pristine NiCo-SAD-NC along with well intact metal-N bonds. (p. 15)

Supplementary Fig. 34a-d showed the STEM images along with intensity profiles and EEL spectra taken at site A confirmed the existence of NiCo dimer sites after the stability test in both alkaline and acidic media. The overlapping Ni/Co K-edge XANES spectra of NiCo-SAD-NC taken before and after the alkaline/acidic stability test further revealed that the Ni/Co oxidation states along with pre-edge features remained intact (Supplementary Fig. 34e,g). Similarly, the Ni/Co K-edge FT-EXAFS spectra confirmed that the Ni/Co-N and Ni-Co bonding in the NiCo-SAD-NC also remained intact after the stability test (Supplementary Fig. 34f,h). (p. S34)

Supplementary Figure 33 | **a,b**, Aberration-corrected HAADF-STEM image with yellow squares showing the dimer sites (**a**), and corresponding intensity profile and EEL spectrum obtained at site A (**b**) for NiCo-SAD-NC after stability test in alkaline media. **c,d**, Aberration-corrected HAADF-STEM image with yellow squares showing the dimer sites (**c**), and corresponding intensity profile and EEL spectrum obtained at site A (**d**) for NiCo-SAD-NC after stability test in acidic media. **e,f,g,h**, experimental Ni K-edge XANES (**e**), Ni K-edge FT-EXAFS (**f**), Co K-edge XANES (**g**), and Co K-edge FT-EXAFS (**h**) spectra of NiCo-SAD-NC after stability test in alkaline and acidic media.

Supplementary Figure 34 |a, b, c, d, and e, XRD pattern, FESEM image, TEM image, HAADF-STEM images and corresponding EDS elemental mapping, and EDS pattern, respectively, of NiCo-SAD-NC after stability test in alkaline media. f, g, and h Fitted deconvoluted N 1s, Ni 2p and Co 2p, respectively, XPS spectra of NiCo-SAD-NC after stability test in alkaline media.

Supplementary Figure 35 |a, b, c, d, and e, XRD pattern, FESEM image, TEM image, HAADF-STEM images and corresponding EDS elemental mapping, and EDS pattern, respectively, of NiCo-SAD-NC after stability test in acidic media. f, g, and h Fitted deconvoluted N 1s, Ni 2p and Co 2p, respectively, XPS spectra of NiCo-SAD-NC after stability test in acidic media.

We truly thank the Reviewer 3 for the insightful comments and kind suggestions.

REVIEWER COMMENTS

Reviewer #1 (Remarks to the Author):

The corrected version of the paper by Kumar et al. has been significantly improved and we acknowledge the efforts done by the authors. However, a series of points are still questionable, as detailed below. Although the assignment of dimeric species is not impossible, I don't see enough positive evidence for this structure to be the only one present and the one that is indeed being active in catalysis. We also noticed that a very similar study was published previously in Angewandte (DOI: 10.1002/anie.201901575), which we did not see previously. The concept of dimeric structure proposed by the author is therefore not new, provided that it would be experimentally confirmed.

These issues therefore prevents us from accepting this paper.

- The lattice dislocation observed by HRTEM and shown on figure S8e are not convincing at all. It is impossible to see what is the common feature between the points on the image selected by the yellow circles. This data is not giving any improvement to the analysis in our opinion.

- The ratio difference (from 0.96 to 1.10) observed on the D and G bands of the Raman spectra are very subtle and hardly convincing. Unless a statistical analysis is provided to show that this type of ratio is highly reproducible in a pristine or metallated sample, than it has no spectroscopic value.

- The EXAFS fittings are interesting and we appreciate the author's efforts to do that. However, the TEM data show some single sites of Ni or Co (figure 2b), probably as much as 30% of the total amount. How does the EXAFS fits take these species into account? I suppose that they don't in the current state of fittings. What is the situation of these single sites? Are they coordinated by nitrogen ligands as well? If so, then they should be accounted for in the N values of both the M-N and M-M interactions.

Concerning the Ni-Co interaction, why is the N value equal to 0.6/0.7? It should be closer to 1 if there were only dimers.

Also, the fitting procedure could be a bit more thorough, with the fits of other possible structures (such as single site hypothesis). The R-factor have no absolute values unless they are compared between different structural hypotheses (monomer vs. dimer, for exemple).

The presence of these single sites is actually a serious concern, because there is no evidence that catalysis is occuring on the dimeric sites rather than on the monomeric ones, hence compromising the novelty of the paper.

- The EDS elemental mapping with the the different elements provided on figure 11 is interesting, but it does not support the author's conclusions. The colocalisation of Ni, Co and N is far from obvious from this image. and does not support the dimeric structure proposed.

- In XPS, the collection of data for reference samples is a very good thing. But what does the peak at 785 eV mean in the Co 2p_{3/2} spectrum?

- Although we appreciate the use of NiPc and CoPc as appropriate reference samples, the analysis of the XANES pre-edge region and the conclusions drawn from there is still not convincing. In CoPc, the pre-edge features at 7709 and 7715 eV are linked to each other and related to the planarity of the system. When the system is very flat (D_{4h}), then the first feature is supposed to be low and the second one is high (see DOI:10.1039/c4ra15306e). As a consequence, a strong first peak associated to a low intensity second peak indicates a deviation from the D_{4h} geometry. The explanation given by the authors that the deviation is due a substitution of the 4th nitrogen in Pc by the second metallic center is not satisfactory. This explanation was given in the first place by Ren et al. in their Angewandte paper (DOI: 10.1002/anie.201901575), but their explanatino is not convincing to me either. It is not clear at all how this substitution would change the geometry, and the DFT calculated structure does not seem to show a serious deviation from a D_{4h} geometry. This explanation is actually not impossible, but a lot of other explanations could also be valid.

Reviewer #2 (Remarks to the Author):

All my concerns are well addressed and I believe this manuscript is now acceptable for publication.

Reviewer #3 (Remarks to the Author):

I think the revised manuscript have meet the high standards of the Nature Communications.

Point-by-point response for Nature Communications manuscript
(ID: NCOMMS-21-19692A)

Manuscript Type: Article

Title: Moving Beyond Bimetallic-Alloy to Single-Atom Dimer Atomic-Interface for All-pH Hydrogen Evolution.

Author(s): Ashwani Kumar, Viet Q. Bui, Jinsun Lee, Lingling Wang, Amol R. Jadhav, Xinghui Liu, Xiaodong Shao, Yang Liu, Jianmin Yu, Yosep Hwang, Huong T. D. Bui, Sara Ajmal, Min Gyu Kim, Seong-Gon Kim, Gyeong-Su Park, Yoshiyuki Kawazoe, and Hyoyoung Lee.

We are grateful to the editor, editorial staff, and reviewers for their critical comments and valuable suggestions. The manuscript has been modified after addressing all the suggestions as listed below:

(The explanations to the comments from reviewers are shown with **yellow highlight**).

Author(s)' Points-by-points responses to Reviewer(s)':

Reviewer: 1

Comments:

Reviewer #1 (Remarks to the Author):

The corrected version of the paper by Kumar et al. has been significantly improved and we acknowledge the efforts done by the authors. However, a series of points are still questionable, as detailed below. Although the assignment of dimeric species is not impossible, I don't see enough positive evidence for this structure to be the only one present and the one that is indeed being active in catalysis. We also noticed that a very similar study was published previously in Angewandte (DOI: 10.1002/anie.201901575), which we did not see previously. The concept of dimeric structure proposed by the author is therefore not new, provided that it would be experimentally confirmed. These issues therefore prevents us from accepting this paper:

Response: We are grateful for the time and effort Reviewer 1 has spent in reviewing our manuscript. The review comments are constructive for further strengthening the manuscript.

1) The lattice dislocation observed by HRTEM and shown on figure S8e are not convincing at all. It is impossible to see what is the common feature between the points on the image selected by the yellow circles. This data is not giving any improvement to the analysis in our opinion.

Answer 1) We sincerely thank the referee for carefully reviewing our manuscript. We agree with the referee that the HRTEM image of the NiCo-SAD-NC does not provide convincing information regarding the lattice distortion defects. As a result, we removed the enlarged view of the HRTEM image and also the previous conclusion regarding the presence of lattice distortion defects from the HRTEM image of NiCo-SAD-NC. Now, from the HRTEM image and SAED pattern of NiCo-SAD-NC (Supplementary Fig. 8e,f), we concluded that they additionally confirmed the absence of any metal aggregations in the form of small clusters, consistent with the XRD pattern, suggesting that the metal atoms were atomically dispersed on the carbon support. The text is modified on p. S11 of the revised supporting information as

The HRTEM image with the corresponding SAED pattern (Supplementary Fig. 8e, f) of NiCo-SAD-NC additionally confirmed the absence of any metal aggregations in the form of small clusters, consistent with the XRD pattern, suggesting that the metal atoms were atomically dispersed on the carbon support.

Supplementary Figure 8 | **a, b, and c**, TEM, HRTEM and STEM-HAADF images with the corresponding element maps (N, Co, Ni), respectively, of NiCo-NP-NC. The inset in **b** shows the corresponding SAED pattern. **d, e, and f**, TEM, HRTEM image, and SAED pattern, respectively, of NiCo-SAD-NC.

Previous Supplementary Figure 8 | **a**, **b**, and **c**, TEM, HRTEM and STEM-HAADF images with the corresponding element maps (N, Co, Ni), respectively, of NiCo-NP-NC. The inset in **b** shows the corresponding SAED pattern. **d**, and **e**, TEM, and HRTEM image, respectively, of NiCo-SAD-NC. (**For reference to referee**)

2) The ratio difference (from 0.96 to 1.10) observed on the D and G bands of the Raman spectra are very subtle and hardly convincing. Unless a statistical analysis is provided to show that this type of ratio is highly reproducible in a pristine or metallated sample, than it has no spectroscopic value.

Answer 2) We once again thank the referee for carefully reviewing our manuscript. We also agree with the referee that the difference in the I_D/I_G ratio between NC and NiCo-SAD-NC is very subtle and it is difficult to conclude that the NiCo-SAD-NC exhibits higher defects compared to NC. As suggested by the referee, we collected multiple Raman spectra of NC and NiCo-SAD-NC, newly provided in **Supplementary Fig. 9** of the revised supporting information. As revealed from **Supplementary Fig. 9**, the I_D/I_G ratio between NC and NiCo-SAD-NC are almost similar. Therefore, we removed the previous conclusion regarding the formation of more carbon defects in NiCo-SAD-NC compared to NC. Now, from the Raman spectra in **Supplementary Fig. 9**, we just concluded that the Raman spectra of both NC and NiCo-SAD-NC showed a characteristic D band at 1338 cm^{-1} and G band at 1591 cm^{-1} , corresponding to carbon lattice defect and sp^2 -hybridized carbon atoms, respectively, consistent with their corresponding XRD results.

The text is modified on p. 9 and S12 of the revised manuscript and supporting information, respectively as

In addition, the Raman spectra of both NC and NiCo-SAD-NC showed the characteristics D and G band, consistent with their corresponding XRD results (Supplementary Fig. 9). (p. 9)

The Raman spectra of both NC and NiCo-SAD-NC showed a characteristic D band at 1338 cm^{-1} and G band at 1591 cm^{-1} , corresponding to carbon lattice defect and sp^2 -hybridized carbon atoms, respectively, consistent with the XRD pattern (Supplementary Fig. 9). (p. S12)

Supplementary Figure 9 | a, and b, Statistical analysis of Raman spectra of NC and NiCo-SAD-NC, respectively.

3) The EXAFS fittings are interesting, and we appreciate the author's efforts to do that. However, the TEM data show some single sites of Ni or Co (figure 2b), probably as much as 30% of the total amount. How does the EXAFS fits take these species into account? I suppose that they don't in the current state of fittings. What is the situation of these single sites? Are they coordinated by nitrogen ligands as well? If so, then they should be accounted for in the N values of both the M-N and M-M interactions. Concerning the Ni-Co interaction, why is the N value equal to 0.6/0.7? It should be closer to 1 if there were only dimers. Also, the fitting procedure could be a bit more thorough, with the fits of other

possible structures (such as single site hypothesis). The R-factor have no absolute values unless they are compared between different structural hypotheses (monomer vs. dimer, for example). The presence of these single sites is actually a serious concern, because there is no evidence that catalysis is occurring on the dimeric sites rather than on the monomeric ones, hence compromising the novelty of the paper.

Answer 3) We thank the referee for the comment, and we are pleased to clarify the issue. We agree with the referee that the identification of the real active site is very important and necessary.

(i) HAADF-STEM and EXAFS fitting results:

The HAADF-STEM image of NiCo-SAD-NC in Fig. 2b demonstrated the existence of isolated Ni-Co dimer sites (marked by yellow square) along with some isolated Ni or Co atoms (marked by the orange circle). Now, we newly estimated and plotted the distribution histogram showing the ratio between dimers and single-atom sites, newly provided in Supplementary Fig. 11a of the revised manuscript. As shown in Supplementary Fig. 11a, the ratio of dimer structure was around 78%, indicating that a significant amount of this type of structure was present in the prepared NiCo-SAD-NC material. In the Ni/Co K-edge EXAFS fitting (Supplementary Fig. 19a,b and Supplementary Table 5), we fitted the Ni/Co K-edge EXAFS spectra with the NiCo-N6 dimer configuration as they were the dominant species and we found that the coordination number for the Ni-Co/Co-Ni path were 0.7/0.6 with very small R-factor values of 0.0001 and 0.0016 for the Co and Ni K-edge EXAFS fitting, respectively. Since the results of EXAFS fitting are the average of all Ni and Co atoms, hence the coordination number of 0.6/0.7 directly indicated that not all the Ni/Co atoms formed dimer structure and there are some Ni or Co single sites also present, consistent with the HAADF-STEM image of NiCo-SAD-NC which showed that ~78% of dimer sites were present along with ~22% of single-atom sites stabilized by the N coordination. These dimer sites in the NiCo-SAD-NC are the real active site for the excellent pH-universal HER activity, whereas the single-atom sites were inactive and extremely sluggish for H₂O dissociation, which we already verified using the theoretical calculations and experimental validation.

In addition, following the referee's suggestions, we newly fitted the Ni/Co K-edge EXAFS spectra of NiCo-SAD-NC with the Co/Ni-N4 single atom configuration (Supplementary Fig. 19c,d and Supplementary Table 5) with R-factor values of 0.0082 and 0.0199 for the Co and Ni K-edge EXAFS fitting, respectively. The NiCo-SAD-NC FT-EXAFS spectra fitted with NiCo-N6 dimer configuration with R-factor values of 0.0001 and 0.0016 for the Co and Ni K-edge EXAFS fitting, respectively, were much smaller than the R-factor values of 0.0082 and 0.0199 for the Co and Ni K-edge EXAFS fitting, respectively, when fitted with Co/Ni-N4 single atom configuration, suggesting that the NiCo-SAD-NC FT-EXAFS spectra were best fitted with the NiCo-N6 dimer configuration, confirming that a significant amount of dimer structure was present in the prepared NiCo-SAD-NC material, consistent with the HAADF-STEM results.

(ii) Theoretical and Experimental validation for dimer as active site:

Our theoretical calculation revealed that for the Co and Ni single-atom sites, the hydronium (H⁺) and hydroxide (OH⁻) species auto-recombined to generate H₂O molecule after the optimization process in the final state of H₂O dissociation, suggesting that both Co and Ni single-atom sites were extremely sluggish for H₂O dissociation. Whereas the NiCo dimer site exhibited the lowest energy barrier for the H₂O dissociation along with the lowest theoretical overpotential for the HER, suggesting that the NiCo dimer site is the real active site for excellent HER performance. In addition, the HER polarization LSV curve in alkaline and acidic media of the NiCo-SAD-NC exhibited much higher performance compared to that of Co/Ni-SA-NC, experimentally confirming that the NiCo dimer site is the real active site for excellent HER performance, validating the DFT results (Supplementary Fig. 22a,c). Therefore, from the HAADF-STEM image along with EXAFS fitting of the NiCo-SAD-NC, we clearly identified that a significant amount (78%) of dimer sites was present in the prepared sample, which is the real active site for the excellent pH-universal HER activity, confirmed by DFT and experimental validation and the presence of a small quantity of inactive single-atom sites does not hamper the catalytic performance.

Previous references on dual atom pair catalysts and their EXAFS fitting results:

Additionally, the coordination number for M₁-M₂ (M₁/M₂: metal atoms) lower than 1 has also been previously reported for Zn-Co and Co-Fe isolated atomic pair catalysts [*Angew. Chem.* **131**, 2648–2652 (2019); *Energy Environ. Sci.*, **11**, 3375-3379 (2018)].

For the Zn-Co atomic pair catalyst [*Angew. Chem.* **131**, 2648–2652 (2019)], their HAADF-STEM image showed the existence of dual atom sites along with some individual Zn or Co single-atom sites. In addition, their EXAFS fitting revealed that the coordination number for Zn-Co/Co-Zn was 0.5±0.1 much lower than 1, suggesting that not all the Zn/Co atoms formed atomic pairs, consistent with the

[*Angew. Chem.* **131**, 2648–2652 (2019)]

HAADF-STEM.

Similarly, in the case of Co-Fe dual atomic site catalysts [*Energy Environ. Sci.*, **11**, 3375-3379 (2018)], their EXAFS fitting also showed that the coordination number for Co-Fe/Fe-Co was 0.6/0.8, again lower than 1, indicating that not all the Fe/Co atoms formed dual atomic pairs.

Considering the above two isolated atomic pair catalysts and their averaged EXAFS fitting results, we can conclude that the presence of a small number of single-atom sites can lower down the coordination number for M_1 - M_2 path below 1. Hence, due to the presence of the small amount of Ni/Co single-atom sites in NiCo-SAD-NC, the coordination number for the Ni-Co/Co-Ni path were 0.7/0.6, lower than the ideal coordination number (1). However, these dimer sites are the real active site for the excellent pH-universal HER activity.

Table S1. EXAFS data fitting results of (Fe,Co)/CNT.

Sample	Edge	Path	N	R(Å)	σ^2 (10 ⁻³ Å ²)	ΔE_0 (eV)	R-factor
(Fe,Co)/CNT	Fe	Fe-N ₁	2.2	1.98	3.9	-3.5	0.007
		Fe-N ₂	1.0	2.05	3.9	-3.5	
		Fe-Co	0.8	2.51	7.4	-4.0	
	Co	Co-N ₁	2.1	1.90	4.4	-6.2	0.007
		Co-N ₂	1.0	2.05	4.4	-6.2	
		Co-Fe	0.6	2.56	10.7	9.9	

N, coordination number; R, distance between absorber and backscatter atoms; σ^2 , the Debye-Waller factor value; ΔE_0 (eV), inner potential correction accounts for the difference in the inner potential between the sample and the reference compound.

[*Energy Environ. Sci.*, **11**, 3375-3379 (2018)]

The text is modified on p. 10, 12, and S21 of the revised manuscript and supporting information as

The ratio of dimer structure was around 78%, indicating the significant amount of this type of structure in the prepared NiCo-SAD-NC material (Supplementary Fig. 11a). (p. 10)

The EXAFS fitting results clearly revealed that the CNs of Ni-N and Co-N were 3.3 and 3.1, respectively, suggesting that the NiCo-SAD-NC was dominated by the Ni-N₃ and Co-N₃ environment. Meanwhile, the Ni-Co/Co-Ni paths were fitted at a position of 2.55 Å with CNs of 0.7/0.6, directly indicating the existence of a significant amount of Ni-Co bonding dimer in the form of NiCo-N₆ structure, consistent with the HAADF-STEM results. (p. 12)

The NiCo-SAD-NC FT-EXAFS spectra were fitted with NiCo-N₆ dimer configuration with R-factor values of 0.0001 and 0.0016 for the Co and Ni K-edge EXAFS fitting, respectively, which were much smaller than the R-factor values of 0.0082 and 0.0199 for the Co and Ni K-edge EXAFS fitting, respectively, when fitted with Co/Ni-N₄ single atom configuration, suggesting that the NiCo-SAD-NC FT-EXAFS spectra were best fitted with the NiCo-N₆ dimer configuration, confirming that a significant amount of dimer structure was present in the prepared NiCo-SAD-NC material, consistent with the HAADF-STEM results. (p. S21)

Text already presents in the theoretical portion of the original manuscript (p. 6 and 7):

“Fig. 1c revealed that the NiCo-SAD-N₆C exhibited the lowest transition state energy barrier for H₂O dissociation compared to other TM-SADs, suggesting that the NiCo-SAD-N₆C with the highest H₂O dissociation rate can be regarded as a promising candidate for alkaline HER (Supplementary Table 2a).”

“Additionally, for the Co and Ni single-atom sites (Co-SA-N₄C and Ni-SA-N₄C), the calculation revealed that the hydronium (H⁺) and hydroxide (OH⁻) species auto-recombined to generate H₂O molecule after the optimization process in the final state of H₂O dissociation, suggesting that only Co and Ni single-atom sites were extremely sluggish for H₂O dissociation, further validating the beneficial role of synergistic-interaction at the Ni-Co atomic-interface in the SAD configurations.”

“By comparing the |U_L| of the TM-SADs, three potential candidates with the lowest theoretical overpotentials were observed in the following order: NiCo-SAD-N₆C (0.460 eV) < CoMn-SAD-N₆C (0.465 eV) < CoFe-SAD-N₆C (0.472 eV)”

Fig. 2 | Structural analysis and electron microscopy of NiCo-SAD-NC. **a**, XRD spectra of NC, NiCo-NP-NC, and NiCo-SAD-NC. **b**, Aberration-corrected HAADF-STEM image of the NiCo-SAD-NC. The yellow squares in **(b)** show the dimer sites, and some of the single Ni/Co-atom sites are highlighted by orange circles. **c,d**, The intensity profile and corresponding EEL spectra obtained at site A **(c)** and site B **(d)** for NiCo-SAD-NC indicate that Ni and Co are coordinated at the atomic scale. **e**, Statistical Ni-Co distance in the observed dimers.

Supplementary Figure 11 | **a**, Distribution histogram showing the ratio between dimers and single atoms. **b**, Overlapping EDS map of NiCo-SAD-NC showing the uniform distribution of Ni, Co and N atoms.

Supplementary Figure 19 | **a**, and **b**, The Ni K-edge, and Co K-edge FT-EXAFS, respectively, of NiCo-SAD-NC along with Ni/Co metal fitted with NiCo-N6 dimer configuration. **c**, and **d**, The Ni K-edge, and Co K-edge FT-EXAFS, respectively, of NiCo-SAD-NC along with Ni/Co metal fitted with Ni/Co-N4 single atom configuration.

Supplementary Table 5. EXAFS fitting parameters at the Ni and Co K-edge for NiCo-SAD-NC and Ni/Co metal ($S_0^2 = 0.780$ (Ni), 0.816 (Co)).

Sample	Shell	N^a	$R(\text{\AA})^b$	$\sigma^2(\text{\AA}^2)^c$	$\Delta E_0(\text{eV})^d$	R factor
Co K-edge (fitted with CoNi-N6 dimer configuration)						
Co Metal	Co-Co	12	2.49	0.0060	6.6	0.0002
NiCo-SAD-NC	Co-N	3.1	2.01	0.0018	-4.5	0.0001
	Co-Ni	0.6	2.55	0.0126		
Ni K-edge (fitted with NiCo-N6 dimer configuration)						
Ni Metal	Ni-Ni	12	2.48	0.0059	6.3	0.0001
NiCo-SAD-NC	Ni-N	3.3	1.86	0.0039	-4.8	0.0016
	Ni-Co	0.7	2.55	0.0121		
Co K-edge (fitted with Co-N4 single atom configuration)						
NiCo-SAD-NC	Co-N	4.0	2.03	0.0094	-2.1	0.0082
Ni K-edge (fitted with Ni-N4 single atom configuration)						
NiCo-SAD-NC	Ni-N	4.0	1.83	0.0064	-8.3	0.0199

^a N : coordination numbers; ^b R : bond distance; ^c σ^2 : Debye-Waller factors; ^d ΔE_0 : the inner potential correction. R factor: goodness of fit. S_0^2 was set to 0.816 for Co and 0.780 for Ni, according to the experimental EXAFS fit of Co and Ni metal reference by fixing CN as the known crystallographic value.

Supplementary Figure 22 | a, and b, HER LSV polarization curve of NiCo-SAD-NC with Ni/Co-SA-NC and different ratio of Ni to Co, respectively, in 1 M KOH. c, HER LSV polarization curve of NiCo-SAD-NC with Ni/Co-SA-NC in 0.5 M H₂SO₄.

4) The EDS elemental mapping with the different elements provided on figure S11 is interesting, but it does not support the author's conclusions. The colocalisation of Ni, Co and N is far from obvious from this image. and does not support the dimeric structure proposed.

Answer 4) We agree with the referee that the superimposed Ni, Co, and N EDS elemental mapping of NiCo-SAD-NC does not provide any information regarding the dimeric structure. As the scale bar is 100 nm, hence it is impossible to detect dimer sites from the EDS mapping. Therefore, we changed our previous conclusion to “HAADF-STEM and EDS elemental mapping revealed that N, Ni, and Co atoms were homogeneously dispersed in the NiCo-SAD-NC, rather than any possible aggregations in the form of nanoparticles (Fig. 2f and Supplementary Fig. 11b)”. The text is modified on p. 10 of the revised manuscript as

Meanwhile, HAADF-STEM and EDS elemental mapping revealed that N, Ni, and Co atoms were homogeneously dispersed in the NiCo-SAD-NC, rather than any possible aggregations in the form of nanoparticles (Fig. 2f and Supplementary Fig. 11b).

Supplementary Figure 11 | **a**, Distribution histogram showing the ratio between dimers, and single atoms. **b**, Overlapping EDS map of NiCo-SAD-NC showing the uniform distribution of Ni, Co and N atoms.

5) In XPS, the collection of data for reference samples is a very good thing. But what does the peak at 785 eV mean in the Co 2p_{3/2} spectrum?

Answer 5) We thank the referee for the comment, and we are pleased to clarify the issue. We apologize for not marking the peak at 785 eV in the Co 2p_{3/2} high-resolution XPS spectra. Now, we clearly marked the peak at 785 eV in the Co 2p_{3/2} high-resolution XPS spectra and revised the **Supplementary Fig. 14c**. The peak at 785 eV in the Co 2p_{3/2} high-resolution XPS spectra (**Supplementary Fig. 14c**) can be ascribed to the presence of a shake-up satellite peak [*Chem. Sci.*, **7**, 5758–5764 (2016); *Applied Catalysis A, General* **543**, 61–66 (2017)].

Supplementary Figure 14 | **a**, and **b**, Fitted deconvoluted Ni $2p_{3/2}$ XPS spectra and the Ni oxidation states analyzed by the XPS peak position, respectively, of NiCo-SAD-NC and Ni-SA-NC with reference samples. **c**, and **d**, Fitted deconvoluted Co $2p_{3/2}$ XPS spectra and the Co oxidation states analyzed by the XPS peak position, respectively, of NiCo-SAD-NC and Co-SA-NC with reference samples.

6) Although we appreciate the use of NiPc and CoPc as appropriate reference samples, the analysis of the XANES pre-edge region and the conclusions drawn from there is still not convincing. In CoPc, the pre-edge features at 7709 and 7715 eV are linked to each other and related to the planarity of the system. When the system is very flat (D_{4h}), then the first feature is supposed to be low and the second one is high (see DOI:10.1039/c4ra15306e). As a consequence, a strong first peak associated to a low intensity second peak indicates a deviation from the D_{4h} geometry. The explanation given by the authors that the deviation is due a substitution of the 4th nitrogen in Pc by the second metallic center is not satisfactory. This explanation was given in the first place by Ren et al. in their *Angewandte* paper (DOI: 10.1002/anie.201901575), but their explanation is not convincing to me either. It is not clear at all how this substitution would change the geometry, and the DFT calculated structure does not seem to show a serious deviation from a D_{4h} geometry. This explanation is actually not impossible, but a lot of other explanations could also be valid.

Answer 6) We sincerely thank the referee for carefully reviewing our manuscript and we are pleased to clarify the issue. We also thank the referee for providing clear information regarding the pre-edge features in the XANES of CoPC/NiPC. We agree with the referee that the pre-edge features in CoPC (7709 and 7715 eV) and NiPC (8333.8 and 8339 eV) are related to the planarity (D_{4h} symmetry) of the system [*RSC Adv.*, **5**, 9374–9380 (2015); *Nat. Energy* **3**, 140–147 (2018)]. For a perfect D_{4h} symmetry system, the intensity of the first peak is supposed to be very weak, and the second peak is high, which is the case in our reference NiPC and CoPC samples (Supplementary Fig. 15a,b). Compared to NiPC and CoPC, the increased intensity of the first pre-edge peak at 8333.8 and 7709 eV in the Ni and Co K-edge XANES spectra, respectively, of NiCo-SAD-NC were ascribed to the increased dipole allowed transition ($1s \rightarrow 4p$), which occurred due to the mixing of $3d$ and $4p$ orbitals as a result of the distorted D_{4h} symmetry [*Nat. Energy* **3**, 140–147 (2018)]. In contrast, the intensity of the second pre-edge peak at 8339 and 7715 eV ($1s \rightarrow 4p_z$) in the Ni and Co K-edge XANES spectra, respectively, of NiCo-SAD-NC were reduced compared to the NiPC/CoPC, thus further confirming the distorted D_{4h} symmetry of the Ni and Co atoms centers in the NiCo-SAD-NC [*Nat. Energy* **3**, 140–147 (2018)]. However, we agree with the referee that the above analysis does not provide convincing information that the distorted D_{4h} symmetry of the Ni and Co atom centers is due to the substitution by the second metal center. Therefore, we removed our previous conclusion that the distorted D_{4h} symmetry was due to another possible coordination such as metal-metal and also removed the cited *Angewandte* reference [*Angew. Chem. Int. Ed.* **58**, 6972–6976 (2019)].

The top view of the DFT structure previously provided in Supplementary Fig. 19c was just to provide the visual representation of the coordination between Co/Ni-N and Ni-Co. However, since the coordination number and bond length were already provided in Supplementary Table 5 and the DFT structure does not provide any information regarding the deviation from the D_{4h} geometry, therefore, we removed that DFT structure. The text is modified on p. 11 and S17 of the revised manuscript and supporting information, respectively, as

Compared to NiPC, the increased pre-edge peak intensity in NiCo-SAD-NC was ascribed to the distorted D_{4h} local symmetry, further implying that the single Ni atom was coordinated with four nearest guest atoms (N or metal) with distorted D_{4h} symmetry (Supplementary Fig. 15a)²⁸. (p. 11)

The increased intensity of the first pre-edge peak at 8333.8 and 7709 eV in the Ni and Co K-edge XANES spectra, respectively, of NiCo-SAD-NC, compared to NiPC and CoPC were ascribed to the increased dipole allowed transition ($1s \rightarrow 4p$), which occurred due to the mixing of $3d$ and $4p$ orbitals as a result of the distorted D_{4h} symmetry. In contrast, the intensity of the second pre-edge peak at 8339 and 7715 eV ($1s \rightarrow 4p_z$) in the Ni and Co K-edge XANES spectra, respectively, of NiCo-SAD-NC were reduced compared to the NiPC/CoPC, thus further confirming the distorted D_{4h} symmetry of the Ni and Co atoms centers in the NiCo-SAD-NC (Supplementary Fig. 15). (p. S17)

Supplementary Figure 15 | **a**, Experimental Ni K-edge XANES spectra, of NiCo-SAD-NC with Ni-SA-NC and NiPC (Inset in **a** show the pre-edge feature). **b**, Experimental Co K-edge XANES spectra, of NiCo-SAD-NC with Co-SA-NC and CoPC (Inset in **b** show the pre-edge feature).

The novelty of our work:

We thank the referee's comments on the novelty of our work, and we are pleased to clarify this issue. We agree that there are few recently published works on dual atom pair catalysts [*Energy Environ. Sci.* **11**, 3375-3379 (2018); *Angew. Chem. Int. Ed.* **131**, 2648–2652 (2019); *Angew. Chem. Int. Ed.* **58**, 6972–6976 (2019)], which we already referred to in our original manuscript's introduction portion. However, evaluating the impact/novelty of work should reply on what it has done to solve the remaining scientific or engineering challenges/questions of the studied chemical reaction or material, not only on the number of published papers. As we have mentioned in our introduction that the main issue related to single-atom catalysts (SACs) is that the isolated single metal sites coordinated with the neighboring nitrogen atoms in the carbon matrix (M-NC) are only capable of catalyzing simple elementary reactions, mainly due to the simplicity of the single-atom center. Because of the simplicity of the M-NC single-atom center, the possibilities for further modification of the active site in SACs are extremely limited, hindering their wide range of applications. Therefore, simple M-NC SACs are insufficient for catalyzing HER in both alkaline/acidic media since the pH-universal HER is a complex process involving additional water dissociation steps along with proton adsorption and dimerization to molecular H₂.

To the best of our knowledge, there is no report on the fabrication of pH-universal transition metal-based HER catalyst with targeted dimeric sites at atomic precision along with appropriate identification of the dimeric structure and deeper understanding of the dual metal atom synergism for catalyzing complex pH-universal HER. Our high-throughput DFT calculations indicated that the conventional Ni and Co SACs (Ni-SA-NC and Co-SA-NC) were extremely sluggish for water dissociation and exhibited poor performance for pH-universal HER. However, after successfully generating the NiCo-SAD atomic interface via our synthetic approach, we found that the electronic d-band center of the NiCo-SAD was significantly upshifted nearest to the fermi level compared to Ni/Co-SA-NC, which facilitated rapid water dissociation and optimal proton adsorption, accelerating pH-universal HER kinetics. Experimentally, the obtained NiCo-SAD made solely of earth-abundant elements exhibited comparable pH-universal HER activity to that of state-of-the-art Pt/C, which is a rare occurrence and much superior to that of conventional Ni-SA-NC, Co-SA-NC, and NiCo-NPs. The HER mass activity of the NiCo-SAD in acidic/alkaline media was much superior to that of Pt/C and the NiCo-NPs counterpart. Taken together, we reasonably believe our idea or approach to obtain a SAD atomic interface for modulating the electronic structure beyond SACs for achieving pH universal HER catalysis along with proper identification of the active site is conceptually novel and can also be extended to the other fields of materials science and electrocatalysis. And the existence of the above-mentioned reference papers could not affect the novelty of our work.

We truly thank Reviewer 1 for the insightful comments and kind suggestions.

Reviewer: 2

Comments: All my concerns are well addressed and I believe this manuscript is now acceptable for publication.

Response: We are grateful for the time and effort Reviewer 2 has spent in reviewing our manuscript. We appreciate the referee for the kind comment.

Reviewer: 3

Comments: I think the revised manuscript have meet the high standards of the Nature Communications.

Response: We are grateful for the time and effort Reviewer 3 has spent in reviewing our manuscript. We thank the referee for appreciating our revised manuscript and for the kind comment.

REVIEWERS' COMMENTS

Reviewer #1 (Remarks to the Author):

I am now convinced by the new formulations and analyzes that the authors included in the paper. The amount and quality of the data provided is truly impressive and I congratulate the authors for their achievement.

Point-by-point response for Nature Communications manuscript
(ID: NCOMMS-21-19692B)

Manuscript Type: Article

Title: Moving Beyond Bimetallic-Alloy to Single-Atom Dimer Atomic-Interface for All-pH Hydrogen Evolution.

Author(s): Ashwani Kumar, Viet Q. Bui, Jinsun Lee, Lingling Wang, Amol R. Jadhav, Xinghui Liu, Xiaodong Shao, Yang Liu, Jianmin Yu, Yosep Hwang, Huong T. D. Bui, Sara Ajmal, Min Gyu Kim, Seong-Gon Kim, Gyeong-Su Park, Yoshiyuki Kawazoe, and Hyoyoung Lee.

We are grateful to the editor, editorial staff, and reviewers for their critical comments and valuable suggestions. The manuscript has been modified after addressing all the suggestions as listed below:

Author(s)' Points-by-points responses to Reviewer(s)':

Reviewer: 1

Comments: I am now convinced by the new formulations and analyzes that the authors included in the paper. The amount and quality of the data provided is truly impressive and I congratulate the authors for their achievement..

Response: We are grateful for the time and effort Reviewer 1 has spent in reviewing our manuscript. We thank the referee for appreciating our revised manuscript and for the kind comment.